

# Acid gases and aerosol measurements in the UK (1999-2015): regional distributions and trends

Y. Sim Tang[1], Christine F. Braban[1], Ulrike Dragosits[1], Ivan Simmons[1], David Leaver[1], Netty van Dijk[1], Janet Poskitt[2], Sarah Thacker[2], Manisha Patel[2], Heather Carter[2], M. Glória Pereira[2], Patrick O. Keenan[2], Alan Lawlor[2], Christopher Conolly[3], Keith Vincent[3], Mathew R. Heal[4] & Mark A. Sutton[1]

[1]CEH, Bush Estate, Penicuik, Midlothian EH26 0QB
[2]CEH, Lancaster Environment Centre, Bailrigg, Lancaster LA1 4AP
[3]Ricardo Energy & Environment, Gemini Building, Fermi Avenue, Harwell, Oxon, OX11 0QR
[4]School of Chemistry, University of Edinburgh, David Brewster Road, Edinburgh EH9 3FJ

*Correspondence to*: Y. Sim Tang (yst@ceh.ac.uk)

**Abstract.** The UK Acid Gases and Aerosol Monitoring Network (AGANet) was established in 1999 (12 sites, increased to 30 sites from 2006), to provide long-term national monitoring of acid gases ($HNO_3$, $SO_2$, HCl) and aerosol components ($NO_3^-$, $SO_4^{2-}$, $Cl^-$, $Na^+$, $Ca^{2+}$, $Mg^{2+}$). An extension of a low-cost denuder-filter pack system (DELTA) that is used to measure $NH_3$ and $NH_4^+$ in the UK National Ammonia Monitoring Network (NAMN) provides additional monthly speciated measurements for the AGANet. A comparison of the monthly DELTA measurement with averaged daily results from an annular denuder system showed close agreement, while the sum of $HNO_3$ and $NO_3^-$ and the sum of $NH_3$ and $NH_4^+$ from the DELTA are also consistent with previous filter pack determination of total inorganic nitrogen and total inorganic ammonium, respectively. With the exception of $SO_2$ and $SO_4^{2-}$, the AGANet provides for the first time the UK concentration fields for each of the other measured species. The ranges in site-annual mean concentrations (nmol m$^{-3}$) in 2015 for the gases were: $HNO_3$: 0.7-17; HCl: 2.4-21; $SO_2$: 0.9-10, while those for aerosol were: $NO_3^-$: 6.3-53; $Cl^-$: 22-89; $SO_4^{2-}$: 2.2-11; $Na^+$: 20-74; $Ca^{2+}$: <lod-2.6; $Mg^{2+}$: 1.3-6.8. The largest concentrations of $HNO_3$, $SO_2$, and aerosol $NO_3^-$ and $SO_4^{2-}$ are found in south and east England and smallest in western Scotland and Northern Ireland. For HCl, highest concentrations are in the southeast and southwest, that may be attributed to dual contribution from anthropogenic (coal combustion) and marine sources (reaction of sea salt with acid gases to form HCl). The spatial distributions of $Na^+$ and $Cl^-$ were similar, with largest concentrations at coastal sites in the south and west and at Shetland, reflecting a contribution from sea salt (NaCl) since a near 1:1 relationship was also observed in their concentrations. Temporally, peak concentrations in $HNO_3$ and $NO_3^-$ occurred in late spring and early summer, due to photochemical processes and transboundary pollutant transport. The spring peak in $SO_4^{2-}$ concentrations coincides with the peak in concentrations of $NH_3$ and $NH_4^+$, and are therefore likely attributable to formation of $(NH_4)_2SO_4$ from reaction with higher concentrations of $NH_3$ in spring. By contrast, peak concentrations of $SO_2$, $Na^+$ and $Cl^-$ during winter are consistent with combustion sources for $SO_2$ and marine sources in winter for sea salt aerosol. Key pollutant events were captured by the AGANet. In 2003, a spring episode with elevated concentrations of $HNO_3$ and $NO_3^-$ was driven by meteorology and transboundary transport of $NH_4NO_3$ from Europe. A second, but smaller episode occurred in September 2014, with elevated



concentrations of SO$_2$, HNO$_3$, SO$_4^{2-}$, NO$_3^-$ and NH$_4^+$ that was shown to be from the Icelandic Holuhraun volcanic eruptions. Since 1999, AGANet has shown substantial decrease in SO$_2$ concentrations relative to HNO$_3$ and NH$_3$, accompanied by large reductions also in the aerosol components, with concentrations of NO$_3^-$ and NH$_4^+$ in molar excess over SO$_4^{2-}$. At the same time, a positive trend in HNO$_3$:NO$_3^-$ and NH$_3$:NH$_4^+$ ratios, contrasting with a negative trend in SO$_2$:SO$_4^{2-}$ ratio provides evidence of

a change in the particulate phase from (NH$_4$)$_2$SO$_4$ to NH$_4$NO$_3$, with indications that atmospheric lifetime of HNO$_3$ and NH$_3$ has increased. Due to different removal rates of the component species by wet and dry deposition, this change is expected to affect spatial patterns of pollutant deposition with consequences for sensitive habitats with exceedance of critical loads of acidity and eutrophication. The changes are also relevant for human health effects assessment, particularly in urban areas as NH$_4$NO$_3$ constitutes a significant fraction of fine particulate matter ($< 2.5$ µm) that are linked to increased mortality from

respiratory and cardiopulmonary diseases.

## 1    Introduction

Monitoring the atmospheric concentrations of acid gases and their aerosol reaction products is important for assessing their effects on human health, ecosystems, long-range transboundary transport and global radiative balance. Concentration data are necessary for quantifying long-term trends and spatial patterns, understanding gas-aerosol phase interactions, and estimating

the contributions of different pollutants to dry deposition fluxes (ROTAP, 2012; AQEG, 2013a; Colette et al., 2016), as well as to provide data for testing the performance of atmospheric models (e.g. Chemel et al., 2010; Vieno et al. 2014, 2016). Acid gases in the atmosphere include sulphur dioxide (SO$_2$), nitrogen oxides (NO$_x$), nitric acid (HNO$_3$), hydrochloric acid (HCl) and nitrous acid (HONO). Secondary inorganic aerosols (SIA) include sulphate (SO$_4^{2-}$), nitrate (NO$_3^-$), chloride (Cl$^-$) and nitrite (NO$_2^-$) that are formed from reactions of SO$_2$ and NO$_x$ (and HNO$_3$, a secondary product of NO$_x$) with ammonia (NH$_3$) in the

atmosphere. These aerosols make an important contribution to concentrations of particulate matter (PM) in the UK (15 to 50 % of the mass of atmospheric PM) and constitute a significant fraction of fine particles that are less than 2.5 µm in diameter (PM$_{2.5}$) implicated in harming human health (AQEG, 2012; 2013b). In addition, base cations in aerosol are also of interest to estimate the extent to which acidity is neutralized and to estimate the contribution of marine influences (ROTAP, 2012; Werner et al., 2011).

Anthropogenic emissions of SO$_2$, NO$_x$, HCl and NH$_3$ in the UK declined by 81, 51, 87 and 13 %, respectively, over the period 1999 to 2015 (Defra, 2017; NAEI 2017). Despite the success in mitigating SO$_2$ emissions however, sulphur still remains a pollutant of national importance, because reduction in sulphur deposition in remote sensitive areas have been more modest than close to major sources (ROTAP, 2012). HCl was also recently identified as another important acidifying pollutant for

sensitive habitats (Evans et al., 2014). Emissions of HCl (from coal burning in power stations) have however declined to very low levels (from 74 kt in 1999 to 9 kt in 2015), although it could still pose a threat to habitats close to sources. For NO$_x$, the





more modest decrease in emissions reflects difficulties in their abatement, while for $NH_3$, the decrease to date is largely a result of changes in animal numbers (Defra 2017).

With the decline in $SO_2$ emissions and deposition, the large number of reactive nitrogen compounds in the atmosphere are assuming greater importance owing to the complexities of the global N cycle and associated challenges in their abatement. These include the gas phase components $NH_3$, with over 80% estimated from agricultural emissions (EEA, 2017) and nitrogen oxides (NO, $NO_2$) from combustion, the secondary gas phase reaction products $HNO_3$, HONO and PAN (peroxyacetyl nitrate) and particulate phase components (($NH_4)_2SO_4$, $NH_4HSO_4$, and $NH_4NO_3$) formed by the reaction between $NH_3$ and acid gases. (AQEG, 2012). Ammonia and the N-containing aerosols are known to cause nitrogen enrichment and eutrophication, as well as contributing to acidification processes (Sutton et al., 2011). Oxidised nitrogen species ($NO_x$) are precursors to ground-level $O_3$ formation, while the production of acids ($HNO_3$, HONO) and PAN in the atmosphere affects air quality and is damaging both to human health and to vegetation (Cowling et al., 1998, Bobbink et al., 2010).

In Europe, air pollution policies regarding acidification and nitrogen eutrophication apply the "critical loads approach" (Bull, 1995; Gregor et al., 2001), which requires that atmospheric deposition inputs be mapped at an appropriate scale for the assessment of effects. In parallel, the "critical levels" of concentrations addresses the direct impacts of concentrations of nitrogen components in the atmosphere (Bull, 1991; Gregor et al., 2001; Cape et al., 2009). Quantifying the dry deposition of reactive nitrogen compounds is a major challenge and a key source of uncertainty for effects assessment (Dentener et al., 2006; Flechard et al., 2011; Schrader et al., 2018; Sutton et al., 2007). While deposition may be estimated using atmospheric transport and chemistry models (e.g. Dore et al., 2015; Flechard et al., 2011; Smith et al., 2000), air concentration data at sufficient spatial resolution are needed, both to assess the atmospheric models and provide input data for estimating deposition using inferential models.

In light of policies to reduce atmospheric emissions, e.g. the amended 2012 Gothenburg Protocol (UNECE, 2018) and the revised National Emissions Ceilings Directive (NECD, EU Directive 2016/2284) (EU, 2016), it is important to assess long-term trends in the measured pollutants, since this provides the only independent means to assess the effectiveness of any abatement policies. Both these international agreements set emissions reduction commitments for $SO_2$, $NO_x$ and $NH_3$, of 59, 42, and 6 %, respectively, by 2020 (with 2005 as base year) and includes $PM_{2.5}$ for the very first time. Under the 2016 NECD, further reduction commitments of 79 % ($SO_2$), 63 % ($NO_x$) and 19 % ($NH_3$) are also set for the EU 28 countries from 2030. Since emissions of these gases comes from different sources, emissions controls require very different strategies, making it important to monitor and assess the  the relative concentrations and deposition of nitrogen and sulphur components.

The spatial and temporal patterns of gases and particulate phases of these pollutants differ substantially. Although it is widely acknowledged that speciation between reactive gas and aerosol measurement is critical, there are few national long-term




monitoring programmes dedicated to measuring separately their concentrations and dry depositions at high spatial resolution (Torseth et al., 2012). Across Europe, the European Monitoring and Evaluation Programme (EMEP, 2014) continues to recommend using a daily filter pack sampling method to measure oxidised nitrogen (total inorganic nitrate, TIN) and reduced nitrogen (total inorganic ammonia, TIA) (Torseth et al., 2012; Colette et al., 2016). The filter pack method is generally

considered as robust for measuring $SO_2$ and $SO_4^{2-}$ concentrations (EMEP, 2014; Hayman et al., 2006; Sickles et al., 1999). However, many papers have shown that there are potential artefacts in filter-pack sampling for $HNO_3$ and $HCl$, due to interactions with $NH_3$ and the volatility of $NH_4NO_3$ and $NH_4Cl$ aerosol (Pio, 1992; Sickles et al., 1999; Cheng et al., 2012). Results from EMEP filter pack measurements are therefore reported as TIN and TIA, due to phase uncertainties in the method (Torseth et al., 2012). This has been complemented by daily measurements of $HNO_3$ and $NO_3^-$ using annular denuders

(Allegrini et al., 1987; EMEP, 2014) that are made at a restricted number of sites because of the resources required. In North America, filter pack sampling is also used in weekly measurements of sulphur and nitrogen species in the CASTnet (Clean Air Status Trends Network) national monitoring network of 95 sites across USA, Canada and Alaska (https://www.epa.gov/castnet). At a small number of CASTnet sites, hourly measurements of water-soluble gases and aerosols are made with the Monitor for AeRosols and GAses in ambient air (MARGA) system (Rumsey and Walker, 2016). In the UK,

the MARGA approach is deployed for continuous hourly measurements at just two sites (Twigg et al., 2016).

High time-resolution measurements of gases and aerosols are useful at selected locations for detailed analysis and model testing, but the high costs and resources required for these measurements make them unsuitable for the assessment of long-term trends at many sites, particularly where spatial patterns are required. To achieve this, a larger number of sites operated at

lower time-resolution is needed. In the UK, the Eutrophying and Acidifying Atmospheric Pollutants (UKEAP) network provides long-term measurements for the UK rural atmospheric concentrations and deposition of air pollutants that contribute to acidification and eutrophication processes (Conolly et al., 2016). UKEAP comprises of two EMEP supersites and four component networks: precipitation network (Precip-net), $NO_2$ diffusion tube network ($NO_2$-net), National Ammonia Monitoring Network (NAMN) and the Acid Gases and Aerosol Network (AGANet). At the two EMEP supersites

(Auchencorth and Harwell – relocated to Chilbolton in 2016), semi-continuous hourly speciated measurements of reactive gases and aerosols are made with the MARGA system (Twigg et al., 2016) that is also deployed at some CASTnet sites (Rumsey and Walker, 2016). These measurements are contributing to the validation and improvement of atmospheric models, such as FRAME (Dore et al., 2015) and EMEP4UK (Vieno et al., 2014, 2016) that are used to develop and provide the evidence base for air quality policies, both nationally and internationally.

The long-term dataset of monthly speciated measurements from the AGANet (1999 – 2015) are analysed in this paper to provide a comprehensive assessment of the spatial, temporal and long-term trends in atmospheric concentrations of the acid gases $HNO_3$, $SO_2$, $HCl$ and related aerosol components $NO_3^-$, $SO_4^{2-}$ and $Cl^-$ (and also base cations $Na^+$, $Ca^{2+}$ and $Mg^{2+}$) across the UK, together with an assessment of the DELTA denuder-filter pack sampling method (Sutton et al., 2001b; Tang et al.,





2009) as compared with other sampling techniques. To aid interpretation of the relative changes and trends in the acid gases and aerosols, $NH_3$ and particulate $NH_4^+$ data from the NAMN (Tang et al., 2018) are included, since atmospheric $NH_3$ is a major interacting precursor gas in neutralisation reactions with the acid gases.

## 2    Methods

### 2.1    Acid Gases and Aerosol monitoring Network (AGANet)

The UK Acid Gases and Aerosol Network (AGANet), known previously as the nitric acid monitoring network, was started in September 1999 under the Acid Deposition Monitoring Network (ADMN, Hayman et al., 2007) to deliver for the very first time, long-term monthly speciated measurement data on gaseous $HNO_3$ and particulate $NO_3^-$ across the UK. Other acid gases ($SO_2$, HCl) and aerosols ($SO_4^{2-}$, $Cl^-$, plus base cations $Na^+$, $Ca^{2+}$, $Mg^{2+}$) are also measured and reported. Since 2009, the AGANet, together with the NAMN (monthly $NH_3$ and $NH_4^+$), Precip-net (2-weekly wet deposition measurements) and $NO_2$-net (4-weekly $NO_2$ concentrations) were unified under the UKEAP network to provide long-term measurements of eutrophying and acidifying atmospheric pollutants (Conolly et al., 2016).

AGANet and NAMN are closely integrated, with AGANet established at a subset of NAMN sites to provide additional speciated measurements of the acid gases and aerosol components. To improve on national coverage, the number of sites in AGANet was increased in 2006 from 12 to 30 (Figure 1, Table 1). At the same time, the Rural Sulphur Dioxide Monitoring Program ceased, replaced by $SO_2$ and $SO_4^{2-}$ measurements made under the expanded AGANet (Hayman et al., 2007). A broad spatial coverage of the UK is provided by the AGANet sites, with a focus on sites providing parallel information on other air pollutants (e.g., co-location with the Automatic Urban and Rural Network that provides compliance monitoring against the Ambient Air Quality Directives (https://uk-air.defra.gov.uk/networks/) and ecosystem assessments (e.g. Environmental Change Network, http://www.ecn.ac.uk/) (Monteith et al., 2016).

**<INSERT Figure 1 HERE>**
**<INSERT Table 1 1HERE>**

### 2.2    Extended DELTA methodology for sampling acid gases and aerosol in AGANet

A low-cost manual denuder-filter pack method, DELTA (DEnuder for Long-Term Air sampling) implemented in the NAMN for measurement of $NH_3$ gas and aerosol $NH_4^+$ (Sutton et al., 2001a,b; Tang et al., 2018) is extended to provide additional simultaneous monthly time-integrated average concentrations of acid gases ($HNO_3$, $SO_2$, HCl) and particulate phase $NO_3^-$, $SO_4^{2-}$, $Cl^-$, $Na^+$, $Ca^{2+}$ and $Mg^{2+}$ for the AGANet (Conolly et al., 2016; Tang et al., 2015).





The DELTA method used in AGANet has also been applied in an extensive European-scale network of 58 sites to deliver 4 years of atmospheric concentrations and deposition data for reactive trace gas and aerosols from 2006 to 2009 (Tang et al., 2009; Flechard et al., 2011). Detailed descriptions of the DELTA method are provided by Sutton et al. (2001b) and by Tang et al. (2009, 2015). In brief, a small air pump is used to provide low sampling rates of 0.2–0.4 L min$^{-1}$, and air volumes are

measured by a high-sensitivity diaphragm gas meter. By sampling air slowly, the method is optimised for monthly measurements, with sufficient sensitivity to resolve low concentrations at clean background sites (e.g. LOD = 0.05 µg m$^{-3}$ for HNO$_3$ for monthly sampling; see Supplement Tables S1, S2). In addition, the power requirement is very small, and low voltage versions (using 6 V and 12 V micro-air pumps) of the system powered by wind-solar energy operate at some remote sites.

An extended denuder-filter pack sampling train is used to provide speciated sampling of reactive gases and aerosols (Supplement Figure S1) (Tang et al., 2009, 2015). A Teflon inlet (2.8 cm long) at the front end ensures development of a laminar air stream (Table S3), followed by a first pair of K$_2$CO$_3$ and glycerol coated denuders to collect HNO$_3$, SO$_2$ and HCl , a second pair of citric acid coated denuders to collect NH$_3$ and a 2-stage filter pack at the end to collect aerosol components. Stage 1 of the filter pack is a cellulose filter impregnated with K$_2$CO$_3$ and glycerol to collect NO$_3^-$, SO$_4^{2-}$, Cl$^-$, Na$^+$, Ca$^{2+}$, Mg$^{2+}$,

with evolved aerosol NH$_4^+$ from this filter collected on the stage 2 citric acid impregnated filter. The separation of gases and aerosol is achieved by higher diffusivities of reactive gases to the denuder walls where they react with the chemical coating and are retained, whereas aerosol components pass through and are retained by post-denuder filters (Ferm, 1979). In this approach, potential artefacts caused by phase interactions associated with filter packs and bubblers are avoided (e.g. Sickles et al., 1999).

For the base coating, K$_2$CO$_3$ is used instead of Na$_2$CO$_3$ (Ferm et al., 1986) to sample acid gases so that the system can also measure aerosol Na$^+$ concentrations. Glycerol increases adhesion, stabilizes the base coating (Ferm, 1986; Finn et al., 2001), and is reported to minimise potential oxidation of nitrite that is also collected on the denuder to nitrate in the presence of atmospheric oxidants such as ozone (Allegrini et al., 1987; Perrino et al., 1990). The lengths of denuders (borosilicate glass

tubes 10 cm and 15 cm long to capture > 95 % of NH$_3$ and acid gases, respectively) in the sampling train was calculated according to the procedures described by Sutton et al. (2001b), based on the calculations derived by Gormley and Kennedy (1948) and Ferm (1979) (see Table S3). All sites were set up as "outdoor" systems sampling directly from the atmosphere, avoiding potential adsorption losses (in particular HNO$_3$, which is highly surface active) and artefacts in air inlet lines. The sampling train is installed inside a simple watertight housing (Figure S1), which is mounted on a steel post in the desired

location. A low density polyethylene funnel (89 mm aperture) is placed at the inlet as a rain shelter, and sampling height is approx. 1.5 m.



### 2.3 Analytical methodology

#### 2.3.1 Base coated denuders and filters

Base-coated denuders and aerosol filters are extracted into 5 mL of deionised $H_2O$ for analysis. Anions ($NO_3^-$, $SO_4^{2-}$ and $Cl^-$) in the denuder and filter extracts are analysed by Ion Chromatography (IC). Base cations $Na^+$, $Mg^{2+}$ and $Ca^{2+}$ from the filter

extracts were analysed by IC between 1999 – Jun 2008 and by Inductively Coupled Plasma-Optical Emission Spectroscopy (ICP-OES/ICP-AES) from Jul 2008. Up to June 2009, analyses were carried out at Harwell Laboratory (Hayman et al., 2007) and from July 2009 at CEH Lancaster (Conolly et al. 2016). The limit of detection (LOD) for the DELTA method for the different components are calculated by analysing a series of laboratory blanks. The mean and standard deviation of the results are calculated and the LOD is calculated as three times the standard deviation divided by 15 $m^3$, the typical volume of air

sampled over a month by the DELTA system. Details of changes in laboratory, analytical methods and LODs for the gases and aerosols are summarised in Tables S1 and S2, respectively.

#### 2.3.2 Acid coated denuders and filters

Acid coated denuders and filter papers are also extracted into deionised $H_2O$ (3 mL and 4 mL, respectively), with analysis of $NH_4^+$ performed on a high sensitivity ammonia flow injection analysis system, as described in Tang et al., (2018).

### 2.4 Calculation of air concentrations

The air concentration ($\chi_a$) of a gas or aerosol is calculated according to equation 1 (see Sutton et al., 2001b, Tang et al., 2018):

$$\chi a = \frac{Q}{V} \qquad (1)$$

where    $Q$ = amount of a gas or aerosol collected on a denuder or aerosol filter, and

       $V$ = volume of air sampled (from gas meter, typically 15 $m^3$ in a month)

The denuder capture efficiency for each of the gas is calculated by comparing the concentrations of the individual gases in the denuder pairs and are applied in an infinite series correction on the raw data to provide corrected air concentrations ($\chi_{a \, (corrected)}$) according to equation 2 (see Sutton et al., 2001b, Tang et al., 2018):

$$\chi_a \, (corrected) = \chi_a \, (Denuder \, 1) \times \frac{1}{1 - \chi_a \left[ \frac{\chi_a (Denuder \, 2)}{\chi_a (Denuder \, 1)} \right]} \qquad (2)$$

Sutton et al. (2001b) and Tang et al. (2003) have shown that this procedure provides an important quality control, flagging up occurrences of poorly coated denuders and/or sampling issues. With denuder capture efficiency better than 90 %, the correction represents < 1 % of the corrected air concentration of the gas. Below 60 %, the correction is large (> 50 %) and is not applied, and the air concentration is then calculated as the sum of concentrations of the denuder pair. The amount of correction for gas





not captured that is added to the corrected gas concentration, is subtracted from the estimated aerosol concentrations of matching anions and cations (see Tang et al., 2018).

## 2.5    Data Quality Control

The following data quality checks are applied to the network data, as part of the network quality management system (Tang and Sutton, 2003; Conolly et al., 2016):

  i)    Air flow rate ($0.2 - 0.4$ L min$^{-1}$): where this is below the expected range for a sampling period, the data is flagged as valid but failing the QC standard.

  ii)   Denuder capture efficiency: where this is less than 75% for a sample, the data is flagged as valid but less certain.

iii)  Ion balance checks: close agreement expected between $NH_4^+$ and the sum of $NO_3^-$ and $2\times SO_4^{2-}$, as $NH_3$ is neutralised by $HNO_3$ and $H_2SO_4$ to form $NH_4NO_3$ and $(NH_4)_2SO_4$, respectively (Conolly et al., 2016), and for $Na^+$ and $Cl^-$, as these are marine (sea salt) in origin.

  iv)   Screening the whole dataset for sampling anomalies and outliers, e.g. due to contamination or other issues.

## 2.6    Bias correction applied to HNO₃ data

Tang et al. (2009, 2015) have identified that $HNO_3$ concentrations ($NO_3^-$ on denuders assumed to be from $HNO_3$) may be overestimated on carbonate coated denuders, due to partial co-collection of other oxidized nitrogen components such as nitrous acid (HONO). In the case of HONO, this collects on the denuder carbonate coating as nitrite ($NO_2^-$), but oxidizes to nitrate ($NO_3^-$) in the presence of oxidants such as ozone (Bytnerowicz et al., 2005) which can result in a bias in $HNO_3$ determination

(Tang et al., 2009, 2015). Other oxidised nitrogen species present in the atmosphere such as peroxyacetyl nitrate (PAN) and nitrogen oxides ($NO_x$) can also potentially contribute to a further small interference (Allegrini et al., 1987; Bai et al., 2003). Based on the tests of Tang et al. (2015), raw $HNO_3$ data are corrected with an empirical factor of 0.45 which is estimated to be uncertain by ±30 %. Apart from where stated, all $HNO_3$ data reported in this study have the 0.45 correction factor applied.

## 2.7    Performance of the DELTA method

### 2.7.1    Measurement reproducibility

Replicated DELTA measurements are made at the Bush OTC site in Scotland (UKA00128). A comparison of the parallel measurements (Figure 2) showed good reproducibility in the method, with close agreement for all components (e.g. mean difference of $< \pm 3$ % for all components and $\pm 6$ % for HCl).





**<INSERT Figure 2 HERE>**

### 2.7.2    Comparison with daily annular denuder measurements

An assessment of the DELTA method for $NH_3$ has previously been reported by Sutton et al. (2001b). Following the extension to additionally sample acid gases and aerosols, the modified system was compared with independent daily measurements from

an annular denuder system (ADS). The ADS (ChemspecTM model 2500 air sampling system, R&P Co. Inc.) was operated at Barcombe Mills in southern England (UKA00069) alongside the AGANet DELTA monthly measurements for a period of 18 months. Due to significant instrument and local site issues resulting in low data capture with the ADS, only 11 months of data were available for intercomparison. The sampling train used in the ADS consisted of 2 $K_2CO_3$ + glycerol-coated annular denuders (same coating as AGANet DELTA), 2 citric acid-coated annular denuders; a cyclone with 2.5 µm cut-off, followed

by a 2-stage filter pack containing a 2 µm PALL Zefluor teflon membrane (collection $NO_3^-$, $SO_4^{2-}$, $Cl^-$, $Na^+$, $Mg^{2+}$, $Ca^{2+}$) and a 1 µm PALL Nylasorb nylon membrane (collection of evolved $NO_3^-$), with a sampling rate of 10 L min$^{-1}$. For comparison against the monthly DELTA measurements, daily ADS values were averaged to the corresponding monthly periods, with results summarised in Table 2 and Figure S2.

**<INSERT Table 2 HERE>**

For $HNO_3$, the DELTA (mean = 1.56 µg m$^{-3}$, $n$ = 11) was on average 23 % higher than the ADS (mean = 1.31 µg m$^{-3}$, $n$ = 11). Since both methods used the same carbonate coating on the denuders to sample acid gases, the $HNO_3$ data here have not been corrected with the bias adjustment factor described in Sect. 2.6. Nitrous acid (HONO) was found to be close to or below limit

of detections for most of the DELTA measurements (mean = of 0.03 µg m$^{-3}$), compared with a significantly higher concentration (mean = 0.41 µg m$^{-3}$) from the ADS. Since the sampling period of the ADS is daily, any HONO collected as nitrite on the ADS is likely to remain as nitrite and not oxidised to nitrate. The very low HONO (nitrite on the denuders assumed to be from HONO) concentrations from the DELTA supports the hypothesis of the retention of HONO that is subsequently oxidised to nitrate, resulting in an artefact in $HNO_3$ determination (Possanzini et al., 1983; Allegrini et al., 1987;

Tang et al., 2015). Further corroboration is provided by the improved agreement between both methods (line of fit closer to the 1:1 line) when comparing the sum of $HNO_3$ and HONO (see graph in supp. Figure S2). Agreement between the DELTA and ADS was within 19 % for $SO_2$ (mean DELTA = 1.75 µg m$^{-3}$ *cf* mean ADS = 2.18 µg m$^{-3}$) and 4 % for HCl (mean DELTA = 0.40 µg m$^{-3}$ *cf* mean ADS = 0.41 µg m$^{-3}$). Given the limited data available, it is not clear why $SO_2$ measured on the ADS is higher than the DELTA, since there was good agreement for HCl.

For the particle-phase components, $NO_3^-$ measured by the DELTA method (mean = 2.59 µg $NO_3^-$ m$^{-3}$) was on average 2-fold higher than the ADS method (mean = 1.32 µg $NO_3^-$ m$^{-3}$), whereas $SO_4^{2-}$ by the DELTA method was on average 23 % lower (DELTA = 2.10 vs ADS = 2.74 µg $SO_4^{2-}$ m$^{-3}$) (Table 2). $NO_3^-$ and $SO_4^{2-}$ are both present as fine mode (< 1 µm) $NH_4NO_3$ and



$(NH_4)_2SO_4$ (Putaud et al., 2010). Some $NO_3^-$ can also be present in the coarse mode (> 2.5 µm), likely as calcium nitrate ($Ca(NO_3)_2$) from a reaction between gas-phase $HNO_3$ (or its precursors) and soil dust particles (Putaud et al., 2010). For $SO_4^{2-}$, some will be coarse mode sea salt $SO_4^{2-}$ (see section 3.3). A particle size cut-off of 4–5 µm was estimated for the DELTA air inlet) (Tang et al., 2015), which would suggest that the DELTA will also sample a small amount of coarse mode aerosols. An

ion balance check of the ratio of µeq $NH_4^+$ to sum µeq ($NO_3^- + SO_4^{2-}$) yielded a near unity value, which confirms that $NO_3^-$ and $SO_4^{2-}$ collected by the DELTA aerosol filter are mainly fine mode $NH_4NO_3$ and $(NH_4)_2SO_4$. In comparison, the ADS has a 2.5 µm cyclone in front of the aerosol filters to collect aerosols < 2.5 µm on the aerosol filters. $NH_4^+$ was unfortunately not analysed in these tests, which would have allowed a similar ion balance check. $Na^+$ and $Cl^-$ concentrations on the DELTA were also on average 331% and 444 % higher than on the ADS and the ion balance check of the ratio of $Na^+$:$Cl^-$ was unity for both methods.

In the absence of analytical errors, loss of $NO_3^-$, $Na^+$ and $Cl^-$ on the surface of the cyclone, coupled to a small fraction of the aerosols > 2.5 µm that is collected (but not analysed) in the cyclone, could partly account for the observed lower concentrations of the aerosol components. Since $Ca^{2+}$ and $Mg^{2+}$ concentrations by both methods were at or below detection limits, comparisons of these are not meaningful and have not been made.

### 2.7.3    Comparisons with filter pack measurements: $HNO_3/NO_3^-$ and $NH_3/NH_4^+$

The EMEP network (www.emep.int) measures atmospheric concentrations and depositions of a wide range of pollutants at rural background sites across Europe (Aas, 2014; Tørseth et al., 2012). For assessment of oxidised and reduced nitrogen species, the daily EMEP filter pack method are implemented at 39 sites across Europe (Colette et al., 2016; Tørseth et al., 2012), with results reported as Total Inorganic Nitrate (TIN: $HNO_3 + NO_3^-$) and Total Inorganic Ammonia (TIA: $NH_3 + NH_4^+$)

(Torseth et al., 2012), as these are considered more reliable than reporting for the gas and aerosol components separately.

**<INSERT Figure 3 HERE>**

At the UK Eskdalemuir site (EMEP station code GB0002R; UKAIR ID UKA00130), a Scottish rural background site on the

border between Scotland and England, daily filter pack measurements of TIN and TIA were made as part of the EMEP network from 1989 to 2000 (EMEP, 2017a). Following installation of the DELTA system in September 1999, both methods were operated in parallel for 14 months at Eskdalemuir, allowing a comparison to be made of TIN and TIA from both systems. Comparison results are shown in Figure 3 of parallel data from the AGANet (sum of $HNO_3$ and $NO_3^-$) and NAMN (sum of $NH_3$ and $NH_4^+$), demonstrating close agreement between the two independent measurements. The EMEP values shown are

daily measurements of TIN and TIA averaged to corresponding monthly means for comparison with the DELTA data. For TIN, the regression between EMEP TIN and AGANet (sum of uncorrected $HNO_3 + NO_3^-$) is close to unity (slope = 0.984, $R^2$ = 0.94), which provided independent verification and support of the DELTA $HNO_3$ measurements at the start of the network. After applying a bias adjustment factor of 0.45 to the $HNO_3$ data (see Sect. 2.6), the AGANet values (sum of corrected $HNO_3$





+ NO$_3^-$) are smaller than the EMEP TIN (slope = 0.835, $R^2$ = 0.95). It is possible that the filter pack method may also be subject to similar artefacts in HNO$_3$ determination due to co-collection of other oxidised nitrogen species (Tang et al., 2015).

### 2.7.4   Comparisons with bubbler and filter pack measurements: SO$_2$ and SO$_4^{2-}$

Independent measurements of SO$_2$ and SO$_4^{2-}$ with a daily bubbler and filter pack method, respectively, are also available for comparison with the DELTA method at the Eskdalemuir site. Daily SO$_2$ data with a bubbler method (Hayman, 2005) from Dec-77 to Dec-01 and daily SO$_4^{2-}$ data with an EMEP filter pack method from Dec-77 to Apr-09 (Hayman, 2006) were downloaded from the EMEP website (EMEP, 2017b). A close agreement is found between the bubbler and DELTA method for SO$_2$ (slope = 0.86, $R^2$ = 0.82), while there is more scatter between the filter pack and DELTA method for SO$_4^{2-}$ (slope = 0.67, $R^2$ = 0.66) (Figure 4). Concentrations of SO$_2$ for the 26 month overlap period were comparable (mean of bubbler method = 0.40 µg S m$^{-3}$ *cf* mean of DELTA method = 0.44 µg S m$^{-3}$), whereas the filter pack SO$_4^{2-}$ concentration (mean = 0.44 µg S m$^{-3}$, $n$ = 87) is larger than the corresponding monthly DELTA measurement (mean = 0.28 µg S m$^{-3}$, $n$ = 87) (Figure 4). An earlier detailed assessment of the DELTA system against filter pack with a focus on SO$_2$ and SO$_4^{2-}$ in 1999 by Hayman et al. (2006) had shown close agreement between the methods. It is therefore unclear why the DELTA gives a reading lower than the filter pack SO$_4^{2-}$ at Eskdalemuir in this assessment, since the dataset was a continuation of the original inter-comparison. Possible explanations include uncertainties associated with limit of detection of the daily filter pack method at the very low concentrations encountered at this site, or the sampling of coarser particles by this method (due to high flow rate and open-face sampling) with higher concentrations of sea salt sulphate. The DELTA methodology was unchanged for the duration of the AGANet dataset (1999 - 2015) in this manuscript, which allows a consistent assessment of overall trends in the SO$_4^{2-}$ data.

**<INSERT Figure 4 HERE>**

### 2.8   Time series trend analyses

Statistical trend analyses using both parametric linear regression (LR) and non-parametric Mann-Kendall (MK) (Gilbert, 1987; Chatfield, 2016) tests were performed on annually averaged data from AGANet, and on a subset of annually averaged data from NAMN made at the same AGANet sites. The datasets are considered sufficiently long-term (>10 years) and produced by a consistent method for effective statistical trend analyses. Both the LR and MK approaches are widely adopted for trend analyses in long-term atmospheric data (e.g. Meals et al., 2011; Colette, 2016; Jones and Harrison, 2011; Marchetto et al., 2013; Hayman et al., 2007; Conolly et al., 2016), and were used in a recent trend assessment of atmospheric NH$_3$ and NH$_4^+$ data (1998 – 2014) from the NAMN (Tang et al., 2018). As described in Tang et al. (2018), LR tests were performed using R, and MK tests used the R 'Kendall' package (McLeod, 2015), with estimation of the MK Sen's slope (fitted median slope of a linear regression joining all pairs of observations) and confidence interval of the fitted trend using the R 'Trend' package (Pohlert, 2016). Results from both tests provides an indication of uncertainty associated with the choice of approach.



## 3    Results and Discussion

### 3.1    AGANet data

Annual data from the AGANet (and also from the NAMN) are submitted to the Department for Environment, Food & Rural Affairs (Defra) UK-AIR database (https://uk-air.defra.gov.uk/), in a format consistent with other UK Authority air quality
networks and relevant reporting requirements. Every concentration value is labelled with a validity flag and an EMEP flag (see http://www.nilu.no/projects/ccc/flags/index.html). Ratified calendar year data are published from around June the year following collection. Currently, work is also in progress for the data to be made available from the EMEP database (http://ebas.nilu.no/). All data used in this paper (up to 2015), except where specified, are accessed from the UK-AIR website (Tang et al. 2017a, b).

### 3.2    Spatial patterns in relation to pollutant sources and transport

The spatial patterns for each of the gas and aerosol components measured are shown in the annual maps for the example year 2013 (Figure 5). A gradient in the concentrations of acid gases $HNO_3$ and $SO_2$, and related aerosols $NO_3^-$ and $SO_4^{2-}$ can be seen across the UK, highest in the south and east (combustion/vehicular sources and long-range transboundary pollutant
transport from Europe) and lowest in the north and west of the UK (fewer sources, furthest from influence of Europe).

The largest $HNO_3$ concentrations were measured at the London Cromwell site (2013 site annual mean = 1.3 µg $HNO_3$ m$^{-3}$ *cf* 2013 mean of 30 sites = 0.40 µg $HNO_3$ m$^{-3}$). London and Edinburgh  are the only two urban sites in the AGANet, with the other 28 sites all in rural environments. $HNO_3$ concentrations in Edinburgh, the capital of Scotland with a population that is
18 times smaller than London (0.5 million *vs* 8.8 million), is about 2 times lower than London, but higher than the national average (2013 annual mean = 0.58 µg $HNO_3$ m$^{-3}$). For $SO_2$, the highest concentrations were recorded at Sutton Bonington due to close proximity to the 2000 MW capacity coal-fired Ratcliffe-on-Soar power station (2 km North). A peak concentration of 10.9 µg $SO_2$ m$^{-3}$ was recorded in May 2000 at this site, with an annual mean concentrations of 5.9 µg $SO_2$ m$^{-3}$ for that year that was also 3 times higher than the national average (mean of 12 sites = 1.9 $SO_2$ m$^{-3}$ *cf.* mean of 11 sites (excl. Sutton
Bonington) = 1.5 $SO_2$ m$^{-3}$). At remote sites further away from sources, concentrations of $HNO_3$ and $SO_2$ are smaller, e.g. Lough Navar in Northern Ireland (2013 annual mean: 0.15 µg $HNO_3$ m$^{-3}$ and 0.21 µg $SO_2$) and Strathvaich Dam in northwest Scotland (2013 annual mean = 0.17 µg $HNO_3$ m$^{-3}$ and 0.18 µg $SO_2$). $NO_3^-$ and $SO_4^{2-}$ as secondary aerosols have longer residence times in the atmosphere and are expected to be more homogeneous than their precursor gases. The spatial distribution in concentrations of particulate $NO_3^-$ (0.33 – 3.1 µg m$^{-3}$) and $SO_4^{2-}$ (0.35 – 1.2 µg m$^{-3}$) are however similar to that of $HNO_3$ (0.12
– 1.3 µg m$^{-3}$) and $SO_2$ (0.10 – 1.1 µg m$^{-3}$), with no clear differences in the main regional patterns from only 30 sites.

**<INSERT Figure 5 HERE>**



For HCl, the highest concentrations are in the southeast and southwest of England, that may be attributed to dual contribution from anthropogenic (coal combustion) and marine sources (reaction of sea salt with $HNO_3$ and $H_2SO_4$ to form HCl) (Roth and Okada 1998; Ianniello et al., 2011). The spatial distributions of $Cl^-$ and $Na^+$ were similar, with largest concentrations at coastal sites in the south (Barcombe Mills) and west (Yarner Wood) and at Shetland, highlighting the importance of marine sources to the sea salt (NaCl) aerosol. Further away from the coast and influence of marine aerosol, the smallest concentrations of $Cl^-$ and $Na^+$ are measured in the west of the country (Lough Navar in Northern Ireland and Cwmystwyth in mid-Wales) and most of Scotland (with the exception of Shetland). For $Mg^{2+}$, the range of concentrations at sites are small ($0.03 - 0.19$ µg m$^{-3}$), but the spatial distribution is similar to $Na^+$ and $Cl^-$ and suggests that it may be in the form of $MgCl_2$. There is however no clear spatial pattern for $Ca^{2+}$, with concentrations that are mostly at or below LOD.

In the case of $NH_3$, the extensive spatial heterogeneity seen is related to large variation in emission sources at ground level across the UK (Tang et al., 2018). Aerosol $NH_4^+$, as expected for a secondary component, show a less variable concentration field. The spatial distribution of $NH_4^+$ is similar to $SO_4^{2-}$ and $NO_3^-$ over the UK (Figure 5), due to the close coupling between species from the formation of particle phase $(NH_4)_2SO_4$ and $NH_4NO_3$ (see next section).

### 3.3     Correlations between gas and aerosol species

Correlations between the gas and aerosol phases of the different components are summarised in Figure 6. The comparison of gas phase concentrations show that gaseous $NH_3$ is poorly correlated with either $SO_2$ or $HNO_3$, as might be expected since the emission sources of these pollutants are different. For $HNO_3$ and $SO_2$, there is a stronger correlation ($R^2 = 0.35$), which may be due to similarity in the regional distribution of their emissions. These comparisons show that there is on average 5 times more $NH_3$ than $SO_2$ and 13 times more $NH_3$ than $HNO_3$ at the AGANet sites (on a molar basis), and that $SO_2$ concentration is nearly 3 times larger than $HNO_3$ (on a molar basis).

<INSERT Figure 6 HERE>

For the aerosol components, there is very high correlation between $NO_3^-$, $SO_4^{2-}$ and $NH_4^+$, and between $Na^+$ and $Cl^-$, but no discernible relationship between $NH_4^+$ and $Cl^-$ (Figure 6). The near 1:1 relationship in the scatter plot of the sum of $NO_3^-$ and $SO_4^{2-}$ (µeq m$^{-3}$) *versus* $NH_4^+$ (µeq m$^{-3}$) (slope = 0.91, R$^2$ = 0.93), in the absence of any correlation between $NH_4^+$ and $Cl^-$, suggests that $H_2SO_4$ and $HNO_3$ in the atmosphere are fully neutralised by $NH_3$ to form $(NH_4)_2SO_4$, $NH_4HSO_4$ and $NH_4NO_3$ (Aneja et al. 2001). For $Cl^-$, the high correlation with $Na^+$ (slope = 1.04, R$^2$ = 0.8) lends support that the $Cl^-$ measured in the DELTA are derived mainly from sea salt (NaCl). Similar to the relative concentrations of gases, $NH_4^+$ concentrations (on a



molar basis) are larger than $SO_4^{2-}$ and $NO_3^-$, but $NO_3^-$ is in molar excess over $SO_4^{2-}$. The correlations between $NH_4^+$ and sum $(NO_3^- + 2 \times SO_4^{2-})$ and for $Na^+$ and $Cl^-$ forms the basis of ion balance checks in data quality assessment, as discussed in section 2.5 and shows that robust data are obtained .

Sea salt aerosol, derived from sea spray has essentially the same composition as seawater (Keene et al., 1986). The marine aerosol comprises two distinct aerosol types: (1) primary sea salt aerosol produced by the mechanical disruption of the ocean surface and (2) secondary aerosol, primarily in the form of non-sea salt (nss) sulphate and organic species, formed by gas-to-particle conversion processes such as binary homogeneous nucleation, heterogeneous nucleation and condensation (O'Dowd and Leeuw, 2007). It has been shown that the ratio of the mass concentrations of $SO_4^{2-}$ and $Cl^-$ to the reference $Na^+$ species in

seawater may be used to estimate mass concentrations of non-sea salt $SO_4^{2-}$ (nss_SO4) and non-sea salt Cl- (nss_Cl) in aerosol, according to equations 3 and 4, respectively (Keene et al., 1986; O'Dowd and de Leeuw, 2007).

$[nss\_SO4] = [SO_4^{2-}] - (0.25 \times [Na^+])$     (3)

$[nss\_Cl] = [Cl^-] - (1.80 \times [Na^+])$     (4)

Applying Equation 3 to the $SO_4^{2-}$ data in Figure 6, nss_SO4 is estimated to comprise on average 25 % (range = 3 – 83 %, $n$ = 187) of the measured $SO_4^{2-}$ aerosol. Regression of nss_SO4 $vs$ $NH_4^+$ (slope = 0.18, intercept = 0.47, $R^2$ = 0.71) (Figure S3) was not significantly different from the regression of total $SO_4^{2-}$ $vs$ $NH_4^+$ (slope = 0.18, intercept = 2.4, $R^2$ = 0.73) (Figure 6). Sources of nss_SO4 are (i) biological oxidation of dimethylsulphide and (ii) oxidation of $SO_2$ (O'Dowd and de Leeuw, 2007).

This analysis demonstrates that sea salt $SO_4^{2-}$ aerosol makes up a significant and variable fraction of the total $SO_4^{2-}$ measured, consistent with observations of the contribution by sea salt $SO_4^{2-}$ to the total $SO_4^{2-}$ in precipitation in the UK (ROTAP 2012). The improved intercept from the nss_SO4 regression (Figure S3) suggests that nss_SO4 are mainly associated with $NH_4^+$.

For $Cl^-$ (mean = 1.3 µg m$^{-3}$, $n$ = 188), estimated nss_Cl concentrations according to Equation 4 were negligible (mean = –0.09

µg m$^{-3}$, $n$ = 188). Studies have shown that part of the chloride of sea salt can be substituted by $SO_4^{2-}$ and $NO_3^-$ through a reaction with $H_2SO_4$ and $HNO_3$, known as the $Cl^-$ deficit (Ayers et al., 1999). The close coupling between $Cl^-$ and $Na^+$ (near 1:1 relationship) presented here suggests that the measured $Cl^-$ in the aerosol are mostly sea salt in origin, with no evidence of depletion of $Cl^-$ from sea salt aerosols.



### 3.4    Temporal trends in acid gases and aerosols

The average seasonal cycles for all gas and aerosol components derived from the mean of monthly data of all sites for the period 2000 to 2015 are compared in Figure 7. Clear differences are observed in these seasonal cycles, influenced by local to regional emissions, climate, meteorology and photochemical processes.

**<INSERT Figure 7 HERE>**

The seasonal cycle for $HNO_3$ and $NO_3^-$ has a maximum in spring (Figure 7). $HNO_3$ is a secondary product of $NO_x$, but $NO_x$ emissions are dominated by vehicular sources which are not expected to show large seasonal variations. Seasonal changes in chemistry and meteorology are therefore more likely to be a source of the observed variations in $HNO_3$ and $NO_3^-$. In spring,
the peak in $HNO_3$ and $NO_3^-$ concentrations are due to i) photochemical processes with elevated ozone in spring (AQEG 2009) leading to formation of $HNO_3$ during this period (Pope et al., 2016), and ii) long-range transboundary pollution from Europe, leading to regional enhancement across the UK (Vieno et al., 2014, 2016).

The ratio of the concentrations of $HNO_3$ and $NO_3^-$ can also be seen to fluctuate throughout the year; largest in July and smallest in spring (Figure 8) which may be explained as follows. An equilibrium exists between gaseous $HNO_3$, $NH_3$ and particulate
$NH_4NO_3$, the latter of which is appreciably volatile at ambient temperatures (Stelson and Seinfeld, 1982). The atmospheric residence times and therefore concentrations of $HNO_3$ and $NH_3$ are strongly dependent on their partitioning between the gas and aerosol phase. $HNO_3$ and $NH_3$ that are not removed by deposition may react together in the atmosphere to form $NH_4NO_3$ aerosol, when the concentration product $[NH_3].[HNO_3]$ exceeds equilibrium values. The formation and dissociation in turn are strongly influenced by ambient temperature and humidity. Warm, dry conditions in summer promotes dissociation, increasing
gas-phase $HNO_3$ relative to particulate-phase $NH_4NO_3$ (Figure 8). In winter, low temperature and high humidity shifts the equilibrium to formation of $NH_4NO_3$ from the gas-phase $HNO_3$ and $NH_3$. Since $NH_3$ concentrations are also lowest in winter, this contributes to the winter minimum in $NH_4NO_3$. The low $HNO_3:NO_3^-$ ratio in spring-time on the other hand is a result of higher $NO_3^-$, from higher concentrations of the precursor gases $HNO_3$ and $NH_3$, and from long-range transboundary transport of particulate $NO_3^-$ e.g. from continental Europe into the UK, as discussed in Vieno et al. (2014, 2016).

**<INSERT Figure 8 HERE>**

For $SO_2$, the temporal profile shows highest concentrations in the winter, with concentrations exceeding summer values on average by a factor of 2 (Figure 7). Higher emissions of $SO_2$ from combustion processes (heating) during the winter months,
coupled to stable atmospheric conditions resulting in build-up of concentrations at ground level contributes to the winter maximum. Since the reaction of $SO_2$ with $NH_3$ to form $(NH_4)_2SO_4$ is effectively irreversible (Bower et al., 1997), the ratio of the concentrations of $SO_2$ and $SO_4^{2-}$ (Figure 8) is largely governed by the availability of $SO_2$ and $NH_3$ to form $(NH_4)_2SO_4$. The





temporal profile of $SO_4^{2-}$ has a peak in concentrations in spring, although not as pronounced as the $NO_3^-$ peak (Figure 7). This may be attributed to enhanced formation of $(NH_4)_2SO_4$, since peaks in concentrations of $NH_3$ and $NH_4^+$ also occur in spring (Figure 7) and from the import of particulates from long-range transboundary transport.

$Na^+$ and $Cl^-$ also have highest concentrations during winter, highlighting the importance of marine sources (more stormy weather) in winter for sea salt aerosol. The seasonal trends in $Mg^{2+}$ is similar to $Na^+$, with maxima during winter and minima in summer (Figure 7). While sea salt aerosols comprise mainly of NaCl, other chemical ions are also common in seawater, such as $K^+$, $Mg^{2+}$, $Ca^{2+}$ and $SO_4^{2-}$ (Keene et al., 1986). Some of the sea salt aerosol may therefore be in the form of $MgCl_2$. Magnesium is however also a crustal element, and so it is not as good as sodium as a tracer for sea salt. Similarly, calcium is

also a rock-derived element and its presence in the atmosphere is thought to come from chemical weathering of carbonate minerals (Schmitt & Stille, 2005). The seasonal cycle of $Ca^{2+}$ is similar to, but less pronounced than $Na^+$ and $Mg^{2+}$. Measured concentrations of $Ca^{2+}$ were mostly at or below the method LOD which makes interpretation uncertain, but the higher concentrations of $Ca^{2+}$ in the winter months is likely to be both crustal dust and sea salt in origin.

**\<INSERT Figure 9 HERE\>**

Large inter- and intra-annual variability are also observed in the long-term mean monthly concentrations of gas and aerosol components, as illustrated in Figure 9. In 2003, elevated concentrations of $HNO_3$ and $NO_3^-$ (and also $NH_4^+$) were observed between February to April that were more pronounced than the normal peak in concentrations that occur in Spring. The large

spike in concentrations was of a sufficient magnitude to elevate the annual mean concentrations for 2003 of $HNO_3$ (0.54 µg m$^{-3}$ *cf* 0.39 and 0.36 µg m$^{-3}$ for 2002 and 2004, respectively), particulate $NO_3^-$ (2.98 µg m$^{-3}$ *cf* 1.99 and 1.93 µg m$^{-3}$ for 2002 and 2004, respectively) and $NH_4^+$ (1.45 µg m$^{-3}$ *cf* 1.06 and 0.97 µg m$^{-3}$ for 2002 and 2004, respectively). In comparison, a much smaller spike in elevated $SO_4^{2-}$ concentrations resulted in a slight increase in annual average $SO_4^{2-}$ (1.79 µg m$^{-3}$ *cf* 1.41 and 1.31 µg m$^{-3}$ for 2002 and 2004, respectively) (Figure 9). Meteorological back trajectory analysis of the period showed air

masses coming across the UK from Europe, and the pollution episode was attributed to the formation and transport of $NH_4NO_3$ from Europe, since other gases ($SO_2$, HCl and $NH_3$) and particulate $Cl^-$ were not affected (Vieno et al., 2014). At the same time, stable atmospheric conditions due to a persistent high pressure system over the UK led to an accumulation of pollutant concentrations from both local and import sources. A similar pollution episode, but of shorter duration occurred in Spring 2014. At the time, the observed elevated PM was blamed on a Saharan dust plume, but which in fact was then shown to be

from long-range transport of $NH_4NO_3$ (Vieno et al., 2016). Although the 2014 episode was not sufficiently large to be captured in the monthly AGANet data, it reaffirms the substantial contribution of long-range transport into the UK of $NH_4NO_3$, with precursor gas emissions from outside of the UK presenting a major driver (Vieno et al., 2016).



A second, but smaller pollutant episode that was captured by the AGANet occurred in September 2014, with elevated concentrations of $SO_2$, $HNO_3$, $SO_4^{2-}$, $NO_3^-$ and $NH_4^+$ that came from the Icelandic Holuhraun volcanic eruptions (Twigg et al., 2016). The elevated $SO_2$ concentration in September 2014 led to a modest increase in annual concentrations in $SO_2$ for 2014 (0.58 µg m$^{-3}$, *cf* annual mean = 0.54 and 0.27 µg m$^{-3}$ for 2013 and 2015, respectively). For the other components ($HNO_3$,

particulate $SO_4^2$, $NO_3^-$ and $NH_4^+$), the spikes in concentrations were smaller than for $SO_2$ and did not noticeably elevate their annual mean concentrations for that year. These pollution events together illustrate very clearly how short pollutant episodes can have a major influence on the measured annual concentrations in the UK, and that changes in meteorological conditions, coupled with long-range transboundary import can have a large effect on the UK concentration field.

**3.5    Long-term trends at Eskdalemuir**

At the Eskdalemuir rural background site, EMEP filter pack data in TIN (sum of $HNO_3$ and $NO_3^-$) and TIA (sum of $NH_3$ and $NH_4^+$) are available going back to 1989 (Sect. 2.7.3). In Figure 10, the EMEP filter pack TIN and TIA time series (Apr-89 to Dec-00) is extended with AGANet ($HNO_3$ and $NO_3^-$) and NAMN ($NH_3$ and $NH_4^+$) DELTA data (Sep-99 to Dec-15), with an overalpping period of 14 months. The combined time series shows that the annual concentrations of TIN has halved in 26

years between 1990 to 2015, from 0.36 to 0.16 µg N m$^{-3}$, compared with a 3-fold reduction in $NO_x$ emissions (from 889 to 280 kt $NO_2$-N) (defra, 2017) over the same period. For TIA, the 52 % decrease between 1990 to 2015 (from 0.93 to 0.45 µg N m$^{-3}$) is larger than the corresponding 10 % reduction in $NH_3$ emissions (from 268 to 241 kt $NH_3$-N) (defra, 2017). Speciated $NH_3$ and $NH_4^+$ data from NAMN over the period 2000 – 2015 shows that the decrease in TIA is mainly driven by $NH_4^+$, which decreased by 59 % between 2000 (annual mean = 0.62 µg $NH_4^+$ m$^{-3}$) and 2015 (annual mean = 0.25 µg $NH_4^+$ m$^{-3}$), compared

with no change in $NH_3$ (annual mean 0.32 µg $NH_3$ m$^{-3}$ in 2000, unchanged in 2015). This is consistent with findings by Tang et al. (2018) that contrary to the reported decrease in UK $NH_3$ emissions, $NH_3$ concentrations at background sites (defined by 5 km grid average $NH_3$ emissions <1 kg N ha$^{-1}$ y$^{-1}$) are showing an indicative increasing trend, while at the same time, a large downward trend in particulate $NH_4^+$ is observed. Together, the AGAnet and NAMN are thus providing an important long-term dataset that distinguishes between the gas and aerosol phase, allowing gas:aerosol phase interactions to be explored.

**<INSERT Figure 10 HERE>**

An extended time series illustrating the continued decline in $SO_2$ and $SO_4^{2-}$ has also been constructed by combining historic $SO_2$ and $SO_4^{2-}$ measurement data at the Eskdalemuir site going back to December 1977 (see Sect. 2.7.4) with AGANet $SO_2$

and $SO_4^{2-}$ data (Sep-99 to Dec-15) (Figure 11). A substantial decline in $SO_2$ is observed, falling by 98 % from 4.5 µg S m$^{-3}$ in 1978 to 0.07 µg S m$^{-3}$ in 2015, in good agreement with similarly large reduction in UK $SO_2$ emissions over the same period of 95 % (from 2530 to 120 kt $SO_2$-S). The decrease in $SO_4^{2-}$ is of a smaller magnitude, declining by 88 % from an annual mean



concentration of 0.89 µg S m$^{-3}$ in 1978 to 0.11 µg S m$^{-3}$ in 2015, highlighting the non-linearity in relationship between the atmospheric gas and aerosol phase of sulphur at this background site.

**<INSERT Figure 11 HERE>**

**3.6    Assessment of trends in relation to UK emissions**

The long-term time series in annually averaged concentrations of the gas and aerosol components are shown in Figure 12a and Figure 12b, respectively. Annually averaged data from the original 12 sites for the period 2000 – 2015 (1999 data excluded since AGANet started in September 1999) and from the full network (30 sites) for the period 2006 – 2015 are plotted alongside each other for comparison. From 2006 – 2015, the decreasing trends for all gas and aerosol components from the expanded 30

sites are seen to be similar to those from the original 12 sites. The annual mean concentrations in gas and aerosol components derived from the expanded 30 sites (2006 – 2015) or from the original 12 sites over the same period are also in general comparable (Table 4). The exceptions are Na$^+$ and Cl$^-$ that have higher mean concentrations from the 30 sites than the original 12 sites, due to the addition of two coastal sites (Shetland and Rum), with larger contribution from sea salt. Larger HNO$_3$ concentrations are due to two urban sites, London and Edinburgh (higher NO$_x$ emissions from vehicular traffic). The addition

of three sites in high NH$_3$ emission (agricultural) areas (Rosemaund in England, Narberth in Wales and Hillsborough in Northern Ireland) also elevated measured annual mean NH$_3$ concentrations. The comparisons here thus illustrates very clearly the need to consider the effect of site changes in a national network and the importance of maintaining consistency and site continuity for assessing long-term trends.

**<INSERT Figure 12 HERE>**

**<INSERT Table 4 HERE>**

Since there was a change in the number of sites during the operation of the AGANet, statistical trend analyses for HNO$_3$, SO$_2$, HCl and particulate NO$_3^-$, SO$_4^{2-}$, Cl$^-$ were performed on annually averaged mean concentrations from two time series: i) the

original 12 AGANet sites for the 16 year period from 2000 to 2015, and ii) the expanded 30 AGANet sites for the 10 year period from 2006 to 2015. NH$_3$ and NH$_4^+$ concentrations from the NAMN that were measured at the same time at the AGANet sites were also included for comparison and to aid interpretation of the acid gas and aerosol data. This approach avoids introducing bias as a result of changes in the sites and ensures site continuity for the long-term trend assessment.

Results of the Linear Regression (LR) and Mann-Kendall (MK) tests (Section 2.8) are summarised in Figure 13 and Table 5 for the original 12 sites (2000 – 2015), and in Figure 14 and Table 6 for the expanded 30 sites (2006 – 2015). To assess changes in measured concentrations over time, annual trends (e.g. µg HNO$_3^{-1}$ y$^{-1}$) were estimated from the regression results of the LR



and MK tests. This is considered to provide a more reliable estimate of trend than comparing measured annual concentrations at the beginning and end of the time series, which is subject to bias due to substantial variability in annual concentrations between years (Tang et al., 2018). The LR % annual trends for each time series are estimated from the LR's slope and intercept, while the MK % median annual trends are estimated from the MK Sen's slope and intercept (equation 5).

$$\% \text{ annual change } = 100 \cdot \frac{[Yi - Yo)}{Yo} \qquad [5]$$

where    $(y_0)$ and $(y_i)$ = estimated annual mean concentrations at the start and end of the selected time period, estimated from the slope and intercept of the LR or MK tests.

The long-term trends in the gas and aerosol components, based on statistical analysis of monthly mean measurement data, are

10    also shown for comparison in Figure S4 (mean monthly data of 12 sites for period 2000-2015) and Figure S5 (mean monthly data of 12 sites for period 2006-2015). Results of the trend analysis on monthly data (Tables S4, S5) were similar to trend analysis results of the annual data (Table 5; Table 6). While not discussed further here, since assessment of long-term trends in this paper focusses on trends in annual mean concentrations for comparison with trends in estimated annual emissions, the monthly plots serves to illustrate the large intra-annual variability of concentrations in gases and aerosols.

**<INSERT Figure 13 HERE>**
**<INSERT Table 5 HERE>**
**<INSERT Figure 14 HERE>**
**<INSERT Table 6 HERE>**

Measured concentrations of $HNO_3$ and particulate $NO_3^-$ from the AGANet show a clear significant decreasing trend, although less substantial than for $SO_2$ and $SO_4^{2-}$ (Figure 12). Between 2000 and 2015, $HNO_3$ decreased by 45 % (MK, LR = –42 %), compared with a 2-fold higher reduction of 81% (MK, LR = –84 %) in $SO_2$ concentrations (Table 5). The decrease in $HNO_3$ between 2006 and 2015 from the expanded 30 sites network (–36 % MK, –30 % LR) is similarly half that of $SO_2$ (–60 % MK

25    / LR) over the same period (Table 6). The decrease in $HNO_3$ is accompanied by a slightly larger decrease in particulate $NO_3^-$ (2000-2015: MK = –52 %, LR = –51 %, 2006–2015: MK = –43 %, LR = –39 %) (Table 7). By contrast, the decline in particulate $SO_4^{2-}$ (2000-2015: MK = –69 %, LR = –70 %, 2006–2015: MK = –54 %, LR = –53 %) is smaller than its precursor $SO_2$ and larger than the decrease in $NO_3^-$ (Table 7).

The formation of both $NO_3^-$ and $SO_4^{2-}$ requires $NH_4^+$ as a counter-ion, which decreased by 62 % (MK, –64 % LR) over the

30    period 2000 - 2015 (Table 5) and by 49 % (MK, –48 % LR) between 2006 and 2015 (Table 6). The decrease in $NH_4^+$ is 2-fold higher than the decline in its precursor $NH_3$ gas (2000-2015 = –30 % MK / LR, 2006-2015 = –18 % MK / –21% LR), and intermediate between that of $NO_3^-$ and $SO_4^{2-}$ (Table 7). For HCl and Cl-, there is a decreasing trend in HCl (2000-2015: MK



= –28 %, LR = –26 %, 2006–2015: MK = –24 %, LR = –21 %), but overall there is no detectable trend in particulate Cl⁻ (Table 5, Table 6).

**<INSERT Table 7 HERE>**

**<INSERT Figure 15 HERE>**

Large decreasing trends in estimated emissions for $NO_x$, and in particular for $SO_2$ continued since monitoring in AGANet began in 1999. $NO_x$ emissions decreased by 58 % between 2000 and 2015, followed closely by a similar reduction in measured $HNO_3$ (–45 % MK / –42 % LR) and $NO_3^-$ concentrations (–52 % MK / –51 % LR) over the same period (Figure 15a; Table 7). Results from the trend analysis of 30 AGANet sites between 2006 and 2015 ($HNO_3$ = –36 % MK / -30 % LR, $NO_3^-$ = –43

% MK / –39 % LR) are also consistent with the 41 % decline in $NO_x$ emissions over this period, in agreement with the emissions inventory (Figure 15b; Table 7). $HNO_3$, as a secondary product of $NO_x$, provides an important measure of the fraction of $NO_x$ emissions that is oxidised within the country and signals any long-term changes in the atmospheric processing timescales of $NO_x$ over the country. Since the decreasing trends in estimated $NO_x$ emissions and measured $HNO_3$ are similar, there appears to have been no change in the oxidation and processing of $NO_x$ in the atmosphere.

**<INSERT Figure 16 HERE>**

For $SO_2$, there has been a more significant decline, both in emissions and measured concentrations during this period (Figure 16). The network annual mean concentration decreased from 1.9 µg $SO_2$ m⁻³ in 2000 to 0.25 µg $SO_2$ m⁻³ in 2015 (mean of 12

sites), continuing the long-term decline in $SO_2$ concentrations observed at the background Eskdalemuir site (Sect. 3.5) and across the UK (ROTAP 2012). From the trend analysis, the decrease in annual mean $SO_2$ concentrations of –81 % (MK, LR = –84 %) between 2000–2015 (Table 5), and –60 % (MK and LR) between 2006 – 2015 (Table 6) are consistent with the substantial reduction of –80 % and –64 % in $SO_2$ emissions over the two overlapping periods, respectively (Figure 16; Table 7). At the same time, the reduction in $SO_2$ emission and measured concentration is accompanied by a smaller negative trend

in particulate $SO_4^{2-}$ (2000-2015: –69 % MK/ -70 % LR; 2006-2015: –54 % MK / –53 % LR) (Table 5; Table 7), with concentrations falling 3-fold from an annual mean of 1.2 µg $SO_4^{2-}$ m⁻³ in 2000 to 0.42 µg $SO_4^{2-}$ m⁻³ in 2015. The smaller decrease in particulate $SO_4^{2-}$ compared with its gaseous precursor, $SO_2$, is similar to that observed at Eskdalemuir (Sect. 3.5).

**<INSERT Figure 17 HERE>**

By contrast, a smaller 30 % decrease is seen in the annually averaged $NH_3$ concentrations at the 12 AGANet sites (2000-2015: –30 % MK/LR) (Figure 13; Table 5). This decrease is larger than the decrease in $NH_3$ emissions of 10 % over the same period (Table 7). A recent assessment by Tang et al. (2018) showed that $NH_3$ trends are highly dependent on site selection and a more





comprehensive analysis of a larger number of sites shows smaller reductions over time. By contrast, a significant decreasing trend in NH$_3$ concentrations was observed in the grouped analysis of sites in areas classed as dominated by pig and poultry emissions, against an upward (non-significant) trend for sites in cattle-dominated areas. Therefore there is a large degree of uncertainty in interpreting the trends in NH$_3$ concentrations from a subset of just 12 sites. What the NH$_3$ data from the 12 sites

does show, however, is that there is a shift from air being dominated by SO$_2$ towards it being dominated by NH$_3$, with a rapidly changing chemical climate in terms of relative concentrations of NH$_3$ and SO$_2$. The ratio of annual mean concentrations of NH$_3$ (80 nmol m$^{-3}$) to SO$_2$ (29 nmol m$^{-3}$) was 2.7 in 2000 (Figure 18). By 2015, this ratio had increased to 15 (annual mean concentrations of NH$_3$ = 58 nmol m$^{-3}$ and SO$_2$ = 4 nmol m$^{-3}$) (Figure 18). At the same time, there is a larger decrease in NH$_4^+$ concentrations (–62 % MK / –64 % LR), contrasting with a smaller decrease in NH$_3$ concentrations over the period 2010–2015

(–30 % MK / LR) (Table 7), with the NH$_3$:NH$_4^+$ ratio increasing with time (Figure 18). This provides evidence for a shift in partitioning from the particulate phase NH$_4^+$ to the gaseous phase NH$_3$ in the UK data, discussed in Tang et al. (2018). The change in partitioning from particulate NH$_4^+$ to gaseous NH$_3$ is also occurring in other parts of Europe, where decreases in NH$_3$ concentrations have been smaller than emission trends would suggest, due to large decreases in SO$_2$ emissions (Bleeker et al., 2009; Horvath et al., 2009).

Studies have shown that the increasing ratio of NH$_3$ to SO$_2$ in the atmosphere leads to increased dry deposition of SO$_2$, accelerating the decrease in atmospheric SO$_2$ concentrations than would be achieved by emissions reduction alone (Fowler et al., 2001; ROTAP 2012). The dry deposition of SO$_2$ and NH$_3$, by uptake of the gases in a liquid film on leave surfaces, are known to be enhanced when both gases are present in a process termed "co-deposition" (Fowler et al., 2001). Where ambient NH$_3$ concentrations exceed that of SO$_2$, there is enough NH$_3$ to neutralize acidity in the liquid film and oxidise deposited SO$_2$,

and maintain large rates of deposition of SO$_2$. Supporting evidence for enhanced deposition of SO$_2$ in a NH$_3$-rich atmosphere is provided by the AGANet data that shows a more rapid decline in SO$_2$ concentrations than SO$_4^{2-}$, and the negative trend in the ratio of SO$_2$:SO$_4^{2-}$.

**<INSERT Figure 18 HERE>**

The substantial decrease in UK SO$_2$ emissions and concentrations, while UK NO$_x$ emissions and concentrations remain relatively high in comparison, set against a much smaller decrease in NH$_3$ emissions and concentrations since 2000 is leading to changes in the respective particulate SO$_4^{2-}$, NO$_3^-$ and NH$_4^+$ concentrations. Since the affinity of H$_2$SO$_4$ (oxidation product of SO$_2$) for NH$_3$ is much larger than that of HNO$_3$ and HCl, available NH$_3$ is first taken up by H$_2$SO$_4$ to form ammonium sulphate compounds (NH$_4$HSO$_4$ and (NH$_4$)$_2$SO$_4$), with any excess NH$_3$ then available to react with HNO$_3$ and HCl to

formNH$_4$NO$_3$ and NH$_4$Cl. The increase in ratio of HNO$_3$:NO$_3^-$ is similar to changes in upward trend in gas-aerosol partitioning between NH$_3$ and NH$_4^+$ over time (Figure 18). The negative trend in ratio of SO$_2$:SO$_4^{2-}$, contrasting with the positive trend in ratio of HNO$_3$:NO$_3^-$ (Figure 18) lends further support that reduction in SO$_2$ emissions is contributing to a more rapid decrease




in particulate $(NH_4)_2SO_4$ concentrations, as more $NH_3$ becomes available to react with $HNO_3$ to form the $NH_4NO_3$, A change to an $NH_4NO_3$ rich atmosphere may account for the smaller negative trend in particulate $NO_3^-$ observed here.

Currently, the critical loads of acidity (sulphur and nitrogen) are exceeded by 44 % of the area of sensitive habitats in the UK (based on mean deposition data for 2012-2014), whereas the figure for exceedance of eutrophication (nutrient nitrogen) is even larger, at 62 % (based on deposition data for 2012 – 2014) (Hall & Smith, 2016). Air quality policies have been very successful in abating $SO_2$ emissions (–80 % between 2000 – 2015) and moderately successful with $NO_x$ emissions (–58 % between 2000 – 2015), with both on course to meet the emission reduction targets set out under the 2012 Gothenburg protocol and 2016 NECD. There remains however little political will to reduce emissions of $NH_3$. Since estimated emissions of $NH_3$ decreased by only 10 % over the same period, it is likely that abatement measures may be required to meet emission reduction targets. At the same time, results from the AGANet show a change in the particulate phase from $(NH_4)_2SO_4$ to $NH_4NO_3$. This change is expected to increase residence times of $NH_3$ and $HNO_3$ in the atmosphere, as the semi-volatile $NH_4NO_3$ will volatilise in warm weather to release $HNO_3$ and $NH_3$. A higher concentration of the gas-phase nitrogen species ($HNO_3$ and $NH_3$) may therefore be maintained in the atmosphere than expected on the basis of the emissions trends in $NO_x$ and $NH_3$. More of the $NH_3$ and $NO_x$ emitted will deposit more locally with a smaller footprint within the UK. Based on the current emission trends and evidence from AGANet and NAMN long-term measurements, atmospheric N deposition from oxidised N ($NO_x$, $HNO_3$ and $NO_3^-$) and from reduced N ($NH_3$, $NH_4^+$) are likely to continue to exceed critical loads of N deposition over large areas of sensitive habitats, with implications for UK's commitment to maintain or restore natural habitats (e.g. Natura 2000 sites; Hallsworth et al., 2010) to a favourable conservation status under the EU Habitats Directive (Council Directive 92/43/EEC). The changes are also relevant for human health effects assessments, since $NH_4NO_3$ and $(NH_4)_2SO_4$ are mainly in the fine mode and constitute a significant fraction of $PM_{2.5}$ that are associated with acute and chronic human health problems. The change in partitioning from $(NH_4)_2SO_4$ to $NH_4NO_3$, coupled to import of $NH_4NO_3$ from long-range transport (driven by emissions of $NH_3$ and $NO_x$ from outside the UK) poses policy challenges in protection of human health from effects of air pollution, particularly in urban areas where concentrations of the $PM_{2.5}$ precursor gases $NO_x$, $SO_2$ and $NH_3$ are higher.

## 4    Conclusions

The UK Acid Gases and Aerosol network (AGANet) is delivering, uniquely, a comprehensive UK long-term dataset of speciated acid gases ($HNO_3$, $SO_2$, $HCl$) and aerosol components ($NO_3^-$, $SO_4^{2-}$, $Cl^-$, $Na^+$, $Ca^{2+}$, $Mg^{2+}$) and also of $NH_3$ and $NH_4^+$ measured within the National Ammonia Monitoring Network (NAMN). Speciated measurements are made with an established low-cost DELTA denuder-filter pack methodology, allowing assessment of atmospheric chemical composition and gas-aerosol phase interactions. Other manual denuder-filter implementations designed for high time-resolution measurements are useful at selected locations for detailed analysis and model testing, but they are resource intensive and expensive. The DELTA



monthly measurements on the other hand are cost-efficient for estimating annual mean concentrations, providing sufficient resolution for analysis of temporal trends and which can be operated at a large number of sites in the network to provide long-term trends and temporal/spatial patterns.

Large regional patterns in concentrations are observed, with the largest concentrations of $HNO_3$, $SO_2$, and aerosol $NO_3^-$ and $SO_4^{2-}$ in south and east England, attributed to anthropogenic (combustion, vehicular) and long-range transboundary sources from Europe, and smallest in western Scotland and Northern Ireland. HCl concentrations are also largest in the southeast and southwest of England, attributed to dual contribution from anthropogenic (coal combustion) and marine sources (reaction of sea salt with $HNO_3$ and $H_2SO_4$ to form HCl) from coal combustion. For $Cl^-$, this has a similar spatial distribution as $Na^+$, with
highest concentrations of at coastal sites, reflecting their origin from marine sources (sea salt).

Distinctive temporal trends are established for the different components, with the seasonal variability influenced by local to regional emissions, climate, meteorology and photochemistry. $HNO_3$ and $NO_3^-$ have a maximum during late spring and early summer, due to photochemical production processes and from long-range transboundary pollutant transport. $SO_4^{2-}$
concentrations also have a peak in spring-time, coinciding with the peak in concentrations of $NH_3$ and $NH_4^+$, and is therefore likely to be attributed to formation of $(NH_4)_2SO_4$ from reaction with a surplus of higher concentrations of $NH_3$ at that time of year. Conversely, peak concentrations of $SO_2$, $Na^+$ and $Cl^-$ occur during winter, likely from combustion processes (heating) for $SO_2$ and marine sources in winter (more stormy weather) for sea salt generation. Magnesium and calcium are both crustal elements, but which are also present in sea salt aerosols. The seasonal trend in $Mg^{2+}$ is similar to $Na^+$, with maxima during
winter and minima in summer; therefore some of the sea salt aerosol may be in the form of $MgCl_2$. For $Ca^{2+}$, the winter maxima is much less pronounced and its seasonal variability is likely to be influenced by both crustal dust and sea salt.

Enhancement of local to regional concentrations of reactive gases and aerosols in the UK from long-range transboundary transport of pollutants into the UK is highlighted by two pollution events, captured in the long-term AGANet monthly
measurements. In 2003, a spring episode with elevated concentrations of $HNO_3$ and $NO_3^-$ was driven by meteorology, with easterly winds transporting $NH_4NO_3$ formed in Europe into the UK and a high pressure system over the UK (Feb-April) that led to a build-up of $NH_4NO_3$ and $HNO_3$ concentrations from both local and transboundary sources. A second, but smaller episode of elevated concentrations of $SO_2$ and $HNO_3$, as well as of particulate $SO_4^2$, $NO_3^-$ and $NH_4^+$, in September 2014 was shown to be from transport of pollutant plume from the Icelandic Holuhraun volcanic eruptions at that time.

After more than 16 years of operation, the AGANet is also capturing important long-term changes in the concentrations and partitioning between gas and aerosol of the N and S components in the atmosphere. For $SO_2$, a significant decreasing trend in annual mean $SO_2$ concentrations of –81 % (MK, LR = –84 %) from 2000–2015 was in agreement with the estimated –80 % $SO_2$ emissions reductions, but larger than the accompanying decline in particulate $SO_4^{2-}$ (–69 % MK/ -70 % LR). A more



modest reduction in HNO$_3$ (–45 % MK / –42 % LR) and particulate NO$_3^-$ (–52 % MK / –51 % LR) are consistent with the estimated 58 % decline in NO$_x$ emissions over this same period. At the same time, NH$_3$ measurements made under NAMN shows a shift from air being dominated by SO$_2$ towards it being dominated by NH$_3$. For particulate NH$_4^+$, the decrease in concentrations (–62 % MK / –64 % LR) is larger than the precursor gas NH$_3$ (2000 – 2015 = –30 % MK / LR), and larger than the estimated decline in estimated NH$_3$ emissions of 10 %. However, it should be noted that NH$_3$ trends are highly dependent on site selection and a more comprehensive analysis of a larger number of sites shows smaller reductions over time.

The substantial decrease in UK SO$_2$ emissions and concentrations, while UK NO$_x$ emissions and concentrations (HNO$_3$) remain relatively high in comparison, set against a much smaller decrease in NH$_3$ emissions and concentrations since 2000 is leaving more NH$_3$ available to react with HNO$_3$ to form the semi-volatile particulate NH$_4$NO$_3$. The change in partitioning from particulate NH$_4^+$ to gaseous NH$_3$ is also occurring in other parts of Europe, where decreases in NH$_3$ concentrations have been smaller then emission trends would suggest, due to successful mitigation in SO$_2$ emissions. At the same time, a change in the particulate phase from (NH$_4$)$_2$SO$_4$ to NH$_4$NO$_3$ would contribute to slowing down the negative trend in particulate NO$_3^-$ observed here. Higher concentrations of the NH$_3$ and HNO$_3$ in the atmosphere will deposit more locally, exacerbating the effects of local N deposition loads over large areas of sensitive habitats, with implications for UK's commitment to maintain or restore natural habitats (e.g. Natura 2000 sites) to a favourable conservation status under the EU Habitats Directive (Council Directive 92/43/EEC). The changes are also important in terms of human effects assessment since NH$_4$NO$_3$ constitute a significant fraction of PM$_{2.5}$ that are implicated in acute and chronic human health effects and linked to increased mortality from respiratory and cardiopulmonary diseases.

**Acknowledgements**

The UK AGANet and NAMN are funded by the Department for Environment, Food and Rural Affairs (Defra) and the devolved administrations, under the UK Acid Deposition Monitoring Network (ADMN: 1999 - 2008), UK Eutrophying and Acidifying Atmospheric Pollutants network (UKEAP: 2009 - present) and from supporting NERC CEH programmes. The authors gratefully acknowledge assistance and contributions from the following: the large numbers of dedicated local site operators without whom the monitoring work would not be possible, site owners for provision of facilities, Harwell Scientifics Laboratory (now Environmental Scientifics Group (ESG) Ltd) for provision of chemical analysis for the AGANet between 1999 to 2009, the Centralised Analytical Chemistry facility in Lancaster (in particular Heather Carter, Darren Sleep and Philip Rowland) for sample preparation and chemical analysis since 2009, and colleagues at both CEH Edinburgh (Robert Storeton-West, Sarah Leeson, Matt Jones, Chris Andrews, Margaret Anderson, Ian D. Leith) and Ricardo Energy & Environment field team (Martin Davies, Tim Bevington, Ben Davies, Chris Colbeck) for assisting in site / equipment maintenance and data collection.





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




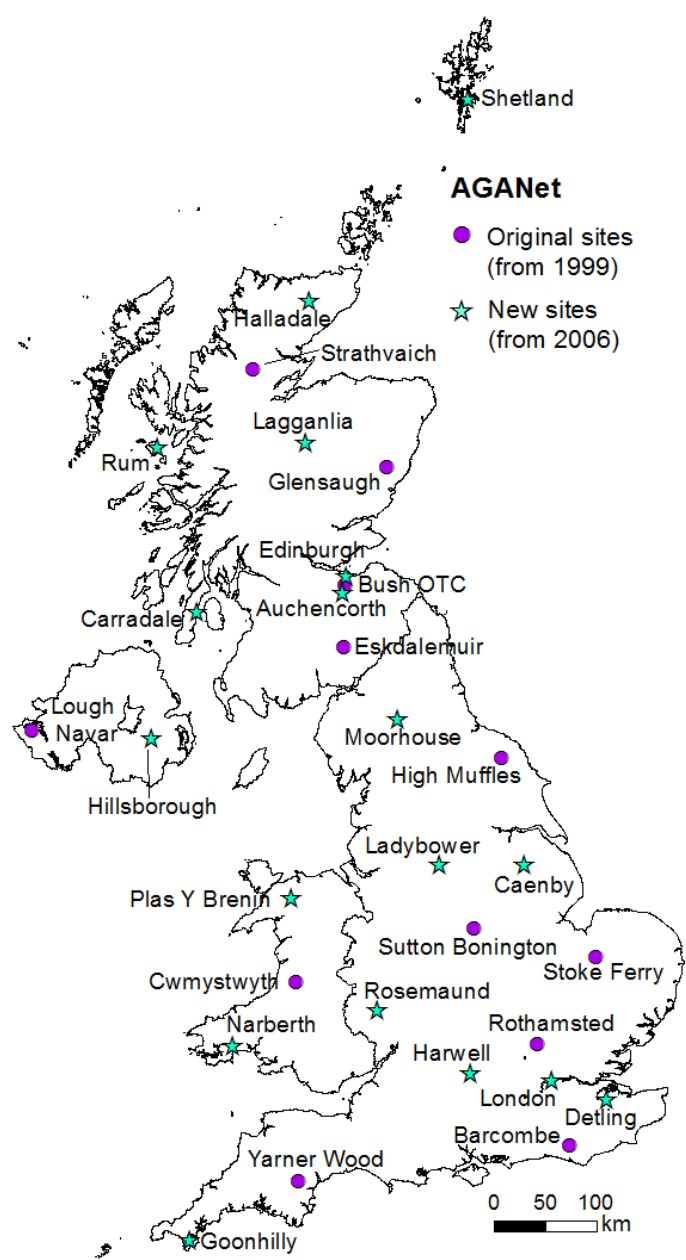

**Figure 1:** Site map of the UK Acid Gases and Aerosol Network (AGANet). The AGANet was established in September 1999 with 12 sites and expanded to 30 sites from January 2006 to improve national coverage. These sites also provide measurements of $NH_3$ and $NH_4^+$ for the UK National Ammonia Monitoring Network (NAMN, Tang et al., 2018).





(a) Gases (HNO$_3$, SO$_2$, HCl, NH$_3$)

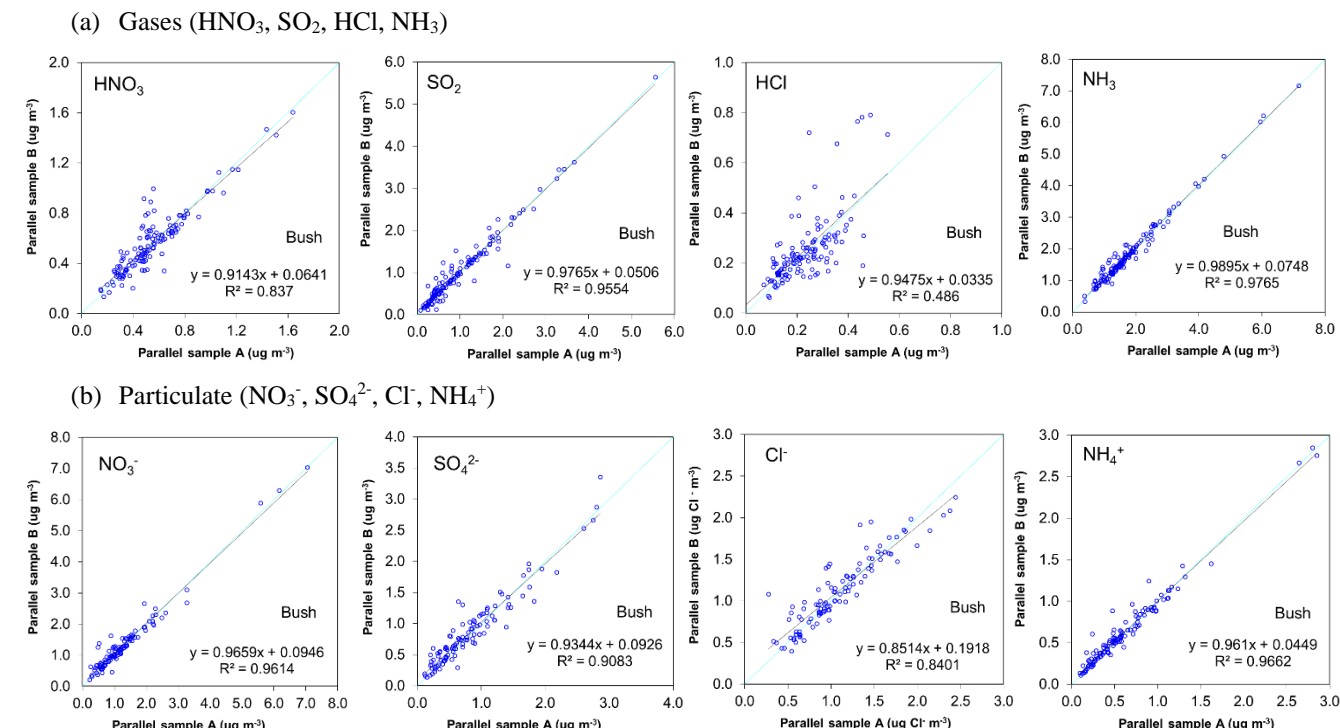

5      (b) Particulate (NO$_3^-$, SO$_4^{2-}$, Cl$^-$, NH$_4^+$)

(c) Summary of regression analysis

|  | Gases | | | | Particulates | | | |
|---|---|---|---|---|---|---|---|---|
|  | HNO$_3$ | SO$_2$ | HCl | NH$_3$ | NO$_3^-$ | SO$_4^{2-}$ | Cl$^-$ | NH$_4^+$ |
| $R^2$ | 0.837*** | 0.955*** | 0.486*** | 0.976*** | 0.961*** | 0.908*** | 0.840*** | 0.966*** |
| slope | 0.914* | 0.976$^{ns}$ | 0.947$^{ns}$ | 0.989$^{ns}$ | 0.966 $^{ns}$ | 0.934$^{*}$ | 0.851*** | 0.961* |
| intercept | 0.064** | 0.051* | 0.033$^{ns}$ | 0.075** | 0.095** | 0.093** | 0.192*** | 0.045*** |
| No. observations ($n$) | 130 | 130 | 128 | 140 | 108 | 108 | 104 | 119 |
| mean A (µg m$^{-3}$) | 0.54 | 1.02 | 0.23 | 1.75 | 1.29 | 0.84 | 1.09 | 0.61 |
| mean B (µg m$^{-3}$) | 0.56 | 1.05 | 0.25 | 1.80 | 1.34 | 0.87 | 1.12 | 0.63 |

Significance level (slope different from 1, intercept = 0): * $p < 0.05$, ** $p < 0.01$, *** $p < 0.001$. $ns$ = not significant ($p > 0.05$)

10   **Figure 2:** Comparisons of parallel measurement of monthly (a) atmospheric reactive gases (HNO$_3$, SO$_2$, HCl and NH$_3$) and (b) particulate (NO$_3^-$, SO$_4^{2-}$, Cl$^-$ and NH$_4^+$) concentrations from duplicate DELTA sampling at the UK Acid Gas and Aerosol Monitoring Network (AGANet) and National Ammonia Monitoring Network (NAMN) site Bush OTC (UKA00128) in Southern Scotland for the period 1999 to2015. (c) A summary of the regression analyses. Each point represents a comparison between the paired monthly DELTA measurements.





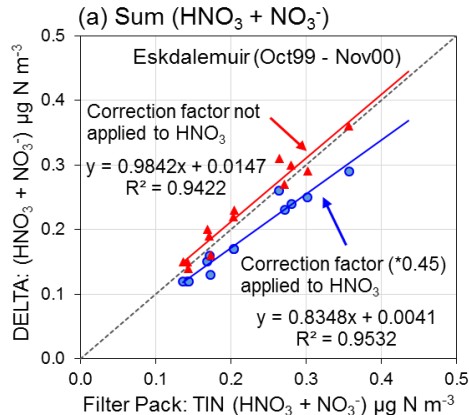
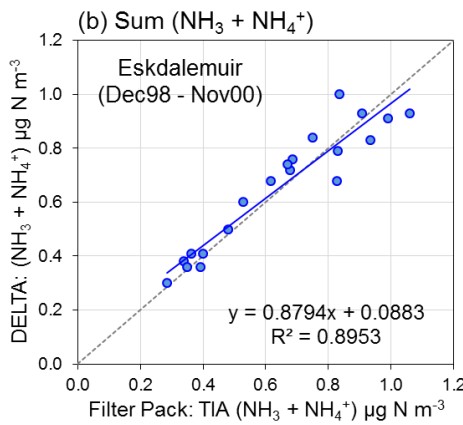

**Figure 3:** Comparison of (a) total inorganic nitrate, TIN (sum of $HNO_3$ + $NO_3^-$) and (b) total inorganic ammonium, TIA (sum of $NH_3$ + $NH_4^+$) concentrations at the Eskdalemuir monitoring station (EMEP station code = GB0002R; UK-AIR ID = UKA00130) measured under the EMEP program with concentrations of the corresponding gas and aerosol from the UK Acid Gases and Aerosol (AGANet, $HNO_3$ and $NO_3^-$) and UK National Ammonia Monitoring Network (NAMN, $NH_3$ and $NH_4^+$). EMEP values (data downloaded from http://ebas.nilu.no/) are means of daily measurements for TIN and TIA by the EMEP filter pack method, matched to the AGANet and NAMN sampling periods (monthly). Filter pack measurements at Eskdalemuir terminated in December 2000.





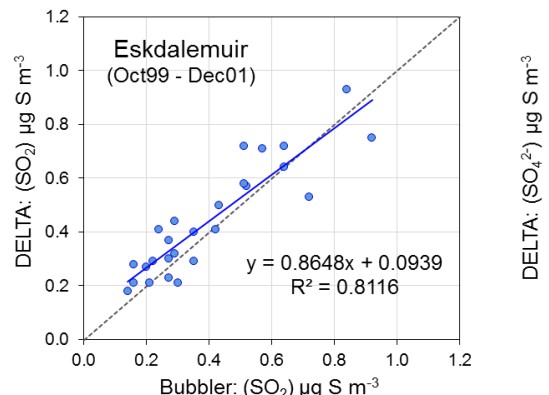
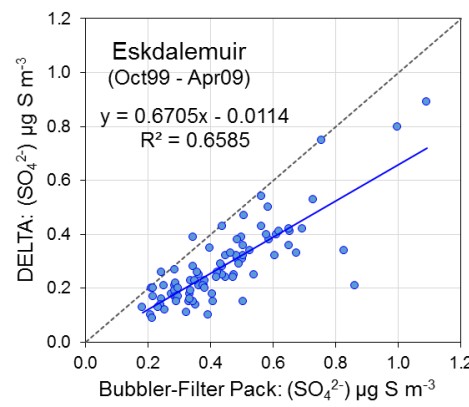

|  | AGANet DELTA: $SO_2$ | Bubbler: $SO_2$ | AGANet DELTA: $SO_4^{2-}$ | Filter Pack: $SO_4^{2-}$ |
|---|---|---|---|---|
| Linear regression: $R^2$ | 0.812*** |  | 0.658*** |  |
| slope | 0.865$^{ns}$ |  | 0.670*** |  |
| intercept | 0.094* |  | 0.011$^{ns}$ |  |
| mean (µg S m$^{-3}$) | 0.44 | 0.40 | 0.28 | 0.44 |
| No. of observations ($n$) | 26 | 26 | 87 | 87 |

Significance level (slope different from 1, intercept = 0): * $p < 0.05$, ** $p < 0.01$, *** $p < 0.001$. $ns$ = not significant ($p > 0.05$)

**Figure 4:** Comparison of gaseous $SO_2$ and particulate $SO_4^{2-}$ concentrations at the Eskdalemuir monitoring station (EMEP station code = GB0002R; UK-AIR ID = UKA00130) measured under the Acid Deposition Monitoring Program (ADMN, Hayman et al., 2007) with the corresponding gas and aerosol from the UK Acid Gases and Aerosol network (AGANet). ADMN values (data downloaded from http://ebas.nilu.no/) are means of daily measurements for $SO_2$ by the bubbler method and $SO_4^{2-}$ by the EMEP filter pack method (Hayman et al., 2007), matched to the AGANet sampling periods (monthly). Bubbler and filter pack measurements at Eskdalemuir terminated in December 2001 and Apr 2009, respectively.



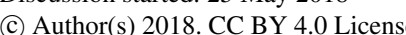

**Figure 5:** Annual mean monitored acid gas ($HNO_3$, $SO_2$, HCl) and aerosol ($NO_3^-$, $SO_4^{2-}$, $Cl^-$, $Na^+$, $Ca^{2+}$, $Mg^{2+}$) concentrations from the UK Acid Gas and Aerosol Monitoring Network (AGANet) across the UK from annual averaged monthly measurements made in 2013. $NH_3$ and $NH_4^+$ measured at the same time from the UK National Ammonia Monitoring Network (NAMN, Tang et al., 2018) are also shown alongside for comparison.



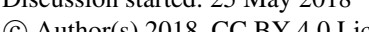



**(a) Gaseous components**

**(b) Particulate components**

**Figure 6:** Scatter plots between concentrations of (a) gaseous species $HNO_3$, $SO_2$ and $NH_3$, and (b) particulate species $NO_3^-$, $SO_4^{2-}$, $NH_4^+$, $Cl^-$ and $Na^+$ from mean monthly measurements (1999-2015) from the 12 sites in the UK Acid Gas and Aerosol Monitoring Network (AGANet) that were operational over the whole period. $NH_3$ and $NH_4^+$ data are from the UK National Ammonia Monitoring Network (NAMN, Tang et al., 2018) made at the same time. Each data point represents a single monthly DELTA measurement.





**Figure 7:** Average annual cycles for $HNO_3$, $SO_2$, HCl and aerosol $NO_3^-$, $SO_4^{2-}$, $Cl^-$, $Na^+$, $Ca^{2+}$ and $Mg^{2+}$ from the UK Acid Gases and Aerosol Monitoring Network (AGANet). The $NH_3$ and $NH_4^+$ concentrations measured at the same time in the UK National Ammonia Monitoring Network (NAMN, Tang et al., 2018) are also shown for comparison. Each data point in the graphs represents the mean ± SD of monthly measurements of all sites in the network.





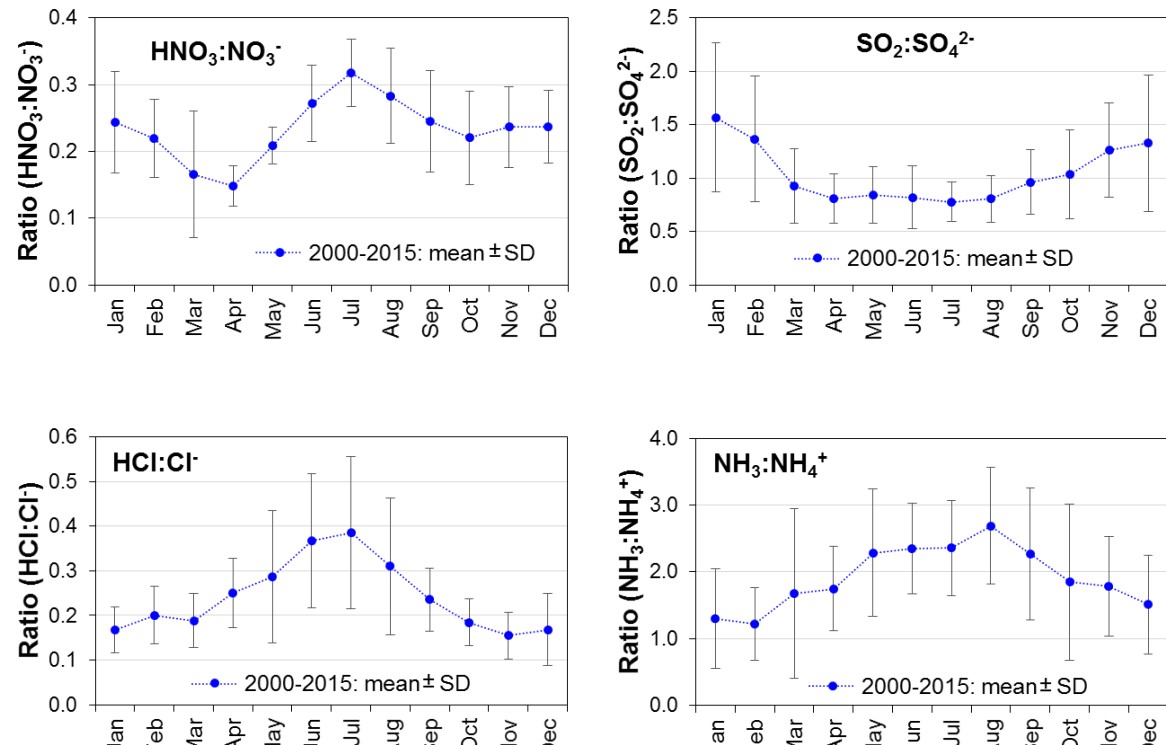

**Figure 8:** Average annual cycles in the ratios of gas:aerosol component concentrations (µg m⁻³). HNO₃, SO₂, HCl and aerosol NO₃⁻, SO₄²⁻, Cl⁻ data are from the UK Acid Gases and Aerosol Monitoring Network (AGANet). NH₃ and NH₄⁺ concentrations (µg m⁻³) that are measured at the same time for the UK National Ammonia Monitoring Network (NAMN, Tang et al., 2018) are also shown for comparison. Each data point in the graphs represents the mean ± SD of monthly measurements of 12 sites operational in the network over the period 2000 to 2015.



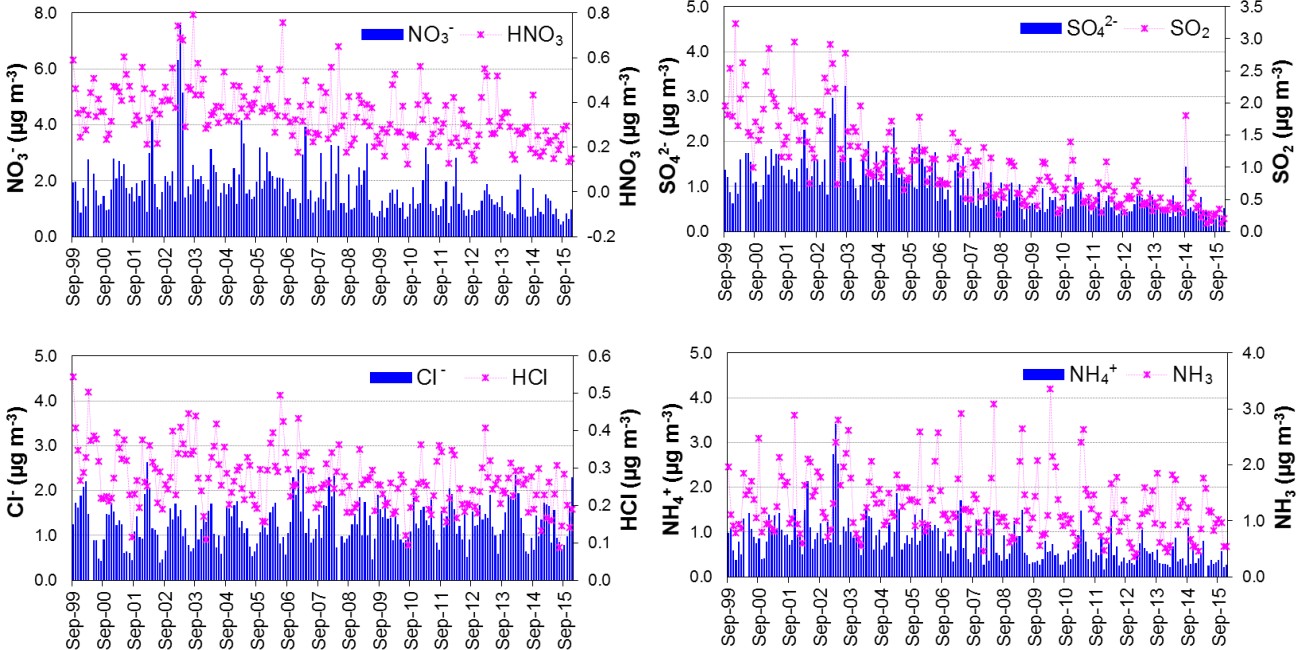

5  **Figure 9:** Monthly mean concentrations in gaseous HNO$_3$, SO$_2$, HCl and aerosol NO$_3^-$, SO$_4^{2-}$, Cl$^-$ from the UK Acid Gases and Aerosol Monitoring Network (AGANet). Monthly mean concentrations of NH$_3$ and NH$_4^+$ that were measured at the same time in the UK National Ammonia Monitoring Network (NAMN, Tang et al., 2018) are also shown for comparison. Each data point in the graphs represents the mean of monthly measurements of 12 sites operational in the network over the period Sep 1999 to December 2015. The same plots for the full 30 site network from 2006 - 2015 are shown in Supp. Figure S6.





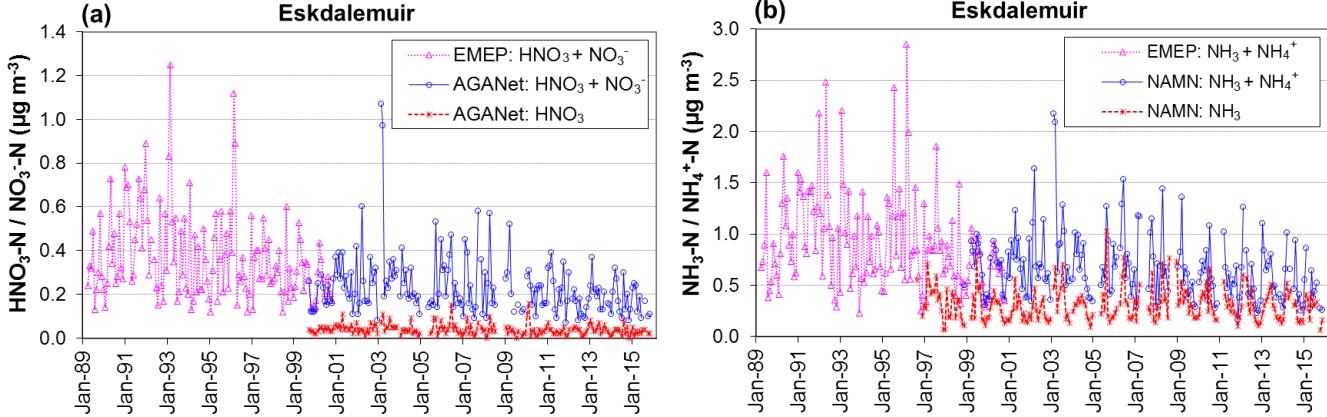

**Figure 10:** Long-term time series of (a) oxidised nitrogen ($HNO_3$ and $NO_3^-$) and (b) reduced nitrogen ($NH_3$ and $NH_4^+$) concentrations at Eskdalemuir (EMEP station code = GB0002R; UK-AIR ID = UKA00130). EMEP values (data downloaded from http://ebas.nilu.no/) are monthly means of daily measurements for total inorganic nitrogen, TIN (sum of $HNO_3$ and $NO_3^-$) and total inorganic nitrogen, TIA (sum of $NH_3$ and $NH_4^+$) by the EMEP filter pack method (Apr-89 – Nov-00), matched to the AGANet and NAMN sampling periods (monthly) where the measurements overlap. The AGANet and NAMN data are for gaseous $HNO_3$ and $NH_3$ and for the sum of ($HNO_3$ + $NO_3^-$) and sum of ($NH_3$ + $NH_4^+$), respectively, by the DELTA method. The AGANet $HNO_3$ values shown here includes the bias correction (Section 2.6).



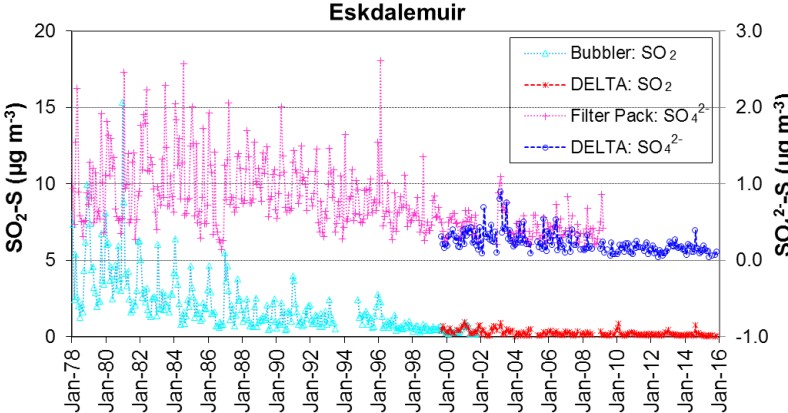

**Figure 11:** Long-term time series of $SO_2$ (Dec-77 – Jul-93) and $SO_4^{2-}$ (Dec-77 – Dec-01) concentrations measured in the UK Acid Deposition Monitoring Network (ADMN) (Hayman et al., 2007) and the AGANet DELTA measurements (Oct-99 – Dec-15) at the Eskdalemuir monitoring station (EMEP station code = GB0002R; UK-AIR ID = UKA00130). ADMN values (data downloaded from http://ebas.nilu.no/) are monthly means of daily measurements for $SO_2$ and $SO_4^{2-}$ by a daily bubbler and filter pack method, respectively, matched to the AGANet sampling periods (monthly) where the measurements overlap.





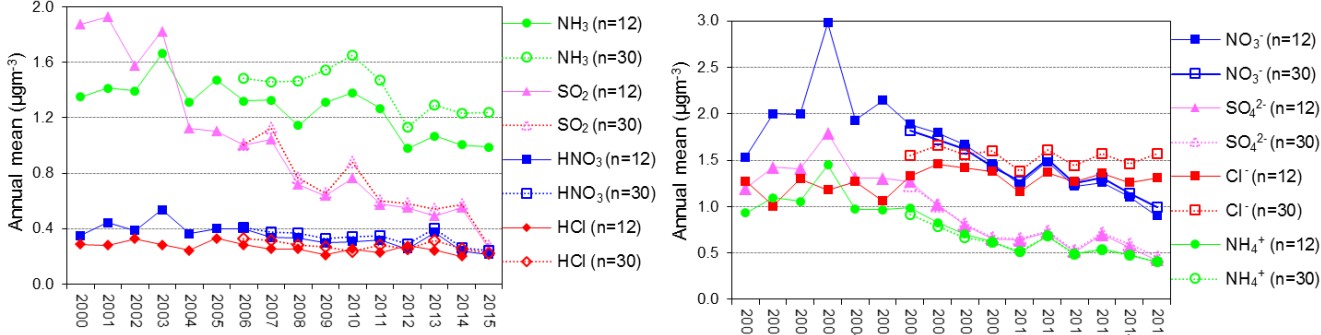

**Figure 12:** Long-term trends in (a) acid gases and (b) aerosol concentrations from the UK Acid Gases and Aerosol Network (AGANet). Each data point represents the annually averaged measurements from either the original 12 AGANet sites for the 16-year period from 2000 to 2015 or the expanded 30 AGANet sites for the 10-year period from 2006 to 2015. $NH_3$ and particulate $NH_4^+$ measured at the same time in the UK National Ammonia Monitoring Network (NAMN, Tang et al., 2018) are also included for comparison.





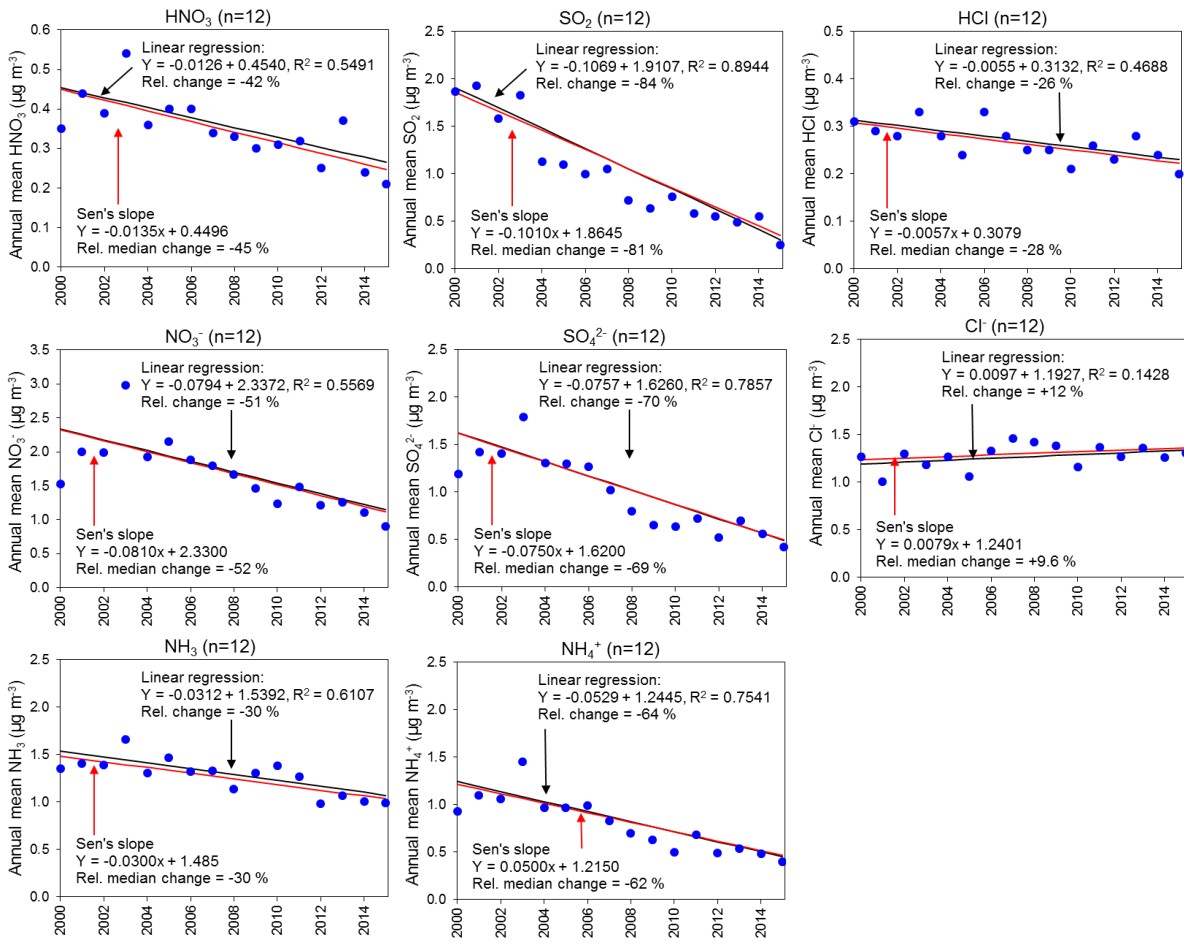

**Figure 13:** Time-series trend analysis by non-parametric Mann-Kendall Sen slope and by parametric linear regression on annually averaged gas and aerosol concentration data from the UK Acid Gases and Aerosol Monitoring Network (AGANet) of 12 sites that were operational over the period 2000 to 2015. $NH_3$ and $NH_4^+$ concentrations measured at the same time in the UK National Ammonia Monitoring Network (NAMN, Tang et al., 2018) are also included for comparison.





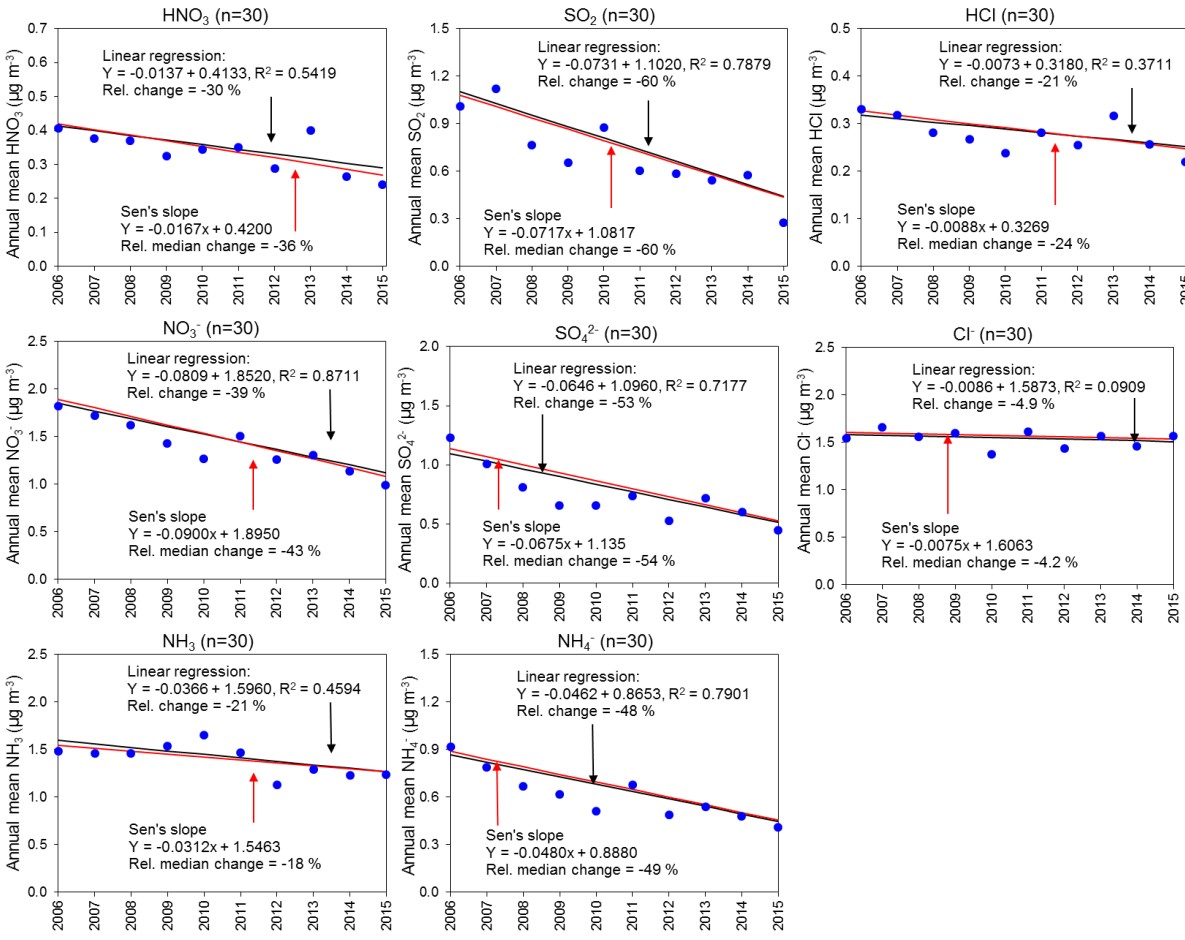

5  **Figure 14:** Time series trend analysis by non-parametric Mann-Kendall Sen slope and by parametric linear regression on annually averaged gas and aerosol concentration data from the UK Acid Gases and Aerosol Monitoring Network (AGANet) of 30 sites that were operational over the period 2006 to 2015. $NH_3$ and $NH_4^+$ concentrations data measured at the same time in the UK National Ammonia Monitoring Network (NAMN, Tang et al., 2018) are also included for comparison.





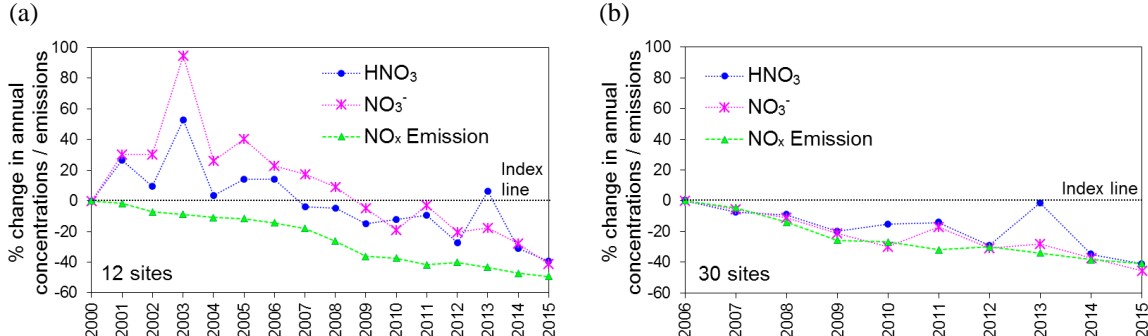

**Figure 15:** Relative trends in UK $NO_x$ emissions (Defra 2017) and in annually averaged $HNO_3$ and particulate $NO_3^-$ concentrations for (a) the original 12 AGANet sites for the 16 year period from 2000 to 2015, and (b) the expanded 30 AGANet sites for the 10 year period from 2006 to 2015.

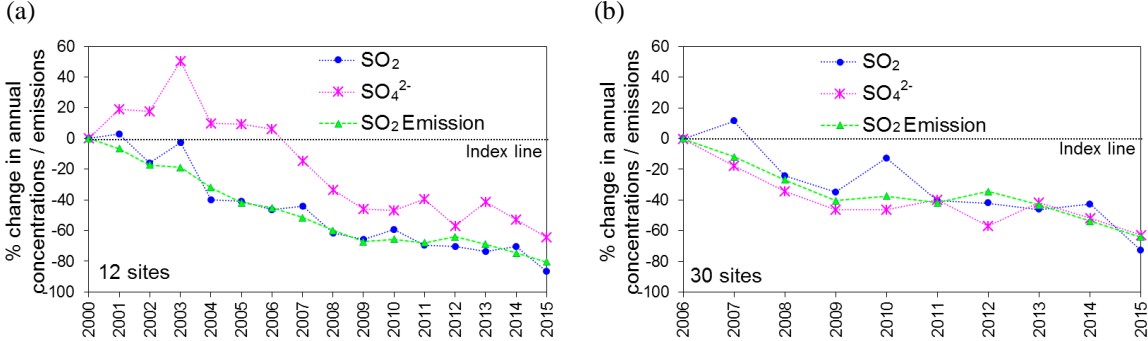

**Figure 16:** Relative trends in UK $SO_2$ emissions (Defra 2017) and in annually averaged $SO_2$ and particulate $SO_4^{2-}$ concentrations for (a) the original 12 AGANet sites for the 16 year period from 2000 to 2015 and (b) the expanded 30 AGANet sites for the 10 year period from 2006 to 2015.

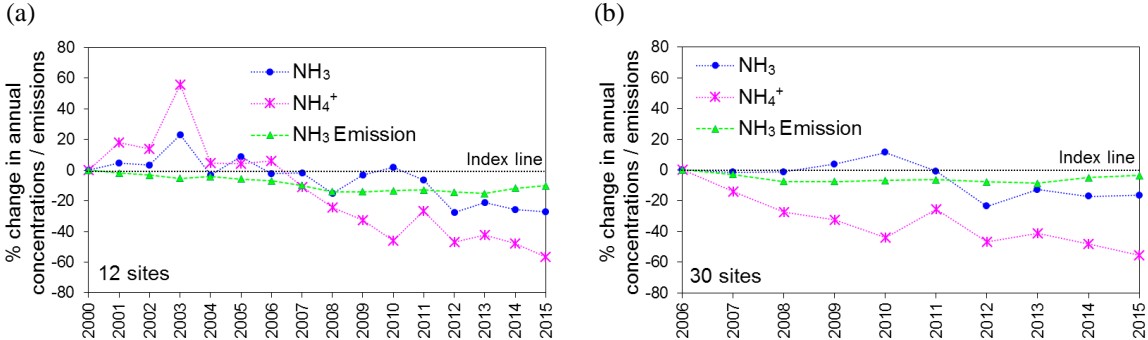

**Figure 17:** Relative trends in UK $NH_3$ emissions (Defra 2017) and in annually averaged $NH_3$ and particulate $NH_4^+$ concentrations from the UK National Ammonia Monitoring Network (NAMN, Tang et al., 2018) for (a) the original 12 AGANet sites for the 16 year period from 2000 to 2015 and (b) the expanded 30 AGANet sites for the 10 year period from 2006 to 2015.




**Figure 18:** (a) Long-term trends in annual mean concentrations of acid gas and aerosol (HNO₃ / NO₃⁻ and SO₂ / SO₄²⁻) from the UK Acid Gases and Aerosol Network (AGANet) and in reduced nitrogen (NH₃ / NH₄⁺) from the UK National Ammonia Monitoring Network (NAMN, Tang et al., 2018) measured at the same time for comparison. Data are the annually averaged concentrations of 12 sites with complete time series from 2000 to 2015, and 30 sites with complete time series from 2006 to 2015. (b) Long-term trends in the gas:aerosol ratio, from the data in figure (a), indicating differences in direction of trends in this ratio with time.



**Table 1:** List of sites in the UK Acid Gas and Aerosol Network (AGANet) with details of locations, start dates and UK-AIR ID (https://uk-air.defra.gov.uk/networks/network-info?view=aganet).

| Site Name | UK-AIR ID | Latitude | Longitude | Start |
|---|---|---|---|---|
| Barcombe Mills | UKA00069 | 50.9191 | 0.0486 | Apr '00 |
| Bush OTC | UKA00128 | 55.8623 | -3.2058 | Sep '99 |
| Cwmystwyth | UKA00325 | 52.3524 | -3.8053 | Sep '99 |
| Eskdalemuir | UKA00130 | 55.3153 | -3.2061 | Sep '99 |
| Glensaugh | UKA00348 | 56.9072 | -2.5594 | Sep '99 |
| High Muffles | UKA00169 | 54.3349 | -0.8086 | Sep '99 |
| Lough Navar | UKA00166 | 54.4395 | -7.9003 | Oct '99 |
| Rothamsted | UKA00275 | 51.8065 | -0.3604 | Sep '99 |
| Stoke Ferry | UKA00317 | 52.5599 | 0.5061 | Sep '99 |
| Strathvaich | UKA00162 | 57.7345 | -4.7766 | Sep '99 |
| Sutton Bonington | UKA00312 | 52.8366 | -1.2512 | Sep '99 |
| Yarner Wood | UKA00168 | 50.5976 | -3.7165 | Sep '99 |
| **New sites added from January 2006** | | | | |
| Auchencorth Moss | UKA00451 | 55.7922 | -3.2429 | Jan '06 |
| Caenby | UKA00492 | 53.3979 | -0.5074 | Feb '06 |
| Carradale | UKA00389 | 55.5825 | -5.4962 | Jan '06 |
| Detling | UKA00481 | 51.3079 | 0.5827 | Feb '06 |
| Edinburgh St Leonards | UKA00454 | 55.9456 | -3.1822 | Jan '06 |
| Goonhilly | UKA00056 | 50.0506 | -5.1815 | Jan '06 |
| Halladale | UKA00314 | 58.4124 | -3.8758 | Jan '06 |
| Harwell | UKA00047 | 51.5711 | -1.3253 | May '06 |
| Hillsborough | UKA00293 | 54.4525 | -6.0833 | Jan '06 |
| Ladybower | UKA00171 | 53.4034 | -1.7520 | Feb '06 |
| Lagganlia | UKA00290 | 57.1110 | -3.8921 | Jan '06 |
| Lerwick | UKA00486 | 60.1392 | -1.1853 | Jan '06 |
| London Cromwell Road 2 | UKA00370 | 51.4955 | -0.1787 | Jan '06 |
| Moorhouse | UKA00357 | 54.6901 | -2.3769 | Jan '06 |
| Narberth | UKA00323 | 51.7818 | -4.6915 | Mar '06 |
| Plas Y Brenin | UKA00493 | 53.1018 | -3.9179 | May '06 |
| Rosemaund | UKA00491 | 52.1214 | -2.6363 | Jan '06 |
| Rum | UKA00276 | 57.0100 | -6.2718 | Feb '06 |





**Table 2:** Comparison of HNO$_3$, HONO, SO$_2$, HCl and aerosol NO$_3^-$, SO$_4^{2-}$, Cl$^-$ concentrations by the Acid Gases and Aerosol Network (AGANet) DELTA method with available measurements from the co-located ChemSpec Daily Annular Denuder system (ADS) at Barcombe Mills (UKA00069). Mean concentrations were derived from the average of daily ADS data for the corresponding DELTA sampling periods (monthly). HNO$_3$ values shown for DELTA and ADS are as calculated from the amount of NO$_3^-$ collected on the denuders and have not been adjusted by a bias correction factor (see Sect. 2.6). Regression plots are shown in Supp. Figure S2.

| | Gases | | | | Particulates | | |
|---|---|---|---|---|---|---|---|
| | HNO$_3$ | HONO | SO$_2$ | HCl | NO$_3^-$ | SO$_4^{2-}$ | Cl$^-$ |
| Linear regression: $R^2$ | 0.813*** | 0.022$^{ns}$ | 0.840*** | 0.282$^{ns}$ | 0.570** | 0.891*** | 0.168$^{ns}$ |
| slope | 0.974$^{ns}$ | 0.033*** | 0.880$^{ns}$ | 0.650$^{ns}$ | 0.570* | 0.691** | 2.339$^{ns}$ |
| intercept | 0.283$^{ns}$ | 0.014$^{ns}$ | -0.163$^{ns}$ | 0.127$^{ns}$ | 1.809*** | 0.204$^{ns}$ | 0.540$^{ns}$ |
| No. observations: $n$ | 11 | 11 | 11 | 10 | 11 | 11 | 11 |
| mean DELTA (µg m$^{-3}$) | 1.56 | 0.03 | 1.75 | 0.40 | 2.59 | 2.10 | 1.24 |
| mean ADS (µg m$^{-3}$) | 1.31 | 0.41 | 2.18 | 0.41 | 1.32 | 2.74 | 0.30 |

Significance level (slope different from 1, intercept = 0): * $p < 0.05$, ** $p < 0.01$, *** $p < 0.001$. $ns$ = not significant ($p > 0.05$)

**Table 3:** Correlation coefficients ($R^2$) for different species across the 30 measurement sites.

| | HNO$_3$ | HCl | SO$_2$ | NO$_3^-$ | Cl$^-$ | SO$_4^{2-}$ | NH$_4^+$ | Na$^+$ |
|---|---|---|---|---|---|---|---|---|
| **HNO$_3$** | 1.00 | 0.25*** | 0.39*** | 0.45*** | 0.07*** | 0.54*** | 0.49*** | 0.02* |
| **HCl** | - | 1.00 | 0.21*** | 0.14*** | 0.01$^{ns}$ | 0.24*** | 0.19*** | 0.04** |
| **SO$_2$** | - | - | 1.00 | 0.30*** | 0.00$^{ns}$ | 0.47*** | 0.37*** | 0.01$^{ns}$ |
| **NO$_3^-$** | - | - | - | 1.00 | 0.00$^{ns}$ | 0.61*** | 0.90*** | 0.02$^{ns}$ |
| **Cl$^-$** | - | - | - | - | 1.00 | 0.04** | 0.01$^{ns}$ | 0.79*** |
| **SO$_4^{2-}$** | - | - | - | - | - | 1.00 | 0.73*** | 0.00$^{ns}$ |
| **NH$_4^+$** | - | - | - | - | - | - | 1.00 | 0.00$^{ns}$ |
| **Na$^+$** | - | - | - | - | - | - | - | 1.00 |

Significance level: * $p < 0.05$, ** $p < 0.01$, *** $p < 0.001$. $ns$ = not significant ($p > 0.05$)

**Table 4:** Comparison of mean concentrations from the original 12 Acid gases and Aerosol Network (AGANet) sites *vs* the expanded 30 AGANet sites for the different gas and aerosol components. NH$_3$ and NH$_4^+$ measured at the same time in the UK National Ammonia Monitoring Network (NAMN, Tang et al., 2018) are also included for comparison. Each data point are the mean ± SD of annual mean concentrations over the period 2006 to 2015.

| | Mean concentration (2006 – 2015), µg m$^{-3}$ | | | | | | | | | | |
|---|---|---|---|---|---|---|---|---|---|---|---|
| | HNO$_3$ | SO$_2$ | HCl | NO$_3^-$ | SO$_4^{2-}$ | Cl$^-$ | Na$^+$ | Ca$^{2+}$ | Mg$^{2+}$ | NH$_3$ | NH$_4^+$ |
| **12 sites (mean±SD)** | 0.31± 0.06 | 0.66 ± 0.24 | 0.25 ± 0.04 | 1.40 ± 0.31 | 0.73 ± 0.25 | 1.33 ± 0.09 | 0.75 ± 0.07 | 0.04 ± 0.03 | 0.06 ± 0.01 | 1.18 ± 0.16 | 0.62 ± 0.18 |
| **30 sites (mean±SD)** | 0.34 ± 0.06 | 0.70 ± 0.25 | 0.28 ± 0.04 | 1.41 ± 0.26 | 0.74 ± 0.23 | 1.54 ± 0.09 | 0.84 ± 0.08 | 0.04 ± 0.02 | 0.07 ± 0.01 | 1.40 ± 0.16 | 0.61 ± 0.16 |





**Table 5:** Summary of Mann-Kendall (MK) and Linear Regression (LR) time series trend analysis on annually averaged gas and aerosol concentrations from the UK Acid Gases and Aerosol Monitoring Network (AGANet) for the 12 sites that were operational over the period 2000 to 2015. $NH_3$ and $NH_4^+$ concentrations measured at the same time in the UK National Ammonia Monitoring Network (NAMN, Tang et al., 2018) are also included for comparison. For the MK tests, the 95% confidence interval (CI) for the median trend and relative change are also estimated.

| 2000 - 2015 (12 sites: annual data) | Mann-Kendall (MK) | | Linear Regression (LR) | | |
|---|---|---|---|---|---|
| | [a]Median annual trend & [95% CI] ($\mu$g y$^{-1}$) | [b]Relative median change 2000-2015 & [95% CI] (%) | [c]Annual Trend ($\mu$g $NH_3$ y$^{-1}$) | [d]Relative change 2000-2015 [%] | $R^2$ |
| $HNO_3$ | -0.0135 [-0.0067, -0.0180] | -45** [-26, -55] | -0.0126 | -42** | 0.549 |
| $SO_2$ | -0.1010 [-0.0729, -0.1250] | -81*** [-72, -91] | -0.1069 | -84*** | 0.894 |
| $HCl$ | -0.0057 [-0.0020, -0.0100] | -28*** [-11, -42] | -0.0055 | -26** | 0.469 |
| $NH_3$ | -0.0300 [-0.0125, -0.0433] | -30** [-13, -39] | -0.0312 | -30*** | 0.611 |
| $NO_3^-$ | -0.0810 [-0.0520, -0.1125] | -52** [-37, -63] | -0.0794 | -51*** | 0.557 |
| $SO_4^{2-}$ | -0.0750 [-0.0450, -0.-0988] | -69** [-52, -82] | -0.0757 | -70*** | 0.786 |
| $Cl^-$ | 0.0079 [-0088, 0.0236] | 9.6[ns] [-9.5, 33] | 0.0097 | +12[ns] | 0.143 |
| $NH_4^+$ | -0.0500 [-0.0375, -0.0675] | -62** [-51, -74] | -0.0529 | -64*** | 0.754 |

Significance level: * $p < 0.05$, ** $p < 0.01$, *** $p < 0.001$, [ns] non-significant ($p > 0.05$)
[a]Median annual trend = fitted Sen's slope of Mann-Kendall linear trend (unit = $\mu$g y$^{-1}$)
[b]Relative median change calculated based on the estimated annual concentration at the start ($y_0$) and at the end ($y_i$) of time series computed from the Sen's slope and intercept (=100*[($y_i$-$y_0$) /$y_0$])
[c]Annual trend = fitted slope of linear regression (unit = $\mu$g $NH_3$ y$^{-1}$)
[d]Relative change calculated based on the estimated annual concentration at the start ($y_0$) and at the end ($y_i$) of time series computed from the slope and intercept (=100*[($y_i$-$y_0$) /$y_0$])





**Table 6:** Summary of Mann-Kendall (MK) and Linear Regression (LR) time series trend analysis on annually averaged gas and aerosol concentrations from the UK Acid Gases and Aerosol Monitoring Network (AGANet) for the 30 sites that were operational over the period 2006 to 2015. $NH_3$ and $NH_4^+$ concentrations data measured at the same time from the UK National Ammonia Monitoring Network (NAMN, Tang et al., 2018) are also included for comparison. For the MK tests, the 95% confidence interval (CI) for the median trend and relative change are also estimated.

| 2006 - 2015 (30 sites: annual data) | Mann-Kendall (MK) | | Linear Regression (LR) | | |
|---|---|---|---|---|---|
| | [a]Median annual trend & [95% CI] (µg $NH_3$ $y^{-1}$) | [b]Relative median change 2000-2015 & [95% CI] (%) | [c]Annual Trend (µg $NH_3$ $y^{-1}$) | [d]Relative change 2000-2015 [%] | $R^2$ |
| $HNO_3$ | -0.0167 [-0.0075, -0.0200] | -36* [-18, -41] | -0.0137 | -30* | 0.542 |
| $SO_2$ | -0.0717 [-0.0300, -0.0108] | -60*** [-33, -73] | -0.0731 | -60*** | 0.788 |
| HCl | -0.0088 [0.0000, -0.0200] | -24* [0.0, -47] | -0.0073 | -21[ns] | 0.371 |
| $NH_3$ | -0.0312 [0.0033, -0.0625] | -18[ns] [+2.0, -31] | -0.0366 | -21* | 0.459 |
| $NO_3^-$ | -0.0900 [-0.0580, -0.1300] | -43*** [-30, -56] | -0.0809 | -39*** | 0.871 |
| $SO_4^{2-}$ | -0.0675 [-0.0233, -0.1167] | -54** [-25, -78] | -0.0646 | -53*** | 0.718 |
| $Cl^-$ | -0.0075 [+0.0167, -0.0300] | -4.2[ns] [+12, -16] | -0.0086 | -4.9[ns] | 0.091 |
| $NH_4^+$ | -0.0480 [0.0267, -0.0700] | -49** [-33, -64] | -0.0462 | -48*** | 0.790 |

Significance level: * $p < 0.05$, ** $p < 0.01$, *** $p < 0.001$, [ns] non-significant ($p > 0.05$)
[a]Median annual trend = fitted Sen's slope of Mann-Kendall linear trend (unit = µg $y^{-1}$)
[b]Relative median change calculated based on the estimated annual concentration at the start ($y_0$) and at the end ($y_i$) of time series computed from the Sen's slope and intercept (=100*[($y_i$-$y_0$) /$y_0$])
[c]Annual trend = fitted slope of linear regression (unit = µg $NH_3$ $y^{-1}$)
[d]Relative change calculated based on the estimated annual concentration at the start ($y_0$) and at the end ($y_i$) of time series computed from the slope and intercept (=100*[($y_i$-$y_0$) /$y_0$])



**Table 7:** Comparison of % change in estimated UK NO$_x$, SO$_2$ and NH$_3$ emissions reported by the National Atmospheric Emission Inventory (NAEI) (data from http://naei.defra.gov.uk/) with % change between 2000-2015 (12 sites with complete time series) and between 2006-2015 (30 sites with complete time series) in annually averaged HNO$_3$ / NO$_3^-$ and SO$_2$ / SO$_4^{2-}$ concentrations from the UK Acid Gas and Aerosol Monitoring Network (AGANet), and annually averaged NH$_3$ / NH$_4^+$ concentrations from the UK National Ammonia Monitoring Network (NAMN, Tang et al., 2018).

| Components | 2000 – 2015 (12 sites) | | | 2006 – 2015 (30 sites) | | |
|---|---|---|---|---|---|---|
| | UK emissions % change | MK Sen Slope % relative median change[a] | LR % relative change[b] | UK emissions % change | MK Sen slope % relative median change[a] | LR % relative change[b] |
| Gas HNO$_3$ | -58 (NO$_x$) | -45** | -42** | -41 (NO$_x$) | -36* | -43* |
| Particulate NO$_3^-$ | | -52*** | -51*** | | -30** | -39*** |
| Gas SO$_2$ | -80 (SO$_2$) | -81*** | -84*** | -64 (SO$_2$) | -60*** | -60*** |
| Particulate SO$_4^{2-}$ | | -69*** | -70*** | | -54** | -53*** |
| Gas NH$_3$ | -10 (NH$_3$) | -30** | -30*** | -3.5 (NH$_3$) | -18[ns] | -21* |
| Particulate NH$_4^+$ | | -62*** | -64*** | | -49** | -48*** |

Significance level: * $p < 0.05$, ** $p < 0.01$, *** $p < 0.001$, [ns] non-significant ($p > 0.05$)
[a]Relative median change calculated based on the estimated annual concentration at the start ($y_0$) and at the end ($y_i$) of time series computed from the Sen's slope and intercept ($=100*[(y_i-y_0)/y_0]$)
[b]Relative change calculated based on the estimated annual concentration at the start ($y_0$) and at the end ($y_i$) of time series computed from the slope and intercept ($=100*[(y_i-y_0)/y_0]$)