# Peer review of "Acid gases and aerosol measurements in the UK (1999-2015): regional distributions and trends"

_Atmospheric Chemistry and Physics, 2018_

## Referee Comment (RC1) · C.R. Flechard (Referee) · 29 Jun 2018

Reviewer's comments on ACP-2018-489 manuscript "Acid gases and aerosol measurements in the UK (1999-2015): regional distributions and trends" by Tang et al.

General Comments

The manuscript describes the results of a long term (15-year) national-scale monitoring network for atmospheric acid gases and aerosols in the UK. Trends in concentrations since ca 2000 are analysed alongside reported changes in emissions, showing non-linearities between emission and concentration changes, caused by shifts in gas/aerosol partitioning. The dataset is fairly unique worldwide and well worth publishing in ACP. The paper is well written, with an abundance of detail and perhaps too

many figures, with the same data sometimes shown twice in different places. A little streamlining could improve the readability of the paper, as one tends to get swamped by the large number of figures and tables. I would recommend some minor changes before final publication (see below).

The mean weakness in the quality of the dataset is likely the large uncertainty in HNO3 caused by interferences by other NOy compounds on potassium carbonate coated denuders. This is mentioned in the methods but not referred to later on in the discussion in relation to trends in measured HNO3 and reported NOx emissions. With a flat and constant correction factor of 0.45 for HNO3 measured from K2CO3 coated denuders (meaning that 55% of the raw concentration is substracted to provide a corrected number), one can wonder whether the apparent decrease in HNO3 since 2000 is significant, or if the slope of the apparent decrease has any meaning. With large changes in NOx emissions and in the general pollution climate of the UK over the last 20 years, and therefore with possibly large changes in the ratios of HNO3 to the interfering NOy gases (NO2, HONO, PAN, etc), it is risky to assume a constant 0.45 correction factor for the whole period, and also across the whole country, given the large differences in pollution profiles between the sites of the network.

Specific Comments

p6, l4, '...sampling rate of 0.2-0.4 l/min...' Please mention at this stage, or just below in the paragraph describing the aerosol collection system, what the particle size cut-off is for the DELTA sampler (mentioned later on p10, l3). It is important to know what the size spectrum of collected aerosols is, and that some (coarse) particles are not sampled, eg dust, large marine aerosols.

p7, l6-7, change of analytical labs from Harwell to CEH Lancaster in 2009: was there a transitional period of overlapping parallel measurements by the two labs, to make sure no bias was introduced in the long term time series by the change of laboratory?

p7, l26, '...flagging up occurrences of poorly coated denuders and/or sampling issues...'

Another possibility is that concentrations are so large that the first denuder saturates and thus much is collected by the second denuder. This can happen for NH3 at agricultural sites after fertilisation; it is much less likely for acid gases due to lower concentrations, unless perhaps at some polluted urban stations?

p8, l15, the term 'bias' is used in relation to the 0.45 correction factor for HNO3, in the title of 2.6 and also other parts of the text. This is perhaps misleading as a bias suggests an offset, while the multiplicative correction applied acts on the span.

Further, in the Tang et al 2015 report, the authors write that '... It is recommended that a correction factor of 0.45 be applied to the historic HNO3 measurements. The range of ratios was 0.44±0.15 (±2SD), i.e. 0.29-0.59, therefore it is reasonably likely that the value lies between 0.4 and 0.5. Therefore a correction factor of 0.45 should be applied...' It is quite clear that the percentage of non-HNO3 NOy compounds that is measured after extracting K2CO3-coated denuders depends on the relative abundances of these gases compared with HNO3, as well as their collection efficiencies on K2CO3 and their oxidation/reaction rate following adsorption. I would expect large seasonal changes, and large spatial/geographical variations, in these concentrations and the associated chemical processes, as reflected in the observed 0.29-0.59 range. Applying the same correction factor at all sites of the network, that range from remote to coastal to rural to sub-urban and urban, does not seem to be adequate. This is hinted at in the Eskdalemuir example of Fig. 3, where applying the large 0.45 multiplier makes the DELTA TIN values diverge from the EMEP filter pack measurements, ie at this rural background site the need for such a large correction is not warranted.

The correction factor should account for the differences in pollution climates between sites, and also for changes over the 20-year period. Could an empirical correction be derived from chemical transport modelling (eg EMEP4UK), whereby the ratios of modelled HNO3 to NO2, HONO, PAN, etc, are used to construct a geographically- and temporally-varying index to drive the correction function? The HNO3 data reported in Tang et al (2015) for NaCl vs K2CO3 coating, with measurements made in contrasted

situations (rural, urban, remote, see Table 1 in that report), may be used for calibrating such a function.

p8-11, section 2.7 Performance of the DELTA method: strictly speaking, this section describes measurement results from intercomaprison experiments or even long term datasets (Bush, Eskdalemuir) does not belong in Methods, and should be moved to the beginning of Section 3- Results

p10, l12: Ca++ and Mg++ concentrations were near detection limits because they are mostly in the coarse fraction, with particle sizes near or above the DELTA cut-off. How much of the NaCl is similarly not collected by the DELTA system?

Also, in relation to the DELTA v. ADS intercomparison, the loss of NO3, Na and Cl on the surface of the cyclone is put forward as an explanation for the lower ADS aerosols concentrations (compared with DELTA) (p10, l10), but why in that case is SO4= 23% larger in the ADS?

p13, l2: the highest HCl concentrations are in the SE and SW of England, but also in the Midlands

p13, l5: '...Further away from the coast and influence of marine aerosol, the smallest concentrations of Cl and Na+ are measured in the west of the country (Lough Navar in Northern Ireland...' Lough Navar is very near the coast (10km) and yet NaCl concentrations are very low (similar to concentrations in the Midlands), compared with all other western sites in the network (Fig. 5); what could be the reason?

p13, l5: For Cl- and Na+, '...largest concentrations at coastal sites in the south (Barcombe Mills) and west (Yarner Wood)...'; actually the highest concentrations are at Goonhilly at the SW tip of Cornwall.

p13, l9-10, '...There is however no clear spatial pattern for Ca2+, with concentrations that are mostly at or below LOD...' For both Ca and Mg, which are mostly in the coarse fraction, it may be argued that the DELTA system does not allow a realistic assessment

of the total concentration, because a large share of coarse particles are not collected. Please comment.

Further, our own tests with DELTA systems at INRA indicated very substantial losses for Mg and Ca in all the non-filter parts of the sampling train (particularly the 6-mm diameter LDPE elbow connecting the 2nd acid denuder to the first NH3 denuder, Fig. S1), which are therefore not measured on the filter. We analysed the loss fraction LDPE / (LDPE + den + filter) for all compounds; for NH4+ and NO3- this was less than 5%; for Cl- and Na+ this was 5-10%; for SO4= and Mg2+ this was 10-15%; while for Ca2+ this was 30-40%. Beyond the question of coarse aerosols that were not sampled at all (did not enter the sampling train), there is the question of those coarse aerosols that 'did not make it' to the filter pack. Did the authors carry out similar tests, and could the results be shown in the supplement? It may be that the new straight design for the DELTA sampling train allowed a reduction of these losses ? Please comment.

p15, l1: This section 3.4 is mostly about sub-annual (seasonal) variations, so could be re-named 'Seasonal variations in acid gases and aerosols', as opposed to long term trends of Sections 3.5-3.6

p15, l9-10: '...In spring, the peak in HNO3 and NO3...' Fig.7 does not actually show any spring peak for HNO3; the late winter (Feb-Mar) concentrations are only marginally higher (but not significantly different accoring to the error bars) than the rest of the year? The opening sentence of the paragraph should read '...maximum in late winter and early spring...'

p15, l22, '...this contributes to the winter minimum in NH4NO3...' : the minimum NO3- actually occurs in July?

p16, l9-13: how far should seasonal cycles for Mg and especially Ca be discussed, given the low collection efficiency (and thus high uncertainty) of filter data (see my comment above on aerosol size cut-off and losses in sampling train for these large aerosols)?

p17 and beyond, general comment on sections 3.5-3.6: a linear regression is fitted to all datasets from 1999 through 2015, but looking closely at the 15-yr time series for the 12 sites (eg Fig. 12-13), for HNO3, NO3-, SO4=, NH4+, NH3, it appears that concentrations were rather stable (with some interannual variability but no trend) in the period 2000-2007, and then only started declining after 2007 . The only exception is SO2 with a continuous decline all the way. Fitting a linear trend is helpful to quantify an multi-annual rate of decrease (which is what you do), but is not an accurate representation of the time course of concentrations. Can you think of any plausible explanation for a change of course around the year 2007: change or implementation of pollution control policies? Decadal change in weather patterns? It might be useful to show (in the supplement)a summary of weather patterns for all sites of the network, the 15-yr time course of temperature, rainfall, wind speed etc.

p18, l6: '...The long-term time series in annually averaged concentrations of the gas and aerosol components are shown in Figure 12a and Figure 12b...': would it be possible to show, alongside the measured DELTA time series, the modelled NO/NO2 time series (from a CTM, eg EMEP4UK) for the same sites? In a way this would account for both NOx emission changes as well as climatic variability over the period.

p18, l12 '...The exceptions are Na+ and Cl- that have higher mean concentrations...' : Na+ is not shown in Fig.12.

Figures 13, 14: use only one type of regression to simplify the figures (LR and MK give almost identical results)

p20, l24-25 '...the reduction in SO2 emission and measured concentration is accompanied by a smaller negative trend in particulate SO4=...', and l27, '...The smaller decrease in particulate SO4= compared with its gaseous precursor, SO2, is similar to that observed at Eskdalemuir...'. Question: Is the smaller reduction rate in SO4= (compared with SO2) a reflection of the fact that increasingly in the UK, total sulphate includes a larger and larger fraction of marine sulphate, such that the decrease in anthropogenic

SO4= (resulting from SO2 abatement) has a increasingly small effect on total sulphate? Is it possible to re-calculate the SO4= trend separately for coastal and inland (eg Midland/London) sites?

p21, l5-22: The argument about the NH3/SO2 ratio impacting the dry deposition velocities of SO2 and NH3 was developed in the 1980s and early 1990s, when SO2 concentrations were still very large in W. Europe. It is no longer sufficient to consider the NH3/SO2 ratio alone, since SO2 no longer massively dominates the acid load in W. European atmospheres. Instead, the ratio NH3/(2*SO2 + HNO3 + HCl) should be computed to analyse long terme trends, as shown in Fowler et al. (Atmospheric Environment 43 (2009) 5193–5267, see Fig. 4.5). It is the combined effects of all acids and NH3 that determines the pH of ecosystem/vegetation surfaces and hence their sink strength for water-soluble pollutants.

p21, l30: '...The increase in ratio of HNO3:NO3- is similar to changes in upward trend in gas-aerosol partitioning between NH3 and NH4+ over time...': what do you call similar? For HNO3/NO3-, the ratio increases by ∼20%, while for NH3/NH4+, the ratio increases by 100% (according to Fig. 18) ?

p22, l11-12, '...a change in the particulate phase from (NH4)2SO4 to NH4NO3. This change is expected to increase residence times of NH3 and HNO3 in the atmosphere...' I am not convinced the shift from ammonium sulphate to ammonium nitrate should increase the residence time, since NH3 and HNO3 will deposit faster (higher deposition velocities) than either aerosol form?

p22, l12 '...expected to increase residence times of NH3 and HNO3 in the atmosphere...' and p22, l15 '...NH3 and NOx emitted will deposit more locally with a smaller footprint...': these two statements appear to contradict each other?

p22, l12: perhaps another way to analyse this trend is to calculate the (changing) linear regression slopes of NO3- vs NH4+ and SO4= vs NH4+, for each year of the 2000-2015 period (as in Fig. 6b), and examine how the two slopes change over time,

as an indicator of the fraction of the total NH4+ that neutralizes NO3- and SOA= and its trend over time.

p24, l14 '...Higher concentrations of the NH3 and HNO3 in the atmosphere will deposit more locally...' But then, NH3 and HNO3 concentrations are actually decreasing; they are not higher than before?

Technical Corrections

Units: different units are used. They should either be harmonized, or else each figure should state explicitly what the unit is, especially for the difference between element (N,S) based or molecule (HNO3, SO2) based. For example, mean HNO3 at the Bush site is reported as 0.55 $\mu$g m-3 in Fig.2 (average of 0.54 and 0.56 for samplers A and B), while the color code on the concentration map (Fig. 5) indicates a concentration in the range 0.15-0.25, from which I infer that Fig.2 is $\mu$g HN03 m-3, while Fig.5 is $\mu$g N m-3 ?

Similarly, p12, l16, is the Cromwell site HNO3 concentration 1.3 $\mu$g HNO3 m-3, or 1.3 $\mu$g HNO3-N m-3? From Figure 5 I expect it is the latter (N, not HNO3 as written in the text). Further below, are the SO2 concentrations at Sutton Bonington given as $\mu$g SO2 m-3, or in fact $\mu$g SO2-S m-3 ? Given that the map in Fig. 5 gives numbers in $\mu$g N or $\mu$S per m3, it would be good to use the same units. Thus I would recommend to check carefully throughout the text in this paragraph and in the whole paper and make the necessary text changes to eliminate the ambiguity in units.

p4, l26-27, delete '...that is also deployed at some CASTnet sites (Rumsey and Walker, 2016).' (already mentioned same page, l14)

p4, l32, suggest change 'temporal' to 'seasonal'

p5, l9-12: this mostly repeats what was said in the introduction p4, l20-25

p7, l20, please provide the equation for the calculation of the denuder capture efficiency

p35: Figure 2 contains scatter plots and a statistical summary table for the Bush DELTA intercomparison (parallel sampling). It would be good to adapt the same or similar style of display for the other intercomparisons (scatter plots + stats table). Thus for the comparison with ADS (2.7.2), take Fig. S2 out of the supplement and stack it above the statistics given in Table 2. Similarly for the intercomparisons of DELTA vs EMEP TIA/TIN (add statistical table), as is already also done for DELTA vs Bubble/FP Eskdalemuir (Fig 4).

p11, sections 2.8 and 3.6: throughout the time series trend analysis, both linear regressions and non-parametric MK tests are used, but as far as I can see, there is essentially no difference between the slopes for any of the pollutant time series. To improve readability and reduce unnecessary redundant information, I would suggest to stick to just one of the methods; it would suffice to say in the methods that both regressions were used and no significant differences were found, and thus henceforth only one regression is displayed.

p12, l22, '...A peak MONTHLY concentration of...'

p12,l28, '...expected to be more SPATIALLY homogeneous...'

p15, l19 '...in summer promotes AEROSOL dissociation...'

p17, l12 change to '...are available SINCE 1989...'

p18, l31, '...To QUANTIFY changes...'

p18, l32, the unit for the annual trend is $\mu$g HNO3-N m-3 y-1

p19, l3: '...The LR % annual trends for each time series...' Delete 'annual', since the % reduction are not expressed per year, but over the whole period ? Note that if the concentration reduction were a constant percentage every year, say -10% per year, then the overall time course over 15 years would not look linear, but exponential: if yr1=100, then yr2=90, yr3=81, yr4=72.9, yr5=65.6, ...yr15=20.6

p19, l6: same as above, delte 'annual'

p21, l17-18 '...The dry deposition... IS known to be enhanced...'

p23, l9, delete 'from coal combustion'

p24, l1, '... modest reductionS in HNO3...' (plural)

p24, l12, '...smaller THAN emission trends...'

All figures: when the units displayed on axes or legends are given in $\mu$g m-3, please specify whether this is on an element basis (NH3-N, HNO3-N, SO2-S) or molecule basis (NH3, HNO3, SO2)

Figure 8: "...Average annual cycles in the ratios of gas:aerosol component concentrations ($\mu$g m-3)...' The unit for the ratio is not $\mu$g m-3, it must be dimensionless, or mol mol-1?

Figures 13-14: keep only one of the two trend lines (LR or MK); and delete Fig.14 but add the n=30 datapoints to Fig. 13 as a different symbol shape or color

Figure 18: the left-hand side panels show the same data as Figs. 13-14 and should therefore not be repeated here.

---

## Referee Comment (RC2) · Anonymous Referee #2 · 20 Aug 2018

Review of Tang et al This is a nice paper that presents the analysis of UK monitoring sites and documents changes in the atmospheric composition over the past fifteen years. This results in a comprehensive view of how the atmosphere has changed during a period of large emission reductions. The authors then discuss the changes in emissions with the observed changes in atmospheric composition. Overall, the manuscript is well written but could benefit from some additional editing in areas outlined below.

General comments

1. Currently the manuscript is primarily focused on documenting trends and events and with a smaller focus on the changes in atmospheric composition and pollutant fate due to changes in emissions. This manuscript would benefit from a bit more focus. I

suggest focusing more on the trends and how they relate to emission changes and less on specific events captured in the data.

2. The discussion of the trends of NH3 and HNO3 are sometimes a bit difficult to follow as the change in aerosol composition and loading over the time frame of the measurements impacts the gas phase concentrations. Consider discussing these trends as total nitrate (gaseous HNO3 + aerosol NO3) and NHx (gaseous NH3 + aerosol NH4).

3. There are lots of small sections in this manuscript, some consisting of single sentences. Consider combining them into more general sections. Specifically, 2.3.1-2 and 2.5-6.

4. Many sentences leading paragraphs are structured as "For {atmospheric constituent}, . . .". This is a bit formulaic and the authors may want to revise these sentences.

Specific comments

1. Abstract: I find the final two sentences of the abstract to be the most compelling. There is a lot of detail, primarily on page 1, that would be better suited for the results section. Consider summarizing the text on the spatial and temporal trends and better connecting them to the changes in atmospheric HNO3 and NH3.

2. Abstract Page 2 lines 5-6: ". . . indications that the atmospheric lifetime of HNO3 and NH3 has increased . . .". This does not seem correct to me. The lifetime of these gases has not increased but rather the phase/composition of these species have. There are now more gaseous and less aerosol bound NO3 and NH3 due to changes in SO2. This likely decreases the atmospheric lifetime of total nitrate and reduced nitrogen compounds as NH3 and HNO3 typically dry deposit faster than aerosol NO3 and NH4.

3. Page 4 lines 17-29: This paragraph contains similar information as the previous paragraph. Consider combining it with the previous paragraph

4. Page 6 lines 21-22: This is an awkward introductory sentence for this paragraph.

Consider revising or adding an introductory paragraph that introduces the importance of the denuder base coating.

5. Sections 2.5: Are data that failed the quality checks removed from the analysis?

6. Section 2.5 iv) What is the criteria to determine anomalies and outlies?

7. Section 2.6 Line 22: Does the empirical factor used for HNO3 bias correction exhibit any dependence on season, temperature or solar radiation? If the bias is due to oxidants, then I would expect a dependence in the bias on seasonal and environmental parameters.

8. Page 9 lines 28-29: The mean difference between the measurements are given here but what is the scatter between the measurements and the median difference. A correlation coefficient would provide some information about the scatter and a median difference would indicate how normal the distribution is and if the bias is being driven by high values in one of the measurement techniques.

9. Page 9 line 32: Difference in the instrumentation flow rates and/or inlets could result in the instruments measuring different sized aerosols and my influence the differences in SO4.

10. Page 16 lines 1-2: The peaks in NHx and SO4 in the spring may just be coincidental. The spring time could also be a time in which the aqueous formation pathway of SO4 is at its maximum or the SO2 emissions from heating or transportation may be larger. In the US, the SO4 concentrations typically peak in the summer while the NH3 concentrations peak in the spring.

11. Page 16 Line 5: "Na+ and Cl-" have highest concentrations during winter . . .' Is salt used for the treatment of road surfaces in the Winter in the UK?

12. Page 20 line 18: Significant has a specific statistical meaning. I think "larger" would be a more appropriate.

13. Page 21 line 5: "... SO2 towards it being dominated by NH3, ..." This appears to be a bit binary. There are lots of constituents in the air, many of which were note measured here. More context is needed.

14. Page 22 line 12: "expected to increase residence times of NH3 and HNO3 in the atmosphere" If we are in an NH3 limited environment, I can see how this would increase HN4NO3 and how that could increase the atmospheric lifetime of HNO3 as it is partitioned to NO3 aerosols. However, I do not see how this increases the NH3 lifetime. NH3 will preferentially partition with SO4, which is more thermodynamically stable than NH4NO3, this should decrease the lifetime of NH3 if anything as the NH4NO3 will evaporate where the (NH4)2SO4 aerosol would not.

15. Figure 8: I am happy to see a measure of scatter on these plots as the SD. However, a 5% and 95% CI would be more informative as it would give the reader an idea about the distribution of the data.

---

## Author Comment (AC1) · 8 Oct 2018

RESPONSE TO REVIEWER 1 C.R. Flechard (Referee)

The authors thank Dr. Flechard for his constructive comments and for taking the time to look at all the details described in the manuscript. We have carefully considered all comments. Please refer to the specific responses.

1. General observations The mean weakness in the quality of the dataset is likely the large uncertainty in HNO3 caused by interferences by other NOy compounds on potassium carbonate coated denuders. This is mentioned in the methods but not referred to

later on in the discussion in relation to trends in measured HNO3 and reported NOx emissions. With a flat and constant correction factor of 0.45 for HNO3 measured from K2CO3 coated denuders (meaning that 55% of the raw concentration is substracted to provide a corrected number), one can wonder whether the apparent decrease in HNO3 since 2000 is significant, or if the slope of the apparent decrease has any meaning. With large changes in NOx emissions and in the general pollution climate of the UK over the last 20 years, and therefore with possibly large changes in the ratios of HNO3 to the interfering NOy gases (NO2, HONO, PAN, etc), it is risky to assume a constant 0.45 correction factor for the whole period, and also across the whole country, given the large differences in pollution profiles between the sites of the network

Author Response: Additional text has been added under new heading "Section 3.3: Uncertainties in HNO3 determination" to address the reviewer's comments. See below: "HNO3 data were corrected for sampling artefacts in the measurements with an empirical correction factor of 0.45 (see section 2.6). Interferences in HNO3 determination arise through the simultaneous collection of reactive oxidized nitrogen species on the K2CO3 coating that forms nitrate ions in the aqueous extracts of exposed denuders. Potential interfering species include HONO, NO2, N2O5 and PAN, as well as other inorganic and organic nitrogen species. HONO is most likely to contribute to the interference, since it is collected effectively on a carbonate coating and concentrations of HONO have been reported to be comparable to, and in some places exceed HNO3 in the UK (e.g. Kitto & Harrison 1992, Connolly et al. 2016). Interference from NO2 should be small, since the reactivity of a carbonate coating surface towards NO2 is low (Allegrini et al., 1987), with capture of NO2 on carbonate ranging from 0.5 to 5 % (Allegrini et al., 1987, Benner et. al. 1991, Fitz 2002) and their concentrations are also small at rural AGANet sites (< 10 $\mu$g NO2 m-3; Conolly et al., 2016). Tests by Steinle et al. (2009) on the AGANet K2CO3/glycerol coated denuders also confirmed low capture (ca 3 %) of NO2. The correction factor was derived from two years of field intercomparison measurements at five sites across a range of pollutant concentrations across the UK, from a clean rural background site in Southern Scotland

(Auchencorth) to a polluted urban site (London Cromwell road) in Southern England (Tang et al., 2015). It is recognised that the correction factor to derive the "real HNO3" signal from the carbonate coated denuders will be dependent on the relative concentrations of HNO3 to interfering species present in the atmosphere and likely to be both site and season specific. The 2 years of data indeed show this variability between sites and between seasons. Given the complexities of atmospheric chemistry of the large family of oxidised nitrogen species, further work is clearly needed to understand what the carbonate denuders is measuring, before an improved correction algorithm for the HNO3 data can be developed with any confidence. The empirical 0.45 HNO3 is therefore at present a best estimate across a range of pollutant concentrations and seasons encountered in the UK, based on available test data from 5 sites. At the cleanest rural sites (e.g. Eskdalemuir), where a much smaller HONO and NO2 interference of the DELTA HNO3 signal is expected, the HNO3 concentrations may be under-estimated after correction. This may partly explain the slope deviating from unity in the comparison of corrected AGANet TIN with EMEP filter pack TIN data (slope = 0.835, R2 = 0.95) at Eskdalemuir (see section 3.1.2). Conversely, at more polluted sites such as London that are affected by a larger interference from HONO and NO2, the HNO3 determination may be over-estimated after correction. Apart from two urban sites (London and Edinburgh), all other sites in the AGANet are rural, located away from traffic, and the 0.45 correction factor should be more representative. Since January 2016, the DELTA denuder sample train configuration in AGANet was changed to two NaCl coated denuders (selective for HNO3, e.g. Allegrini et al., 1987), with a third K2CO3/glycerol coated denuder to collect SO2. At three sites (Auchencorth, Bush OTC and Stoke Ferry), parallel measurements of the old configuration (two K2CO3/glycerol coated denuders) and new configuration (two NaCl coated denuders + K2CO3/glycerol coated denuder) were conducted over 12 months in 2016. In the new configuration, nitrate measured on the NaCl denuders are reported as HNO3, whereas nitrate on the K2CO3 denuder are assumed to come from other oxidised nitrogen species and are not reported. Comparing the sum of nitrate concentrations from the new (2xNaCl + 1xK2CO3) with the old

(2xK2CO3) configurations indicated matching capture of total nitrate by the two parallel systems (new:old nitrate ratio = 0.95). A comparison of nitrate concentrations on the 2xNaCl denuders only (new configuration) with the 2xK2CO3 denuders (old configuration) yielded an average ratio of 0.42, lending further support to the 0.45 empirical factor. Additionally, the new sample train configuration is providing an extensive dataset which will allow the magnitude of HNO3 interference at each site to be quantified, by comparing the amount of nitrate measured on the NaCl and K2CO3 coated denuders. Initial analysis of 2016 data (unpublished data) showed that the mean ratio of nitrate on NaCl:K2CO3 of all sites was 0.44, ranging from 0.31 (Bush OTC) to 0.59 (Moorhouse). Seasonally, the average monthly ratio (taken as the mean across all sites for each month) was lowest in winter (0.25 in December and 0.27 in January) and highest between May to June (0.59, 0.56 and 0.57). It may therefore be possible to derive an improved correction algorithm that is both site and season specific, and work is ongoing to make this assessment. A detailed assessment of sampling artefacts in the DELTA method and the effects of a method change in the AGANet forms the subject for a next paper. " Additional text also added to revised/expanded text in Section 3.6 Seasonal variation in acid gases and aerosols (paragraph 2): "HNO3 is a secondary product of NOx, but NOx emissions are dominated by vehicular sources which are not expected to show large seasonal variations. Seasonal changes in chemistry and meteorology are therefore more likely to be a source of the observed variations in HNO3 and NO3- (Figure 8). A weak seasonal cycle is observed in HNO3, with slightly higher concentrations in late winter and early spring that may be due to photochemical processes with elevated ozone in spring (AQEG 2009) leading to formation of HNO3 during this period (Pope et al., 2016). As discussed in section 3.3, a constant correction factor was applied to all HNO3 data, which does not take into account seasonal dependency. The concentrations in HNO3 may therefore be over-estimated in winter (less HNO3 formed from photochemical processes) and under-estimated in summer (larger HNO3 concentrations due to increased .OH radicals for reaction with NO2 to form HNO3), masking the true extent in the seasonal profile."

2) Specific Comments p6, l4, '...sampling rate of 0.2-0.4 l/min...' Please mention at this stage, or just below in the paragraph describing the aerosol collection system, what the particle size cut-off is for the DELTA sampler (mentioned later on p10, l3). It is important to know what the size spectrum of collected aerosols is, and that some (coarse) particles are not sampled, eg dust, large marine aerosols.

Author Response: The text below was inserted at the end of the paragraph 3 (section 2.2 Extended DELTA methodology for sampling acid gases and aerosol in AGANet). "A particle size cut-off of around 4.5 $\mu$m was estimated for the DELTA air inlet (Tang et al., 2015). The DELTA will therefore also sample fine mode aerosols in the PM2.5 fraction, as well as some of the coarse mode aerosols < PM4.5."

3) p7, l6-7, change of analytical labs from Harwell to CEH Lancaster in 2009: was there a transitional period of overlapping parallel measurements by the two labs, to make sure no bias was introduced in the long term time series by the change of laboratory?

Author Response: There was no transitional period of overlapping measurements, but measurements of replicated samples were compared between the two labs to ensure that there was no bias in chemical analysis prior to the lab. switch. CEH Lancaster laboratory is UKAS accredited, with experience of DELTA measurements prior to taking over the network measurements from Harwell lab.

4) p7, l26, '...flagging up occurrences of poorly coated denuders and/or sampling issues...' Another possibility is that concentrations are so large that the first denuder saturates and thus much is collected by the second denuder. This can happen for NH3 at agricultural sites after fertilisation; it is much less likely for acid gases due to lower concentrations, unless perhaps at some polluted urban stations?

Author Response: At high concentrations, saturation of the first denuder can indeed lead to lower gas capture efficiencies (breakthrough and capture on second denuder). The monitoring network sites are however located away from sources to monitor ambient concentrations. In 2015, the mean capture efficiencies for NH3, HNO3, SO2 and

HCl were 96%, 83%, 91% and 79 %, respectively.

5) p8, l15, the term 'bias' is used in relation to the 0.45 correction factor for HNO3, in the title of 2.6 and also other parts of the text. This is perhaps misleading as a bias suggests an offset, while the multiplicative correction applied acts on the span. Further, in the Tang et al 2015 report, the authors write that '... It is recommended that a correction factor of 0.45 be applied to the historic HNO3 measurements. The range of ratios was 0.44±0.15 (±2SD), i.e. 0.29-0.59, therefore it is reasonably likely that the value lies between 0.4 and 0.5. Therefore a correction factor of 0.45 should be applied...' It is quite clear that the percentage of non-HNO3 NOy compounds that is measured after extracting K2CO3-coated denuders depends on the relative abundances of these gases compared with HNO3, as well as their collection efficiencies on K2CO3 and their oxidation/reaction rate following adsorption. I would expect large seasonal changes, and large spatial/geographical variations, in these concentrations and the associated chemical processes, as reflected in the observed 0.29-0.59 range. Applying the same correction factor at all sites of the network, that range from remote to coastal to rural to sub-urban and urban, does not seem to be adequate. This is hinted at in the Eskdalemuir example of Fig. 3, where applying the large 0.45 multiplier makes the DELTA TIN values diverge from the EMEP filter pack measurements, ie at this rural background site the need for such a large correction is not warranted. The correction factor should account for the differences in pollution climates between sites, and also for changes over the 20-year period. Could an empirical correction be derived from chemical transport modelling (eg EMEP4UK), whereby the ratios of modelled HNO3 to NO2, HONO, PAN, etc, are used to construct a geographically- and temporally-varying index to drive the correction function? The HNO3 data reported in Tang et al (2015) for NaCl vs K2CO3 coating, with measurements made in contrasted situations (rural, urban, remote, see Table 1 in that report), may be used for calibrating such a function.

Author Response: The tile of section 2.6 has been changed to "HNO3 measurement artefacts and correction"

Regarding the empirical correction of the HNO3 data, please see author response to general comments on pages 1 - 2.

6) p8-11, section 2.7 Performance of the DELTA method: strictly speaking, this section describes measurement results from intercomaprison experiments or even long term datasets (Bush, Eskdalemuir) does not belong in Methods, and should be moved to the beginning of Section 3- Results

Author Response: Section 2.7 Performance of the DELTA method was included in the method section to separate this component from the main focus of presenting AGANet data in the results and discussion section. But agree: Moved to beginning of section 3 – Results and Discussion. 3.1 Performance of DELTA method 3.1.1 Comparison with daily annular denuder measurements 3.1.2 Comparisons with filter pack measurements: HNO3/NO3- and NH3/NH4+ 3.1.3 Comparisons with bubbler and filter pack measurements: SO2 and SO42-

7) p10, l12: Ca++ and Mg++ concentrations were near detection limits because they are mostly in the coarse fraction, with particle sizes near or above the DELTA cut-off. How much of the NaCl is similarly not collected by the DELTA system? Also, in relation to the DELTA v. ADS intercomparison, the loss of NO3, Na and Cl on the surface of the cyclone is put forward as an explanation for the lower ADS aerosols concentrations (compared with DELTA) (p10, l10), but why in that case is SO4= 23% larger in the ADS?

Author Response: NaCl is the main constituent of seasalt aerosol and size of seasalt aerosols ranges widely from ~0.05 to 10 $\mu$m in diameter, with their particle sizes also varying with humidity. The DELTA cut-off is estimated to be around 4.5 $\mu$m, so the DELTA will sample NaCl aerosols in the PM4.5 particle size region. Na+ measured on the DELTA are above detection limits, with concentrations ranging between 0.4 to 1.8 ug Na m-3 (annual mean in 2015). In the DELTA v ADS intercomparison of base cation measurements, the slope for Na was 3.0 (R2 = 0.24) and for Mg, the slope was 2.4

(R2 = 0.24), but a lot of scatter for Ca2+ as both ADS and DELTA Ca2+ data were at or below LOD. The DELTA therefore captures Mg2+ and Na+ well, but not Ca2+, which is what we find in the AGANet data.

The slope for SO42- in DELTA v ADS intercomparison is 0.69 (R2 = 0.89). The smaller SO42- signal on the DELTA may be due to incomplete capture of fine mode sulphate on the DELTA base coated cellulose filters. In the DELTA assessment report by Tang et al. (2015), up to 30 % of the total acid sulphate was measured on a $2\mu$m porosity PTFE membrane placed behind the K2CO3 coated filter to capture break-through. Since 2016, an additional PTFE membrane is added in front of the carbonate and acid coated cellulose filters.

A detailed assessment of the DELTA system against filter pack with a focus on SO2 and SO42- in 1999 by Hayman et al. (2006) had previosuly shown close agreement between the two methods, providing confidence in SO2 and SO42- measurements by the DELTA. Sulphur measurements provided by the DELTA replaced filter pack measurements in 1999.

Further work is ongoing to understand, assess and correct the bias in SO42- measurements in historic data. Since the DELTA method was unchanged for the assessment period in this paper, the bias in SO42- should not influence the interpretation of long-term trends in the data.

8) p13, l2: the highest HCl concentrations are in the SE and SW of England, but also in the Midlands

Author Response: Thank you. For an international audience, they may not know where the Midlands is. I propose to use "central England" instead of "Midlands",

Revised text below: "HCl in the atmosphere are mostly emitted from coal combustion and the highest concentrations of HCl are in the source areas in SE and SW of England, and also in central England (north of the Ratcliffe-on-Soar power station)...,..."

9) p13, l5: '...Further away from the coast and influence of marine aerosol, the smallest concentrations of Cl and Na+ are measured in the west of the country (Lough Navar in Northern Ireland...' Lough Navar is very near the coast (10km) and yet NaCl concentrations are very low (similar to concentrations in the Midlands), compared with all other western sites in the network (Fig. 5); what could be the reason?

Author Response: The Lough Navar site is actually approx. 40 km inland, close to the border between Northern Ireland and Republic of Ireland, within a forested area. The UK maps in the manuscript are all shown without Republic of Ireland, which may have given a false impression of Lough Navar being closer to the sea than it is in reality. Given its location inland, and the prevailing wind direction coming from the SW, it is far from the influence of seasalts.

10) p13, l5: For Cl- and Na+, '...largest concentrations at coastal sites in the south (Barcombe Mills) and west (Yarner Wood)...'; actually the highest concentrations are at Goonhilly at the SW tip of Cornwall.

Author Response: Thank you for spotting that. Barcombe Mills and Yarner Woods are two of the original 12 sites that were established in 1999 in AGANet. As coastal sites, Na+ and Cl- were always highest at these two sites up to the point when the new Goonhilly site in Cornwall was added as part of the network expansion in 2006. Na+ and Cl- are indeed higher at Goonhilly than Barcombe Mills and Yarner woods.

Text has been corrected accordingly. "The spatial distributions of Cl- and Na+ were similar, with largest concentrations at the coastal sites Goonhilly in SW England and Lerwick-Shetland in the Shetland Isles,. . .."

11) p13, l9-10, '...There is however no clear spatial pattern for Ca2+, with concentrations that are mostly at or below LOD...' For both Ca and Mg, which are mostly in the coarse fraction, it may be argued that the DELTA system does not allow a realistic assessment of the total concentration, because a large share of coarse particles are not collected. Please comment.

Author Response: Ca, Na and Mg are mainly in the 1 – 10 $\mu$m fraction in ambient aerosol. The size cut-off for the DELTA is around 4.5 $\mu$m, which means it will sample base cations in the PM4.5 fraction.

In the DELTA (PM4.5) v ADS (PM2.5) intercomparison of base cation measurements (see response to comment 7 earlier), the slope for Na was 3.0 (R2 = 0.24) and for Mg, the slope was 2.4 (R2 = 0.24), but no relationship was established for Ca2+ as both ADS and DELTA Ca2+ data were at or below LOD.

This suggests that the DELTA captures Mg2+ and Na+ in the PM4.5 fraction reasonably well. At all AGANet sites, Na and Mg measurements are above LOD, whereas Ca2+ are mostly at or below LOD. Aerosol filter blanks for Ca2+ are also much more variable than Na or Mg. Ca2+ is particularly problematic in chemical analysis as adsorption losses readily occur due to electrostatic interaction between Ca2+ and surfaces, especially plastic. To this end, aerosol sample extracts are acidified to minimise adsorption of Ca2+ to surfaces.

Sampling of Ca2+ is also likewise problematic as Ca2+ can potentially stick to inlets and surfaces. Tests conducted to assess adsorption losses of components to the connecting 6-mm diameter LDPE tube in the DELTA sampling train showed that measured concentrations of Ca2+ were within the noise of the LDPE tube blanks (i.e. clean LDPE tubes extracted with deionised water), adding to the uncertainty in Ca2+ measurements (see also response to next reviewer comment 12).

12) Further, our own tests with DELTA systems at INRA indicated very substantial losses for Mg and Ca in all the non-filter parts of the sampling train (particularly the 6-mm diameter LDPE elbow connecting the 2nd acid denuder to the first NH3 denuder, Fig.S1), which are therefore not measured on the filter. We analysed the loss fraction LDPE / (LDPE + den + filter) for all compounds; for NH4+ and NO3- this was less than 5%; for Cl- and Na+ this was 5-10%; for SO4= and Mg2+ this was 10-15%; while for Ca2+ this was 30-40%. Beyond the question of coarse aerosols that were not sampled

at all (did not enter the sampling train), there is the question of those coarse aerosols that 'did not make it' to the filter pack. Did the authors carry out similar tests, and could the results be shown in the supplement? It may be that the new straight design for the DELTA sampling train allowed a reduction of these losses? Please comment.

Author Response: Potential loss of particulate components to the connecting tube 6-mm diameter LDPE in the DELTA sampling train (Fig.S1) was investigated and reported in the DELTA assessment report by Tang et al. (2015). Our test results (extracted from Tang et al., 2015) are similar to the INRA findings outlined above: NO3-: loss to LDPE tube is negligible (2.4 $\pm$ 0.8 % (mean $\pm$ SD) across all sites for all available data). NO2-, SO42- and Cl-: losses to LDPE are small (< 6%). Base cations Na+ and Mg2+: losses to LDPE are slightly higher (<7%). Base cations Ca2+: there is a large degree of uncertainty in the calcium assessment, due to 1) variability of Ca2+ in the blank LDPE tube extracts and 2) very low Ca2+ on LDPE tubes from sites, that were similar to blank values and close to the detection limit (LOD = 0.05 mg/L Ca2+). Since January 2016, the new DELTA sample train configuration is linear, eliminating the use of the LDPE connecting tube.

13) P15, l1: This section 3.4 is mostly about sub-annual (seasonal) variations, so could be re-named 'Seasonal variations in acid gases and aerosols', as opposed to long term trends of Sections 3.5-3.6

Author Response: OK. Agree. Renamed "3.6 Seasonal variation in acid gases and aerosols"

14) p15, l9-10: '...In spring, the peak in HNO3 and NO3...' Fig.7 does not actually show any spring peak for HNO3; the late winter (Feb-Mar) concentrations are only marginally higher (but not significantly different accoring to the error bars) than the rest of the year? The opening sentence of the paragraph should read '...maximum in late winter and early spring...'

Author Response: Thank you. Text revised in "Section 3.6 Seasonal variation in acid

gases and aerosols" See below: "HNO3 is a secondary product of NOx, but NOx emissions are dominated by vehicular sources which are not expected to show large seasonal variations. Seasonal changes in chemistry and meteorology are therefore more likely to be a source of the observed variations in HNO3 and NO3- (Figure 8). HNO3 has a weak seasonal cycle with slightly higher concentrations in late winter and early spring that may be due to photochemical processes with elevated ozone in spring (AQEG 2009) leading to formation of HNO3 during this period (Pope et al., 2016). As discussed in section 3.3, a constant correction factor was applied to all HNO3 data, which does not take into account seasonal dependency. The concentrations in HNO3 may therefore be over-estimated in winter (less HNO3 formed from photochemical processes) and under-estimated in summer (larger HNO3 concentrations due to increased .OH radicals for reaction with NO2 to form HNO3), masking the true extent in the seasonal profile."

15) p15, l22, '...this contributes to the winter minimum in NH4NO3...' : the minimum NO3- actually occurs in July?

Author Response: Thank you. Text revised in "Section 3.6 Seasonal variation in acid gases and aerosols" See below: "Warm, dry conditions in summer promotes dissociation, increasing gas-phase HNO3 relative to particulate-phase NH4NO3, limiting peak NO3- aerosol concentrations (Figure 8). This process accounts for the minima in NO3- concentrations (Figure 7) and the highest ratio of HNO3 to NO3- seen in July (Figure 8). Cooler conditions in the spring than early autumn sees a larger fraction of the volatile NH4NO3 remaining in the aerosol phase. The peak in NO3- concentrations and the low HNO3:NO3- ratio in spring-time (Figure 8) is thus a combination of larger NO3- from reaction between higher concentrations of the precursor gases HNO3 and NH3, and partitioning to the aerosol phase. Import from long-range transboundary transport of particulate NO3- e.g. from continental Europe into the UK, as discussed in Vieno et al. (2014, 2016) adds to the elevated NO3- concentrations. In winter, low temperature and high humidity also shifts the equilibrium to formation of NH4NO3 from the gasphase HNO3 and NH3. Since NH3 concentrations are lowest in winter however, with less NH3 available for reaction, NH4NO3 concentrations are correspondingly smaller in winter than in spring or autumn."

16) p16, l9-13: how far should seasonal cycles for Mg and especially Ca be discussed, given the low collection efficiency (and thus high uncertainty) of filter data (see my comment above on aerosol size cut-off and losses in sampling train for these large aerosols)? Author Response: The discussion of the seasonal cycles on Mg2+ and Ca2+ are based on what the measurement shows. Mg2+ measurements were above LOD, with similar trends (spatial and seasonal) to Na+, so a discussion on seasonal cycle for Mg is warranted. In the case of Ca2+, uncertainties in interpretation of the Ca2+ data is discussed.

17) p17 and beyond, general comment on sections 3.5-3.6: a linear regression is fitted to all datasets from 1999 through 2015, but looking closely at the 15-yr time series for the 12 sites (eg Fig. 12-13), for HNO3, NO3-, SO4=, NH4+, NH3, it appears that concentrations were rather stable (with some interannual variability but no trend) in the period 2000-2007, and then only started declining after 2007. The only exception is SO2 with a continuous decline all the way. Fitting a linear trend is helpful to quantify an multi-annual rate of decrease (which is what you do), but is not an accurate representation of the time course of concentrations. Can you think of any plausible explanation for a change of course around the year 2007: change or implementation of pollution control policies? Decadal change in weather patterns? It might be useful to show (in the supplement)a summary of weather patterns for all sites of the network, the 15-yr time course of temperature, rainfall, wind speed etc.

Author Response: "Section 3.8 Assessment of trends in relation to UK emissions" has been revised and expanded to include a more thorough discussion of trends under new sub-headings. 3.8.1 Trends in HNO3 and NO3- vs NOx emissions 3.8.2 Trends in SO2 and SO42- vs SO2 emissions 3.8.3 Trends in HCl and Cl- vs HCl emissions 3.8.4 Trends in NH3 and NH4- vs NH3 emissions 3.8.5 Changes in UK chemical climate

Revised/expanded text: "The overall downward trends in HNO3 and NO3- are seen to be broadly consistent with the −49 % fall in estimated NOx emissions (NAEI, 2018) over the 16 year period between 2000 and 2015 (Figure 13). Reductions in combustion (power stations and industrial) and vehicular sources (fitting of catalytic converters), coupled to tighter emission regulations are major contributory factors to the decrease in UK NOx emissions. The rate of reduction however stagnated in the period 2009 and 2012 (improvement in emissions abatement offset by proportionate increase from diesel combustion and increase in vehicle numbers), followed by a 16 % decrease between 2012 and 2015 due to the closure of a number of coal-fired power stations. It is notable that the first 6 years (2000-2006) of HNO3 and NO3- annual data show substantial inter-annual variability and in particular are dominated by the large 2003 peak in concentrations (see sect. 3.6). Variability in the annual data thus highlights the sensitivity of the trend assessment to the selection of a reference start for the time series, since the annual mean concentrations of both HNO3 and NO3- in 2000 are in fact smaller than concentrations in the following 6 years. Re-analysis of the same annual data normalised against 2001 instead of 2000 takes the relative trend line for HNO3 and NO3- much closer to the relative trend line in NOx emissions. In the later period between 2006 and 2015, the relative trend lines in HNO3 and NO3- using mean data from 12 or 30 sites were not significantly different and emissions and concentrations trends followed each other closely."

Regarding the reviewers comment on the possibility of change in weather patterns to explain the apparent biphasic trend, the UK annual average temperature and rainfall (https://www.metoffice.gov.uk/climate/uk/summaries) show no overall trend in the 16 years of climate data between 1998 and 2015. 2010 was however an unusual year, with a lower than average mean annual temperature of 7.9 C due to an exceptional cold winter, with Dec 2010 recorded as the coldest for over 100 years (cf. 9.2 C average for 2000 to 2015) and lower than average rainfall of 950 mm (cf. 1180 mm average for 2000 to 2015). Graph of UK annual mean temperature and rainfall has been added to supplementary materials.

In terms of implementation of pollution control policies that could explain the change in course of the pollutant trend concentrations: In 2007, the designation of Nitrate Vulnerable Zones (NVZs) in the UK was introduced to strengthen the range of measures in the Nitrates Action Programme under the Nitrates Directive (91/676/EEC). NVZs are areas designated as being at risk from agricultural nitrate pollution and farms within NVZs must comply with the rules laid down on use of nitrogen fertiliser and storage of organic manure. Adoption of NAP by farms will also likely reduce emissions of NH3. NH3 data from the 12 sites in AGANet were stable from 2000-2010 and decreased between 2010 and 2012 with concentrations again stabilising after 2012. It could be surmised that there was more NH3 before 2007 to react with the acid gases and form / maintain higher concentrations of aerosols. But it has also to be borne in mind that the period between 2000 and 2007 was subject to a pollutant episode in 2003 and the data, as you also pointed out, is extremely variable. The apparent change in course of pollutant concentrations in NOǎ3- and SO42- is more likely due to influences of import from long range transboundary pollutant transport and meteorology.

18) p18, l6: '...The long-term time series in annually averaged concentrations of the gas and aerosol components are shown in Figure 12a and Figure 12b...': would it be possible to show, alongside the measured DELTA time series, the modelled NO/NO2 time series (from a CTM, eg EMEP4UK) for the same sites? In a way this would account for both NOx emission changes as well as climatic variability over the period.

Author Response: Dr Massimo Vieno (CEH) is currently working on a paper comparing EMEP4UK with measurement data from NAMN and AGANet.

NO2 concentrations is however measured at rural sites across the UK in the UKEAP NO2-net (NO2 diffusion tube network), some of which are co-located with the AGANet. The network average in annual mean NO2 concentrations showed a downward trend, decreasing from $\sim$8 $\mu$g NO2 m-3 in 2000 to $\sim$ 4 $\mu$g NO2 m-3 in 2015 (Conolly et al. 2016).

In terms of climatic variability from the UK, there is no apparent trend in the UK rainfall and temperature data (see earlier response to comment 17).

Additional text added in section 3.8.1 Trends in HNO3 and NO3- vs NOx emissions, end of last paragraph. A comparison of the network averaged NO2 concentrations with NOx emissions by Conolly et al (2016) showed matching decreasing trends between 2000 and 2015, with annual mean NO2 concentrations falling 2-fold to 4 $\mu$g NO2 m-3 in 2015 (Conolly et al. 2016). Although there is uncertainty in the corrected HNO3 data (see section3.3), the encouraging agreement between HNO3, NO2 concentrations and NOx emissions lends support to a linear response in HNO3 concentrations to reductions in NOx emissions.

19) p18, l12 '...The exceptions are Na+ and Cl- that have higher mean concentrations...' : Na+ is not shown in Fig.12.

Author Response: Mean concentration of Na+ from 12 and 30 sites are compared in Table 4. Table 4 inserted at the end of the sentence: "The exceptions are Na+ and Cl- that have higher mean concentrations from the 30 sites than the original 12 sites (Table 4)."

20) Figures 13, 14: use only one type of regression to simplify the figures (LR and MK give almost identical results)

Author Response: Figures 13 and 14 have been amalgamated into one single figure, with LR analyses taken out and moved to supplementary materials.

21) p20, l24-25 '...the reduction in SO2 emission and measured concentration is accompanied by a smaller negative trend in particulate SO4=...', and l27, '...The smaller decrease in particulate SO4= compared with its gaseous precursor, SO2, is similar to that observed at Eskdalemuir...'. Question: Is the smaller reduction rate in SO4= (compared with SO2) a reflection of the fact that increasingly in the UK, total sulphate includes a larger and larger fraction of marine sulphate, such that the decrease in an-

thropogenic SO4= (resulting from SO2 abatement) has a increasingly small effect on total sulphate? Is it possible to re-calculate the SO4= trend separately for coastal and inland (eg Midland/London) sites?

Author Response: Additional text added in "section 3.8.2 Trends in SO2 and SO42- vs SO2 emissions" to discuss sea salt SO42- (SS_SO4) – see below: "Sea salt SO42- (SS_SO4) aerosol, as discussed in section 3.5, makes up a significant fraction of the total SO42-. It is possible that the smaller reduction in particulate SO42-, compared with SO2, may be explained by an underlying increase in the relative proportion of SS_SO4 to total SO42-. To assess the contribution of SS_SO4 to the observed trends in total SO42-, SS_SO4 concentrations (estimated according to the empirical equation described in Sect. 3.5) and NSS_SO4- (= total SO42- – SS_SO4) are compared with the long-term trends in total SO42- in Figure 16. Overall, there is no trend in the long-term annual mean SS_SO4 data, with concentrations in range of 0.16 to 0.21 $\mu$g SO42-. Since SS_SO4 is derived from an empirical relationship with Na+ (sect.3.5), the long-term trend data for Na+ is also included in the analysis (Figure 16). Similar to SS_SO4, there is no overall trend in the Na+ data either, with small inter-annual variability and annual mean concentrations in the range of 0.65 − 0.85 $\mu$g Na+ m-3. SS_SO4 made up just 10% of the total SO42- in 2000, but by 2015, this had increased to just over 50% due to the decrease in NSS_SO4 over that time. MK analysis of the NSS_SO4 (Tables 4 and 5) showed decrease in concentrations of –78 % (2000-2015) and –62% (2006-2015), similar to that observed in SO2 (–81 %: 2000 –2015 and –60 %: 2006 – 2015), indicating a closer relationship between NSS_SO4 and SO2 than between total SO42- and SO2."

22) p21, l5-22: The argument about the NH3/SO2 ratio impacting the dry deposition velocities of SO2 and NH3 was developed in the 1980s and early 1990s, when SO2 concentrations were still very large in W. Europe. It is no longer sufficient to consider the NH3/SO2 ratio alone, since SO2 no longer massively dominates the acid load in W. European atmospheres. Instead, the ratio NH3/(2*SO2 + HNO3 + HCl) should

be computed to analyse long terme trends, as shown in Fowler et al. (Atmospheric Environment 43 (2009) 5193–5267, see Fig. 4.5). It is the combined effects of all acids and NH3 that determines the pH of cosystem/vegetation surfaces and hence their sink strength for water-soluble pollutants.

Author Response: Additional analysis of the change in molar ratios of NH3 to acid gases and molar ratios of NH4+ to NO3- and SO42- with time has been carried out – new figure added in manuscript: Figure 18: Long-term changes between 2000 and 2015 in (a) molar ratio of NH3 to acid gases (SO2, HNO3 and HCl) and (b) molar ratio of particulate NH4+ to acid aerosols (SO42- and NO3-) from measurements made at 12 sites in AGANet.

Revised/expanded text added, replacing text on p21, l5-22 "3.8.5 Changes in UK chemical climate" "Past studies have shown that the increasing ratio of NH3 to SOǍň2 in the atmosphere leads to enhanced dry deposition of SO2, accelerating the decrease in atmospheric SO2 concentrations than would be achieved by emissions reduction alone (Fowler et al., 2001, 2009; ROTAP 2012). The dry deposition of SO2 and NH3, by uptake of the gases in a liquid film on leave surfaces, is known to be enhanced when both gases are present in a process termed "co-deposition" (Fowler et al., 2001). Where ambient NH3 concentrations exceed that of SO2, there is enough NH3 to neutralize acidity in the liquid film and oxidise deposited SO2, and maintain large rates of deposition of SO2. With changes in the relative concentrations of acid gases in the UK and across Europe however, the deposition rates will increasingly be controlled by the NH3/combined acidity (sum of SO2, HNO3 and HCl) molar ratio (Fowler et al., 2009). To look at the UK situation, an analysis of the molar ratios of NH3 to acid gases is presented in Figure 18a. The molar ratio of NH3 to acid gases (sum of SO2, HNO3 and HCl) increased with time, from 1.9 in 2000 to 4.7 in 2015, confirming that NH3 is increasingly in molar excess over atmospheric acidity. The ratio of annual mean molar concentrations of NH3 (80 nmol m-3) to SO2 (29 nmol m-3 = 58 neq. m-3) was 2.7 in 2000, which increased in 2015 to 15 (annual mean concentrations of NH3 = 58 nmol

m-3 cf. SO2 = 4 nmol m-3 = 8 neq. m-3). Molar concentrations of HNO3 (4 nmol m-3) and HCl (6 nmol m-3) were comparable to SO2 in 2015, highlighting the increasing importance of HNO3 and HCl in contributing to atmospheric acidity. A larger decrease in SO2 (−81 %) than particulate sulphate (−69%) in the AGANet data (Table 4) would appear at first to suggest that the large NH3:SO2 ratio is contributing to a more rapid decrease in SO2 concentrations. However, when the seasalt fraction of SO42- is removed from the sulphate trend, the decrease in NSS_SO4 (−78%) is similar to SO2 (−81%) which would suggest that maximum deposition rates for SO2 may have been reached with the smaller SO2 concentrations since 2000."

23) p21, l30: '...The increase in ratio of HNO3:NO3- is similar to changes in upward trend in gas-aerosol partitioning between NH3 and NH4+ over time...': what do you call similar? For HNO3/NO3-, the ratio increases by _20%, while for NH3/NH4+, the ratio increases by 100% (according to Fig. 18) ?

Author Response: Apologies for the ambiguity in the sentence. I simply meant that both sets of data (HNO3:NO3- and NH3:NH4+) show an upward trend.

Text revised/expanded in section 3.8.5. Changes in UK chemical climate, paragraph 5. "A change to an NH4NO3 rich atmosphere and the potential for NH4NO3 to release NH3 and HNO3 in warm weather, together with the surfeit of NH3 also means that a larger fraction of the reduced and oxidised N is remaining in the gas phase as NH3 and HNO3. An increased partitioning to the gas phase may account for the larger decrease in particulate NH4+ (MK −62% between 2000-2015, n = 12) and NO3- (MK −52% between 2000-2015, n = 12) than NH3 (MK −30% between 2000-2015, n = 12) and HNO3 (MK −45 % between 2000-2015, n = 12) (Table 5) and the increase in gas to aerosol ratios (NH3:NH4+ and HNO3:NO3-) over the 16 year period (Figure 17). A higher concentration of the gas-phase HNO3 and NH3 may therefore be maintained in the atmosphere than expected on the basis of the emissions trends in NOx and NH3. Given the larger deposition velocities of NH3 and HNO3 compared to aerosols, more of the NH3 and HNO3 emitted will have the potential to deposit more locally with a

smaller footprint within the UK. "

24) p22, l11-12, '...a change in the particulate phase from (NH4)2SO4 to NH4NO3. This change is expected to increase residence times of NH3 and HNO3 in the atmosphere...' I am not convinced the shift from ammonium sulphate to ammonium nitrate should increase the residence time, since NH3 and HNO3 will deposit faster (higher deposition velocities) than either aerosol form?

Author Response: See revised/expanded text in "section 3.6. Seasonal variation in acid gases and aerosols" Specifically: "In contrast, the seasonal cycle for particulate NO3- is more distinct with a large peak in concentrations that occur every spring, together with a second smaller peak in autumn (Figure 8). NH3, the main neutralising gas in the atmosphere that reacts with HNO3 to form NH4NO3, has a correspondingly large peak in concentration in spring, a second smaller peak in autumn, but with elevated concentrations in summer and lowest in winter (Figure 8). Although particulate NO3- formation is dependent upon the availability of NH3 for reaction with HNO3, its' concentration is also governed by the equilibrium that exists between gaseous HNO3, NH3 and particulate NH4NO3, the latter of which is appreciably volatile at ambient temperatures (Stelson and Seinfeld, 1982). Partitioning between the gas and aerosol phase is therefore also a key driver for their atmospheric residence times and concentrations. HNO3 and NH3 that are not removed by deposition may react together in the atmosphere to form NH4NO3, when the concentration product [NH3].[HNO3] exceeds equilibrium values, with NH4NO3 serving as a potential reservoir for the gases. Since NH4NO3 is semi-volatile, any that is not dry or wet deposited can potentially dissociate to release NH3 and HNO3, effectively increasing their residence times in the atmosphere. The formation and dissociation in turn are strongly influenced by ambient temperature and humidity."

25) p22, l12 '...expected to increase residence times of NH3 and HNO3 in the atmosphere...' and p22, l15 '...NH3 and NOx emitted will deposit more locally with a smaller footprint...': these two statements appear to contradict each other?

Author Response: See response to comment 23 above and response to comment 26 after this.

  26) p22, l12: perhaps another way to analyse this trend is to calculate the (changing) linear regression slopes of NO3- vs NH4+ and SO4= vs NH4+, for each year of the 2000-2015 period (as in Fig. 6b), and examine how the two slopes change over time, as an indicator of the fraction of the total NH4+ that neutralizes NO3- and SOA= and its trend over time.

Author Response: New Figure 18: Long-term changes between 2000 and 2015 in (a) molar ratio of NH3 to acid gases (SO2, HNO3 and HCl) and (b) molar ratio of particulate NH4+ to acid aerosols (SO42- and NO3-) from measurements made at 12 sites in AGANet. Text revised/expanded in section 3.8.5. Changes in UK chemical climate (paragraph 3) Specifically: "To look at the UK situation, an analysis of the molar ratios of NH3 to acid gases is presented in Figure 18a. The molar ratio of NH3 to acid gases (sum of SO2, HNO3 and HCl) increased with time, from 1.9 in 2000 to 4.7 in 2015, confirming that NH3 is increasingly in molar excess over atmospheric acidity. The ratio of annual mean molar concentrations of NH3 (80 nmol m-3) to SO2 (29 nmol m-3) was 2.7 in 2000, which increased in 2015 to 15 (annual mean concentrations of NH3 = 58 nmol m-3 cf. SO2 = 4 nmol m-3). Molar concentrations of HNO3 (4 nmol m-3) and HCl (6 nmol m-3) were comparable to SO2 in 2015, highlighting the increasing importance of HNO3 and HCl in contributing to atmospheric acidity. A larger decrease in SO2 (−81 %) than particulate sulphate (−69%) in the AGANet data (Table 4) would appear at first to suggest that the large NH3:SO2 ratio is contributing to a more rapid decrease in SO2 concentrations. However, when the seasalt fraction of SO42- is removed from the sulphate trend, the decrease in NSS_SO4 (−78%) is similar to SO2 (−81%) which would suggest that maximum deposition rates for SO2 may have been reached with the smaller SO2 concentrations since 2000."

27) p24, l14 '...Higher concentrations of the NH3 and HNO3 in the atmosphere will deposit more locally...' But then, NH3 and HNO3 concentrations are actually decreasing;

they are not higher than before?

Author Response: Text revised/expanded in section 3.8.5. Changes in UK chemical climate (paragraph 5) "A change to an NH4NO3 rich atmosphere and the potential for NH4NO3 to release NH3 and HNO3 in warm weather, together with the surfeit of NH3 also means that a larger fraction of the reduced and oxidised N is remaining in the gas phase as NH3 and HNO3. The increased partitioning to the gas phase may account for the larger decrease in particulate NH4+ (MK −62% between 2000-2015, n=12) and NO3- (MK −52% between 2000-2015, n=12) than their gaseous precursors (NH3: MK −30% between 2000-2015, n=12 and HNO3: MK −45 % between 2000-2015, n=12) (Table 5) and the increase in ratios of NH3:NH4+ and HNO3:NO3- over the 16 year period (Figure 15). A higher concentration of the gas-phase nitrogen species (HNO3 and NH3) may therefore be maintained in the atmosphere than expected on the basis of the emissions trends in NOx and NH3. Given the larger deposition velocity of NH3 and HNO3 compared to particulate NH4+ and NO3-, more of the NH3 and HNO3 emitted will have the potential to deposit more locally with a smaller footprint within the UK."

28) Technical Corrections Units: different units are used. They should either be harmonized, or else each figure should state explicitly what the unit is, especially for the difference between element (N,S) based or molecule (HNO3, SO2) based. For example, mean HNO3 at the Bush site is reported as 0.55 $\mu$g m-3 in Fig.2 (average of 0.54 and 0.56 for samplers A and B), while the color code on the concentration map (Fig. 5) indicates a concentration in the range 0.15-0.25, from which I infer that Fig.2 is $\mu$g HN03 m-3, while Fig.5 is $\mu$g N m-3 ? Similarly, p12, l16, is the Cromwell site HNO3 concentration 1.3 $\mu$g HNO3 m-3, or 1.3 $\mu$g HNO3-N m-3? From Figure 5 I expect it is the latter (N, not HNO3 as written in the text). Further below, are the SO2 concentrations at Sutton Bonington given as $\mu$g SO2 m-3, or in fact $\mu$g SO2-S m-3 ? Given that the map in Fig. 5 gives numbers in $\mu$g N or $\mu$S per m3, it would be good to use the same units. Thus I would recommend to check carefully throughout the text in this

paragraph and in the whole paper and make the necessary text changes to eliminate the ambiguity in units.

Author Response: Thank you – checked and corrected.

29) p4, l26-27, delete '...that is also deployed at some CASTnet sites (Rumsey and Walker, 2016).' (already mentioned same page, l14) Author Response: OK – deleted.

30) p4, l32, suggest change 'temporal' to 'seasonal' Author Response: OK. Changed 'temporal' to 'seasonal'

31) p5, l9-12: this mostly repeats what was said in the introduction p4, l20-25 Author Response: OK – sentence below deleted. "Since 2009, the AGANet, together with the NAMN (monthly NH3 and NH4+), Precip-net (2-weekly wet deposition measurements) and NO2-net (4-weekly NO2 concentrations) were unified under the UKEAP network to provide long-term measurements of eutrophying and acidifying atmospheric pollutants (Conolly et al., 2016)."

32) p7, l20, please provide the equation for the calculation of the denuder capture efficiency Author Response: Calculation of denuder capture efficiency is described in "section 4 Calculation of air concentrations" "The denuder capture efficiency for each of the gas is calculated by comparing the concentrations of the individual gases in the denuder pairs"

Equation is now also provided: Denuder capture efficiency (% CE) =100 x (Denuder 1)/((Denuder 1+ Denuder 2))   33) p35: Figure 2 contains scatter plots and a statistical summary table for the Bush DELTA intercomparison (parallel sampling). It would be good to adapt the same or similar style of display for the other intercomparisons (scatter plots + stats table). Thus for the comparison with ADS (2.7.2), take Fig. S2 out of the supplement and stack it above the statistics given in Table 2. Similarly for the intercomparisons of DELTA vs EMEP TIA/TIN (add statistical table), as is already also done for DELTA vs Bubble/FP Eskdalemuir (Fig 4).

Author Response:

Thank you for suggestions: Fig S2 and Table 2 combined into Figure 2 Figure 3 (DELTA vs EMEP TIA/TIN), summary stats table added.

There are now however quite a large number of figures.

34) p11, sections 2.8 and 3.6: throughout the time series trend analysis, both linear regressions and non-parametric MK tests are used, but as far as I can see, there is essentially no difference between the slopes for any of the pollutant time series. To improve readability and reduce unnecessary redundant information, I would suggest to stick to just one of the methods; it would suffice to say in the methods that both regressions were used and no significant differences were found, and thus henceforth only one regression is displayed.

Author Response: Thank you for the suggestion. Figures 13 and 14: graphs with both linear regression and MK analyses moved to supplementary materials. Replaced by a single Figure 14 showing results of MK analysis only for both time series.

Tables 5 and 6: summary tables comparing LR and MK moved to supplementary section. Replaced by a single Table 4 showing results of MK analysis only for both time series.

Additional text included at end of <section 2.8 Time series trend analyses "...but since there was no difference between either tests, MK results only are presented and discussed in the paper. A comparison of trend analyses from both approaches is however provided in supplementary materials (Figures S7, S8 and Tables S4 - S6). "

35) p12, l22, '...A peak MONTHLY concentration of...' Author Response: Thank you – corrected

36) p12,l28, '...expected to be more SPATIALLY homogeneous...' Author Response: Thank you – corrected

37) p15, l19 '...in summer promotes AEROSOL dissociation...' Author Response: Thank you – corrected

38) p17, l12 change to '...are available SINCE 1989...' Author Response: Thank you – corrected

39) p18, l31, '...To QUANTIFY changes...' Author Response: Thank you – corrected (paragraph moved to section 2.7 Time series trend analyses)

40) p18, l32, the unit for the annual trend is $\mu$g HNO3-N m-3 y-1 Author Response: The unit is for annual trend is $\mu$g HNO3 m-3 y-1 Units used in trend analysis are on a molecule basis

41) p19, l3: '...The LR % annual trends for each time series...' Delete 'annual', since the % reduction are not expressed per year, but over the whole period ? Note that if the concentration reduction were a constant percentage every year, say -10% per year, then the overall time course over 15 years would not look linear, but exponential: if yr1=100, then yr2=90, yr3=81, yr4=72.9, yr5=65.6, ...yr15=20.6

Author Response: Thank you - text corrected: "The LR and MK % change in annual mean concentrations for the two time series are estimated from the slope and intercept. ..."

42) p19, l6: same as above, delte 'annual' Author Response: Thank you - equation corrected: % change =100 . ([Yi-Yo))/Yo

43) p21, l17-18 '...The dry deposition... IS known to be enhanced...' Author Response: Thank you - corrected

44) p23, l9, delete 'from coal combustion' Author Response: Thank you - deleted

45) p24, l1, '... modest reductionS in HNO3...' (plural) Author Response: Thank you - corrected

46) p24, l12, '...smaller THAN emission trends...' Author Response: Thank you – corrected

47) All figures: when the units displayed on axes or legends are given in $\mu$g m-3, please specify whether this is on an element basis (NH3-N, HNO3-N, SO2-S) or molecule basis (NH3, HNO3, SO2) Author Response: Figure 10a, Y-axis changed to Oxidised N ($\mu$g N m-3) Figure 10b, Y-axis changed to Reduced N ($\mu$g N m-3) Figure 11, Y-axes changed to SO2 ($\mu$g S m-3) and SO42- ($\mu$g S m-3) To show more clearly that the units are on an element basis (NH3-N, HNO3-N, SO2-S, etc.)

All other figures are on a molecule basis (NH3, HNO3, SO2, etc)– axis and legends should be correct.

48) Figure 8: "...Average annual cycles in the ratios of gas:aerosol component concentrations ($\mu$g m-3)...' The unit for the ratio is not $\mu$g m-3, it must be dimensionless, or mol mol-1?

Author Response: The Y axis label on the graphs are dimensionless. In the figure caption, ($\mu$g m-3) is the unit of gas and aerosol concentrations that are compared. As this is causing confusion, the caption in Figure 8 (now Figure 9, because Suppl. Figure S2 added as Figure 2) has been revised to:

"Figure 9: Average annual cycles in the ratios of gas:aerosol component concentrations. HNO3, SO2, HCl and aerosol NO3-, SO42-, Cl- data (annual mean, $\mu$g m-3) are from the UK Acid Gases and Aerosol Monitoring Network (AGANet). NH3 and NH4+ data (annual mean, $\mu$g m-3) are from the UK National Ammonia Monitoring Network (NAMN, Tang et al., 2018) measured at the same time. Each data point in the graphs represents the mean $\pm$ SD of monthly measurements of 12 sites operational in the network over the period 2000 to 2015.

49) Figures 13-14: keep only one of the two trend lines (LR or MK); and delete Fig.14 but add the n=30 datapoints to Fig. 13 as a different symbol shape or color Author Response: Figures 13 – 14 replaced with a single figure as suggested by reviewer

above.

50) Figure 18: the left-hand side panels show the same data as Figs. 13-14 and should therefore not be repeated here. Author Response: The left hand panels provides a direct comparison of the concentrations and trends of each of the gas and aerosol pairs (HNO3/ NO3-, SO2/SO42- etc.). The author agrees with the reviewer that the left hand panels shows the same data as Figures 13 and 14 and they have been removed.

---

## Author Comment (AC2) · 8 Oct 2018

The authors thank reviewer 2 for his constructive comments and for taking the time to look at all the details described in the manuscript. We have carefully considered all comments. Please refer to the specific responses.

General comments 1) Currently the manuscript is primarily focused on documenting trends and events and with a smaller focus on the changes in atmospheric composition and pollutant fate due to changes in emissions. This manuscript would benefit from a bit more focus. I suggest focusing more on the trends and how they relate to emission changes and less on specific events captured in the data.

Author Response: Section 3.8 Assessment of trends in relation to UK emissions" has
been revised and expanded to include a more thorough discussion of trends under new sub-headings. 3.8.1 Trends in HNO3 and NO3- vs NOx emissions 3.8.2 Trends in SO2 and SO42- vs SO2 emissions 3.8.3 Trends in HCl and Cl- vs HCl emissions 3.8.4 Trends in NH3 and NH4- vs NH3 emissions 3.8.5 Changes in UK chemical climate Discussion on specific events captured in the data have not been revised/truncated as they are important for interpreting anomalies in the trends.

3.8 Assessment of trends in relation to UK emissions. The long-term time series in annually averaged concentrations of the gas and aerosol components are shown in Figure 13a and Figure 13b, respectively. Annually averaged data from the original 12 sites for the period 2000 – 2015 (1999 data excluded since AGANet started in September 1999) and from the full network (30 sites) for the period 2006 – 2015 are plotted alongside each other for comparison. From 2006 – 2015, the decreasing trends for all gas and aerosol components from the expanded 30 sites are seen to be similar to those from the original 12 sites. The annual mean concentrations in gas and aerosol components derived from the expanded 30 sites (2006 – 2015), or from the original 12 sites over the same period are also in general comparable (Table 3). The exceptions are Na+ and Cl- that have higher mean concentrations from the 30 sites than the original 12 sites, due to the addition of two coastal sites (Shetland and Rum), with larger contribution from sea salt. Larger HNO3 concentrations are due to two urban sites, London and Edinburgh (higher NOx emissions from vehicular traffic). The addition of three sites in high NH3 emission (agricultural) areas (Rosemaund in England, Narberth in Wales and Hillsborough in Northern Ireland) also elevated measured annual mean NH3 concentrations. The comparisons here thus illustrates very clearly the need to consider the effect of site changes in a national network and the importance of maintaining consistency and site continuity for assessing long-term trends.

In the gas phase, SO2 decreased 7-fold from an annual mean concentration of 1.9 $\mu$g SO2 m-3 in 2000 to 0.25 $\mu$g SO2 m-3 in 2015 (n = 12), compared with more modest reductions in HNO3 (from 0.35 to 0.21 $\mu$g HNO3 m-3), NH3 (from 1.4 to 1.0 $\mu$g NH3

m-3) and HCl (from 0.31 to 0.20 $\mu$g HCl m-3) over the same period (Figure 13a). Particulate SO42-, NO3- and NH4+ also decreased in concentrations with time, but unlike their gas phase precursors, the trends of these aerosol components track each other closely, differing only in the magnitude of concentrations (Figure 13b), illustrating very clearly the close coupling between these components. On the other hand, the absence of a trend in the particulate Cl- is likely to reflect the sea salt origin of Cl- which is not expected to vary over time.

Important changes in the chemical climate is captured by the parallel monitoring of acid gases and aerosols in AGANet and of NH3, NH4+ in NAMN. It is clear from the long-term data that there is substantial intra- (Figure 10) and inter-annual variability in the annual mean concentrations of both the gas and aerosol phases (Figure 13), in particular the spike in concentrations in 2003 (see Sect. 3.6) that buckles the trend. An interpretation of the direct relationship between emissions and concentrations in the atmosphere is therefore not straight forward, as the concentrations are also influenced by other factors such as variations in meteorological conditions and long-range transboundary import into the UK.

In Figure 14, the relative trends in UK NOx, SO2, HCl and NH3 emissions (NAEI, 2018) are compared with the annually averaged gas and particulate concentrations measured in the AGANet and NAMN for (i) original 12 sites for the 16 year period from 2000 to 2015, (ii) original 12 sites for the 15 year period from 2001 to 2015, and (iii) expanded 30 sites and also original 12 sites for the 10 year period from 2006 to 2015. All data were normalised to zero for the start years in each of the comparison. Annual trends (e.g. $\mu$g HNO3 m-3 y-1) and changes in measured concentrations over time (% median change) estimated from MK tests are summarised in Figure 15 and Table 4.

Since there was a change in the number of sites during the operation of the AGANet, statistical trend analyses for HNO3, SO2, HCl and particulate NO3-, SO42-, Cl- were performed on annually averaged mean concentrations from two time series: the original 12 AGANet sites for the 16 year period from 2000 to 2015, and the expanded 30

AGANet sites for the 10 year period from 2006 to 2015 (Figure 15, Table 4). NH3 and NH4+ concentrations from the NAMN that were measured at the same time at the AGANet sites were also included for comparison (Figure 15, Table 4) and to aid interpretation of the acid gas and aerosol data. This approach avoids introducing bias as a result of changes in the sites and ensures site continuity for the long-term trend assessment.

The long-term trends in the gas and aerosol components, based on both LR and MK statistical analysis of monthly mean measurement data, are also shown for comparison in Figure S4 (mean monthly data of 12 sites for period 2000-2015) and Figure S5 (mean monthly data of 12 sites for period 2006-2015). Results of the trend analysis on monthly data (Tables S4, S5) were similar to trend analysis results of the annual data (Table 4). While not discussed further here, since assessment of long-term trends in this paper focusses on trends in annual mean concentrations for comparison with trends in estimated annual emissions, the monthly plots serves to illustrate the large intra-annual variability of concentrations in gases and aerosols. Changes in the different gas and aerosol concentrations in relation to emission trends, and their interactions in a changing chemical climate are discussed further in the next sections.

3.8.1 Trends in HNO3 and NO3- vs NOx emissions. The overall downward trends in HNO3 and NO3- are seen to be broadly consistent with the $-49$ % fall in estimated NOx emissions (NAEI, 2018) over the 16 year period between 2000 and 2015 (Figure 14). Reductions in combustion (power stations and industrial) and vehicular sources (fitting of catalytic converters), coupled to tighter emission regulations are major contributory factors to the decrease in UK NOx emissions. The rate of reduction however stagnated in the period 2009 and 2012 (improvement in emissions abatement offset by proportionate increase from diesel combustion and increase in vehicle numbers), followed by a 16 % decrease between 2012 and 2015 due to the closure of a number of coal-fired power stations.

It is notable that the first 6 years (2000-2006) of HNO3 and NO3- annual data show

substantial inter-annual variability and in particular is dominated by the large 2003 peak in concentrations (see sect. 3.6). This highlights the sensitivity of the trend assessment to the selection of a reference start for the time series, since the annual mean concentrations of both HNO3 and NO3- in 2000 are in fact smaller than concentrations in the following 6 years. Re-analysis of the same annual data normalised against 2001 instead of 2000 takes the relative trend lines for HNO3 and NO3- much closer to the relative trend line in NOx emissions. In the later period between 2006 and 2015, the relative trend lines in HNO3 and NO3- derived from the mean of either 12 or 30 sites were not significantly different, and the relative trend lines in emission and concentrations followed each other closely.

The reductions in annual HNO3 concentrations are statistically significant for both periods (Figure 15; Table 4). The MK % median change in annual mean HNO3 was $-45$ % (2000 $-$ 2015, n = 12) and $-36$ % (2006 $-$ 2015, n = 30), consistent with the $-49$ % and $-40$ % fall in estimated NOx emissions over the corresponding periods (Table 5). The decrease in HNO3 is accompanied by a larger decrease in particulate NO3- (2000 - 2015: MK = $-52$ % (n = 12), 2006 $-$ 2015: MK = $-43$ % (n = 30)) (Table 4). Since HNO3 is one of the major oxidation products of NOx, through reaction with OH. or heterogeneous conversion of N2O5, it provides an important measure of the fraction of NOx emissions that is oxidised and signals any long-term changes in the atmospheric processing timescales of NOx over the UK. NO2 is measured at 24 rural sites across the UK in the UKEAP NO2-net (Conolly et al., 2016), with 11 sites co-located with the AGANet. A comparison of the network averaged NO2 concentrations with NOx emissions by Conolly et al (2016) showed matching decreasing trends between 2000 and 2015, with annual mean NO2 concentrations falling 2-fold to 4 $\mu$g NO2 m-3 in 2015. Despite the uncertainty in corrected HNO3 data (see section 3.3), the encouraging agreement between trends in HNO3 and NO2 concentrations and NOx emissions lends support to a linear response in HNO3 concentrations to reductions in NOx emissions.

3.8.2 Trends in SO2 and SO42- vs SO2 emissions. Unlike NOx, there has been a more substantial decline in SO2, both in emissions and measured concentrations (Figure 14, Table 5). Between 2000 and 2009, SO2 emissions fell substantially by 66 % from 1286 to 432 kt SO2. The reduction reflects mitigation measures introduced in the 1980s (fitting of flue gas desulphurisation to coal fired power stations) to control S pollution, reductions in energy production and manufacturing and the switch from coal to gas at the same time. Similar to the trends in NOx emission, the decreasing trend in SO2 emissions plateaued between 2009 and 2012 and then decreased again by a further 45% between 2012 and 2015 following the closure of a number of coal-fired power stations, as well as conversion of some coal-fired stations to burn biomass.

Over the same period, the network annual mean concentration decreased from 1.9 $\mu$g SO2 m-3 in 2000 to 0.25 $\mu$g SO2 m-3 in 2015 (mean of 12 sites), continuing the long-term decline in SO2 concentrations observed at the background Eskdalemuir site (Sect. 3.5) and across the UK (ROTAP 2012). The relative trends in SO2 emissions and concentrations tracked each other closely for all the time periods considered and it is clear that these decreases are highly correlated (Figure 14). In the case of particulate SO42- however, there is an apparent "gap" between emissions and concentrations in the trend normalised against the year 2000. Like NO3-, re-analysis of the same annual data normalised against 2001 instead of 2000 takes the relative trend line for SO42- closer to the trend lines in both SO2 emissions and concentrations (Figure 14), thus again highlighting the potential bias in the use of a measured value at a specific time point in trend assessments when there is substantial inter-annual variability in the data.

From the MK trend analysis, the decrease in annual mean SO2 concentrations of –81 % (2000 – 2015, n = 12) , and –60 % (2006 – 2015, n = 30) (Figure 15; Table 4) are consistent with the substantial reduction of –80 % and –64 % in SO2 emissions across the two overlapping periods, respectively (Table 5). The decrease in SO2 is also twice as large as HNO3 over the same period (Table 5), illustrating the greater success in mitigating sulphur than nitrogen and the increasing dominance of N components in the

atmosphere with larger decline in SO2 than NOx.

At the same time, the reduction in SO2 emission and measured concentration is accompanied by a smaller negative trend in particulate SO42- (2000-2015: –69 % MK; 2006-2015: –54 % MK) (Figure 15; Table 4), with concentrations falling 3-fold from an annual mean of 1.2 $\mu$g SO42- m-3 in 2000 to 0.42 $\mu$g SO42- m-3 in 2015. The smaller decrease in particulate SO42- compared with SO2, is similar to that observed at Eskdalemuir (Sect. 3.1.3). A similar picture is also seen in Europe, where atmospheric concentrations of gas phase SO2 decreased by about 92 % compared with a smaller reduction of 65 % in particulate SO42- in response to sulphur emissions abatement over the 1990-2012 period in the EMEP region (EMEP 2016).

Sea salt SO42- (SS_SO4) aerosol, as discussed in section 3.5, makes up a significant fraction of the total SO42-. It is possible that the smaller reduction in particulate SO42-, compared with SO2, may be explained by an underlying increase in the relative proportion of SS_SO4 to total SO42-. To assess the contribution of SS_SO4 to the observed trends in total SO42-, SS_SO4 concentrations (estimated according to the empirical equation described in Sect. 3.5) and NSS_SO4- (= total SO42- – SS_SO4) are compared with the long-term trends in total SO42- in Figure 16. Overall, there is no trend in the long-term annual mean SS_SO4 data, with concentrations in range of 0.16 to 0.21 $\mu$g SO42-. Since SS_SO4 is derived from an empirical relationship with Na+ (sect. 3.5), the long-term trend data for Na+ is also included in the analysis (Figure 16). Similar to SS_SO4, there is no overall trend in the Na+ data either, with small inter-annual variability and annual mean concentrations in the range of 0.65 – 0.85 $\mu$g Na+ m-3. SS_SO4 made up just 10% of the total SO42- in 2000, but by 2015, this had increased to just over 50% due to the decrease in NSS_SO4 over that time. MK analysis of the NSS_SO4 (Table 4; Table 5) showed decrease in concentrations of –78 % (2000-2015) and –62% (2006-2015), similar to that observed in SO2 (–81 %: 2000 –2015 and –60 %: 2006 – 2015), indicating a closer relationship between NSS_SO4 and SO2 than between total SO42- and SO2.

3.8.3 Trends in HCl and Cl- vs HCl emissions. HCl emissions in the UK also decreased substantially by 89 % between 2000 and 2015, from 82 kt to 9 kt in 2015 (NAEI 2018), contrasting with a smaller, but non-significant decreasing trend in HCl concentrations (Figure 14; Figure 15; Table 5). The annual mean monitored concentrations in HCl over this period decreased from 0.30 $\mu$g HCl m-3 in 2000 to 0.19 $\mu$g HCl m-3 in 2015. Most of the reduction in HCl emissions occurred before 2006 ($-79\%$, from 82kt in 2000 to 17kt in 2006), with emissions plateauing since 2006. A corresponding decrease is not seen in the HCl measurement data, where concentrations remained fairly stable at between 0.31 $\mu$g m-3 HCl in 2000 to 0.33 $\mu$g m-3 HCl in 2006. Since 2006 however, the relative change in HCl emissions is closely tracked by changes in concentrations of both the annual mean data from the original 12 sites and from the expanded 30 sites in the AGANet, with the small peak in HCl emissions in 2013 also captured in the annual mean data. This part of the time series therefore clearly shows a direct relationship between emissions and concentrations (Figure 14).

So why is the most significant fall in HCl emissions between 2000 and 2006 not captured by the network? HCl are mainly released as point sources. Coal burning, particularly from coal-fired power stations, is responsible for the majority of UK emissions: 92 % in 1990 and 76 % in 2015 and reductions in HCl emissions in the UK inventory is largely as a result of declining coal use and the installation of emissions abatement measures at coal-fired power stations (implemented since 1993) aimed at reducing S that also coincidentally reduced HCl emissions. It may be that a network of only 12 sites in the early periods failed to capture peak emissions and changes in source areas. While there is an indicative, but non-significant decreasing trend in HCl (2000-2015: MK = $-28$ %, 2006–2015: MK = $-24$ %), no detectable trend in particulate Cl- can be seen (Table 4). Since Cl- is mainly associated with Na+ (seasalt) in the AGANet measurements (Sect. 3.5), the absence of a trend in Cl- and Na+ (Sect. 3.8.2, Figure 16) provides evidence of a constant background in seasalt in the UK atmosphere.

3.8.4 Trends in NH3 and NH4+- vs NH3 emissions. In comparison to the acid gases,

there is a more modest decrease of −10 % in NH3 emissions, from 254 kt NH3 in 2000 to 231 kt NH3 in 2015 (NAEI, 2018). This is smaller than the larger 30 % decrease seen in the annually averaged NH3 concentrations at the 12 AGANet sites (2000-2015: −30 % MK) (Figure 14, Figure 15, Table 4) over the same period (Table 5). A recent assessment by Tang et al. (2018) showed that NH3 trends are highly dependent on site selection and categorisation of sites in the analysis. A more comprehensive analysis of a larger number of sites shows smaller reductions over time, whereas a significant decreasing trend in NH3 concentrations was observed in the grouped analysis of sites in areas classed as dominated by pig and poultry emissions, against an upward (non-significant) trend for sites in cattle-dominated areas. Therefore there is a large degree of uncertainty in interpreting the trends in NH3 concentrations from a subset of just 12 sites, since NH3 emissions are dominated by agricultural emissions (> 80%) that vary hugely on a local to regional scale across the UK.

At the same time, there is a larger decrease in particulate NH4+ concentrations (−62 % MK), contrasting with the smaller decrease in NH3 concentrations (−30 % MK) over the period 2010–2015 (Table 4), with the NH3:NH4+ ratio increasing with time (Figure 17). This provides evidence for a shift in partitioning from the particulate phase NH4+ to the gaseous phase NH3 in the UK data, discussed in Tang et al. (2018). The change in partitioning from particulate NH4+ to gaseous NH3 is also occurring in other parts of Europe, where decreases in NH3 concentrations have been smaller than emission trends would suggest, due to large decreases in SO2 emissions (Bleeker et al., 2009; Horvath et al., 2009).

3.8.5 Changes in UK chemical climate. Atmospheric SO2 concentrations in the UK has declined to very low levels over the 16 years of measurements in AGANet, with annual mean concentrations in 2015 (0.25 $\mu$g SO2 m-3, n = 12) approaching that of the other acid gases HNO3 (0.21 $\mu$g HNO3 m-3, n = 12) and HCl (0.20 $\mu$g HCl m-3, n = 12). NH3 measured at the same time at the AGANet sites also decreased, but to a smaller extent, to a mean concentration of 1.0 $\mu$g NH3 m-3 (n = 12) in 2015. The

changes in measured concentrations of SO2, HNO3, HCl and NH3 are consistent with the estimated decrease in emissions of SO2, NOx, HCl and NH3 since 2000. SO2 is therefore no longer the dominant acid gas, with HNO3 and HCl together contributing a larger fraction of the total acidity in the UK atmosphere.

Past studies have shown that the increasing ratio of NH3 to SOǍ2 in the atmosphere leads to enhanced dry deposition of SO2, accelerating the decrease in atmospheric SO2 concentrations than would be achieved by emissions reduction alone (Fowler et al., 2001, 2009; ROTAP 2012). The dry deposition of SO2 and NH3, by uptake of the gases in a liquid film on leave surfaces, is known to be enhanced when both gases are present in a process termed "co-deposition" (Fowler et al., 2001). Where ambient NH3 concentrations exceed that of SO2, there is enough NH3 to neutralize acidity in the liquid film and oxidise deposited SO2, and maintain large rates of deposition of SO2. With changes in the relative concentrations of acid gases in the UK and across Europe however, the deposition rates will increasingly be controlled by the NH3/combined acidity (sum of SO2, HNO3 and HCl) molar ratio, rather than based on SO2 alone (Fowler et al., 2009).

To look at the UK situation, an analysis of the molar ratios of NH3 to acid gases is presented in Figure 18a. The molar ratio of NH3 to acid gases (sum of SO2, HNO3 and HCl) increased with time, from 1.9 in 2000 to 4.7 in 2015, confirming that NH3 is increasingly in molar excess over atmospheric acidity. The ratio of annual mean molar concentrations of NH3 (80 nmol m-3) to SO2 (29 nmol m-3) was 2.7 in 2000, which increased in 2015 to 15 (annual mean concentrations of NH3 = 58 nmol m-3 cf. SO2 = 4 nmol m-3). Molar concentrations of HNO3 (4 nmol m-3) and HCl (6 nmol m-3) were comparable to SO2 in 2015, highlighting the increasing importance of HNO3 and HCl in contributing to atmospheric acidity. A larger decrease in SO2 (−81 %) than particulate sulphate (−69%) in the AGANet data (Table 4) would appear at first to suggest that the large NH3:SO2 ratio is contributing to a more rapid decrease in SO2 concentrations. However, when the seasalt fraction of SO42- is removed from

the sulphate trend, the decrease in NSS_SO4 ($-78\%$) is similar to SO2 ($-81\%$) which would suggest that maximum deposition rates for SO2 may have been reached with the smaller SO2 concentrations since 2000.

At the same time, reduction in emissions of the precursor gases have also led to a lower formation of particulate phase NH4+, NO3- and SO42- in the atmosphere and changes in atmospheric composition. Since the affinity of H2SO4 (oxidation product of SO2) for NH3 is much larger than that of HNO3 and HCl, available NH3 is first taken up by H2SO4 to form ammonium sulphate compounds (NH4HSO4 and (NH4)2SO4), with any excess NH3 then available to react with HNO3 and HCl to form NH4NO3 and NH4Cl that are volatile. Analysis of the different particulate components in sect. 3.5 showed that the ammonium aerosols are mainly made up of (NH4)2SO4 and NH4NO3. With the large reduction in SO2, more NH3 is available to react with HNO3 to form NH4NO3 and concentrations of NH4+ and NO3- are now observed to be in molar excess over SO42-, providing evidence of a change in the particulate phase from (NH4)2SO4 to NH4NO3 (Figure 18b). A change to an NH4NO3 rich atmosphere and the potential for NH4NO3 to release NH3 and HNO3 in warm weather, together with the surfeit of NH3 also means that a larger fraction of the reduced and oxidised N is remaining in the gas phase as NH3 and HNO3. An increased partitioning to the gas phase may account for the larger decrease in particulate NH4+ (MK $-62\%$ between 2000-2015, n = 12) and NO3- (MK $-52\%$ between 2000-2015, n = 12) than NH3 (MK $-30\%$ between 2000-2015, n = 12) and HNO3 (MK $-45\%$ between 2000-2015, n = 12) (Table 5) and the increase in gas to aerosol ratios (NH3:NH4+ and HNO3:NO3-) over the 16 year period (Figure 17). A higher concentration of the gas-phase HNO3 and NH3 may therefore be maintained in the atmosphere than expected on the basis of the emissions trends in NOx and NH3. Given the larger deposition velocities of NH3 and HNO3 compared to aerosols, more of the NH3 and HNO3 emitted will have the potential to deposit more locally with a smaller footprint within the UK.

Currently, the critical loads of acidity (sulphur and nitrogen) are exceeded by 44 % of

the area of sensitive habitats in the UK (based on mean deposition data for 2012-2014), whereas the figure for exceedance of eutrophication (nutrient nitrogen) is even larger, at 62 % (based on deposition data for 2012 – 2014) (Hall & Smith, 2016). Air quality policies have been very successful in abating SO2 emissions (–80 %: 2000 – 2015) and moderately successful with NOx emissions (–58 %: 2000 – 2015), with both on course to meet the emission reduction targets set out under the 2012 Gothenburg protocol and 2016 NECD. NH3 emissions however has decreased by only 10 % over the same period, with an increasing trend since 2013 and it is likely that abatement measures may be required to meet emission reduction targets. In recognising In recognising the need to tackle the increasing NH3 emissions, the Code of Good Agricultural Practice (COGAP) was published under the UK government's Clean Air Strategy (launched in July 2018) as a step towards reducing NH3 emissions from agriculture.

Based on the current emission trends and evidence from AGANet and NAMN long-term measurements, atmospheric N deposition from oxidised N (NOx, HNO3 and NO3-) and from reduced N (NH3, NH4+) are likely to continue to exceed critical loads of N deposition over large areas of sensitive habitats, with implications for UK's commitment to maintain or restore natural habitats (e.g. Natura 2000 sites; Hallsworth et al., 2010) to a favourable conservation status under the EU Habitats Directive (Council Directive 92/43/EEC). The changes are also relevant for human health effects assessments, since NH4NO3 and (NH4)2SO4 are mainly in the fine mode and constitute a significant fraction of PM2.5 that are associated with acute and chronic human health problems. The change in partitioning from (NH4)2SO4 to NH4NO3, coupled to import of NH4NO3 from long-range transport (driven by emissions of NH3 and NOx from outside the UK) poses policy challenges in protection of human health from effects of air pollution, particularly in urban areas where concentrations of the PM2.5 precursor gases NOx, SO2 and NH3 are higher.

2) The discussion of the trends of NH3 and HNO3 are sometimes a bit difficult to follow as the change in aerosol composition and loading over the time frame of the measurements impacts the gas phase concentrations. Consider discussing these trends as total nitrate (gaseous HNO3 + aerosol NO3) and NHx (gaseous NH3 + aerosol NH4).

Author Response: Interactions and partitioning between the gas phase (SO2, HNO3, NH3) and aerosol phase (SO42-, NO3-, NH4+) are important drivers for concentrations and trends in the respective components. Discussion of the gas phase and particulate phase atmospheric components for oxidised and reduced nitrogen, rather than total inorganic nitrate (TIN, sum of gaseous HNO3 + aerosol NO3-) and total inorganic NHx (TIA, sum of gaseous NH3 + aerosol NH4+) allows a clearer understanding of the processes occurring in the atmosphere, which drive trends and environmental effects. TIN and TIA is only considered in the manuscript for comparing DELTA with EMEP filter pack measurements at Eskdalemuir.

The expanded "Section 3.8 Assessment of trends in relation to UK emissions" (see response to your comment 1 above) should hopefully provide a clearer discussion on the change in gas and aerosol composition and their interactions in a changing chemical climate.

3) There are lots of small sections in this manuscript, some consisting of single sentences. Consider combining them into more general sections. Specifically, 2.3.1-2 and 2.5-6.

Author Response: Following your suggestion: "2.3.1 Base coated denuders and filters" and "2.3.1 Acid coated denuders and filters" combined into a single section "2.3.1 Chemically coated denuders and filters"

"2.5 Data Quality Control" and "2.6 Bias correction applied to HNO3 data" have not been combined as they cover different aspects.

4) Many sentences leading paragraphs are structured as "For {atmospheric constituent},. . .". This is a bit formulaic and the authors may want to revise these sentences. Author Response: Thank you. We have gone through and revised where
appropriate.

Specific comments 1) Abstract: I find the final two sentences of the abstract to be the most compelling. There is a lot of detail, primarily on page 1, that would be better suited for the results section. Consider summarizing the text on the spatial and temporal trends and better connecting them to the changes in atmospheric HNO3 and NH3.

Author Response: Text revised in abstract.

2) Abstract Page 2 lines 5-6: ". . . indications that the atmospheric lifetime of HNO3 and NH3 has increased . . .". This does not seem correct to me. The lifetime of these gases has not increased but rather the phase/composition of these species have. There are now more gaseous and less aerosol bound NO3 and NH3 due to changes in SO2. This likely decreases the atmospheric lifetime of total nitrate and reduced nitrogen compounds as NH3 and HNO3 typically dry deposit faster than aerosol NO3 and NH4.

Author Response: See also response to Reviewer 1 (comment 32). Text revised in abstract: "Since 1999, AGANet has shown substantial decrease in SO2 concentrations relative to HNO3 and NH3, accompanied by large reductions also in the aerosol components, with evidence of a shift in the particulate phase from (NH4)2SO4 to NH4NO3. The potential for NH4NO3 to act as a reservoir for NH3 and HNO3, together with the surfeit of NH3 means that a larger fraction of the nitrogen is remaining in the gas phase, maintaining higher concentrations of NH3 and HNO3 in the UK. . . ."

3) Page 4 lines 17-29: This paragraph contains similar information as the previous paragraph. Consider combining it with the previous paragraph Author Response: Replicated information deleted.

4) Page 6 lines 21-22: This is an awkward introductory sentence for this paragraph. Consider revising or adding an introductory paragraph that introduces the importance of the denuder base coating.

Author Response: Revised to: "Sodium carbonate (Na2CO3) is an effective sorbent

for acid gases, allowing simultaneous collection of HNO3, SO2 and HCl on denuders (e.g. Ferm 1986). Since the measurement of aerosol Na+ is also of key interest in AGANet however, a potassium carbonate (K2CO3) coating is used instead to eliminate the possibilities of Na+ contamination from Na2CO3. Glycerol is added to the K2CO3 coating, as it increases adhesion. . . . . . .,"

5) Sections 2.5: Are data that failed the quality checks removed from the analysis?

Author Response: Sections 2.5 Data Quality Control "i) Air flow rate (0.2 – 0.4 L min-1): where this is below the expected range for a sampling period, the data is flagged as valid but failing the QC standard. ii) Denuder capture efficiency: where this is less than 75% for a sample, the data is flagged as valid but less certain. iii) Ion balance checks: close agreement expected between NH4+ and the sum of NO3- and 2ïĆťSO42-, as NH3 is neutralised by HNO3 and H2SO4 to form NH4NO3 and (NH4)2SO4, respectively (Conolly et al., 2016), and for Na+ and Cl-, as these are marine (sea salt) in origin."

Data failing the above quality check are not automatically removed from analysis.

Air flow rates: The air pumps used are relatively stable, at 0.2 – 0.4 l min-1. If a low air flow rate is due to temporary loss of power and/or air pump issue, the data is accepted provided that the flow rate does not drop below 25% of the normal range. The data is flagged as valid (EMEP data flag), but has higher uncertainty. If low air flow rate is due to a leak or obstruction (e.g. kinking of tubing), the data is rejected.

Denuder capture efficiency: Two denuders in series are used for every sample to check capture efficiency for reactive gas: two carbonate denuders for capture of HNO3, SO2 and HCl and two acid coated denuders for capture of NH3. Samples with < 75% of the total gas captured in the first of the two denuders are accepted but are flagged as valid (EMEP data flag) but has higher uncertainty.

Ion balance checks: Ratio of NH4+ ($\mu$eq): (2*SO42- + NO3-) ($\mu$eq) Expect 1:1 as NH3

neutralised by HNO3 and SO2 (H2SO4) to form NH4NO3 and (NH4)2SO4 Acceptable range = 0.2 to 3. Ion balance checks are carried out at site level and in collated file (with regression plots for outliers) Data are rejected if the ratio is outside the range. Ratio of Na ($\mu$eq):Cl ($\mu$eq). Expect 1:1 as Na and Cl aerosols derived mainly from sea salt. Acceptable range 0.2 to 3. Ion balance checks are carried out at site level and in collated file (with regression plots for outliers) Data are rejected if the ratio is outside the range.

6) Section 2.5 iv) What is the criteria to determine anomalies and outlies?

Author Response: This is a screening process carried out for a small number of runs where there was clearly a sampling malfunction. This exclusion includes events such as vandalized or damaged samples, water ingress or equipment/analytical problems (e.g. mix-up between carbonate and acid coated filters).

7) Section 2.6 Line 22: Does the empirical factor used for HNO3 bias correction exhibit any dependence on season, temperature or solar radiation? If the bias is due to oxidants, then I would expect a dependence in the bias on seasonal and environmental parameters.

Author Response: See response to reviewer 1 (comment 1, pages 1 – 2).

8) Page 9 lines 28-29: The mean difference between the measurements are given here but what is the scatter between the measurements and the median difference. A correlation coefficient would provide some information about the scatter and a median difference would indicate how normal the distribution is and if the bias is being driven by high values in one of the measurement techniques.

Author Response: Page 9 lines 28-29: "Agreement between the DELTA and ADS was within 19 % for SO2 (mean DELTA = 1.75 $\mu$g m-3 cf mean ADS = 2.18 $\mu$g m-3) and 4 % for HCl (mean DELTA = 0.40 $\mu$g m-3 cf mean ADS = 0.41 $\mu$g m-3)." Linear regression (R2) is provided in table 2. Regression plots (DELTA v ADS) were provided in Supplementary materials (Figure S2) – since there were already a lot of figures and tables in the paper.

Supplement Figure S2 and Table 2 has been combined into a single Figure 2 – see response to review 1 (comment 32, page 16).

9) Page 9 line 32: Difference in the instrumentation flow rates and/or inlets could result in the instruments measuring different sized aerosols and my influence the differences in SO4.

Author Response: Page 10 line 3: "A particle size cut-off of around 4.5 $\mu$m was estimated for the DELTA air inlet) (Tang et al., 2015)". Page 10 lines 6 to 7: "In comparison, the ADS has a 2.5 $\mu$m cyclone in front of the aerosol filters to collect aerosols < 2.5 $\mu$m on the aerosol filters"

ADS size cut-off (cyclone) = 2.5 um @ flow rate of 10 LPM. Note that the cyclone is in front of the aerosol filters, and not at the inlet, as in the case for the URG ADS (http://www.urgcorp.com/index.php/systems/manual-sampling-systems/annular-denuder-system)

10) Page 16 lines 1-2: The peaks in NHx and SO4 in the spring may just be coincidental. The spring time could also be a time in which the aqueous formation pathway of SO4 is at its maximum or the SO2 emissions from heating or transportation may be larger. In the US, the SO4 concentrations typically peak in the summer while the NH3 concentrations peak in the spring.

Author Response: In the UK, SO2 concentrations are highest in winter (January and February) and lowest in summer (June – August). The peak in SO4$^{2-}$ concentrations however occur in March every year, where peaks in concentrations of NH3 and NH4+ from measurements made at the same time from the same sites are also observed (please note that the seasonal cycle of ammonia vary, depending on the emission source types in the vicinity of the measurement site, see Tang et al. 2018). Formation of SO42- is largely governed by the availability of SO2 and NH3 to form the stable (NH4)2SO4, and the spring peak may be attributed to enhanced formation of (NH4)2SO4, since peaks in concentrations of NH3 and NH4+ also occur in spring. Import of particulate (NH4)2SO4 from long-range transboundary transport also enhances the springtime concentration. In summer, sunny, warmer conditions increases photochemical oxidation of SO2 to H2SO4 and subsequent formation of sulphate aerosols leading to higher aerosol SO42- concentrations in summer than in winter.

Revised/expanded text in "section 3.6 Seasonal variation in acid gases and aerosols (paragraph 5) "SO2, by contrast, are highest in the winter, with concentrations exceeding summer values on average by a factor of 2 (Figure 8). Increased emissions of SO2 from combustion processes (heating) during the winter months, coupled to stable atmospheric conditions resulting in build-up of concentrations at ground level contributes to the winter maximum. Since the reaction of SO2 with NH3 to form (NH4)2SO4 is effectively irreversible (Bower et al., 1997), the ratio of the concentrations of SO2 and SO42- (Figure 9) is largely governed by the availability of SO2 and NH3 to form (NH4)2SO4. The temporal profile of SO42– has a peak in concentrations in spring, although not as pronounced as the NO3- peak (Figure 8). The spring peak may be attributed to enhanced formation of (NH4)2SO4, since peaks in concentrations of NH3 and NH4+ also occur in spring (Figure 8) and from the import of particulates from long-range transboundary transport. Unlike SO2, aerosol SO42- concentrations are higher in summer than in winter, due to increased photochemical oxidation of SO2 to H2SO4 and subsequent formation of sulphate aerosols in sunnier and warmer conditions (Mihalopoulos et al., 2007). In winter, lower SO2 oxidation rates limits H2SO4 formation and therefore also the formation of (NH4)2SO4."

11) Page 16 Line 5: "Na+ and Cl-" have highest concentrations during winter . . .' Is salt used for the treatment of road surfaces in the Winter in the UK?

Author Response: Yes, indeed rocksalt (NaCl) is used to treat road surfaces in the winter in the UK. About 2 million tonnes are used every year, with most of it going

on motorways, trunk roads and main roads. Aerial salt spray has been shown to increase the concentrations of particulate NaCl in the immediate vicinity of treated road surfaces. For example, a study by Palmer et al. (2004) found that the concentrations of particulate NaCl at the road edge (2m) was 25 - 70% higher than at 10 m away from the road edge, with concentrations at 10 m approaching background NaCl concentrations (ca. 35 nmoles m-3 from UK AGANet). Aerial salt spray from treatment of road surfaces in winter is however unlikely to contribute to an increase at regional / national level in the UK. Sources of seasalt aerosol (NaCl) in UK has been shown by high time-resolution measurements (MARGA) in the UK to originate from air masses coming into the UK with the predominant south-westerly winds (seasalt aerosol generated off the atlantic) (pers. comm.). AGANet sites are all located away from roads, and should not be affected by local seasalt from gritting. Reference: Palmer S.C.F, Cape J.N, Truscott A-M, Black H.I.J, Tang Y.S, Swaine, Van Dijk N, Smart S.M, Sutton M.A, Fowler D, Biodiversity in roadside verges: CEH Final Report to SEERAD. 108 pages. 2004.

12) Page 20 line 18: Significant has a specific statistical meaning. I think "larger" would be a more appropriate. Author Response: Thank you. "significant" replaced with "substantial"

13) Page 21 line 5: ". . . SO2 towards it being dominated by NH3, . . ." This appears to be a bit binary. There are lots of constituents in the air, many of which were note measured here. More context is needed.

Author Response: OK, thanks. Sentence has been deleted. Changes in NH3 relative to other acid gases are discussed in section 3.8.5 Changes in UK chemical climate.

14) Page 22 line 12: "expected to increase residence times of NH3 and HNO3 in the atmosphere" If we are in an NH3 limited environment, I can see how this would increase HN4NO3 and how that could increase the atmospheric lifetime of HNO3 as it is partitioned to NO3 aerosols. However, I do not see how this increases the NH3 lifetime. NH3 will preferentially partition with SO4, which is more thermodynamically stable

than NH4NO3, this should decrease the lifetime of NH3 if anything as the NH4NO3 will evaporate where the (NH4)2SO4 aerosol would not.

Author Response: See revised/expanded text in "3.8.5 Changes in UK chemical climate (paragraph 5 and 6)" "At the same time, reduction in emissions of the precursor gases have also led to a lower formation of particulate phase NH4+, NO3- and SO42- in the atmosphere and changes in atmospheric composition. Since the affinity of H2SO4 (oxidation product of SO2) for NH3 is much larger than that of HNO3 and HCl, available NH3 is first taken up by H2SO4 to form ammonium sulphate compounds (NH4HSO4 and (NH4)2SO4), with any excess NH3 then available to react with HNO3 and HCl to form NH4NO3 and NH4Cl that are volatile. Analysis of the different particulate components in sect. 3.5 showed that the ammonium aerosols are mainly made up of (NH4)2SO4 and NH4NO3. With the large reduction in SO2, more NH3 is available to react with HNO3 to form NH4NO3 and concentrations of NH4+ and NO3- are now observed to be in molar excess over SO42-, providing evidence of a change in the particulate phase from (NH4)2SO4 to NH4NO3 (Figure 18b). A change to an NH4NO3 rich atmosphere and the potential for NH4NO3 to release NH3 and HNO3 in warm weather, together with the surfeit of NH3 also means that a larger fraction of the reduced and oxidised N is remaining in the gas phase as NH3 and HNO3. An increased partitioning to the gas phase may account for the larger decrease in particulate NH4+ (MK −62% between 2000-2015, n = 12) and NO3- (MK −52% between 2000-2015, n = 12) than NH3 (MK −30% between 2000-2015, n = 12) and HNO3 (MK −45 % between 2000-2015, n = 12) (Table 5) and the increase in gas to aerosol ratios (NH3:NH4+ and HNO3:NO3-) over the 16 year period (Figure 17). A higher concentration of the gas-phase HNO3 and NH3 may therefore be maintained in the atmosphere than expected on the basis of the emissions trends in NOx and NH3. Given the larger deposition velocities of NH3 and HNO3 compared to aerosols, more of the NH3 and HNO3 emitted will have the potential to deposit more locally with a smaller footprint within the UK. "

15. Figure 8: I am happy to see a measure of scatter on these plots as the SD. However, a 5% and 95% CI would be more informative as it would give the reader an idea about the distribution of the data.

Author Response: The 4 graphs in Figure 8 have been replotted showing the 95% CI (T test).

————————————————————

---

## Author Response (AR1)

**Author Response:**
Additional text has been added under new heading "Section 3.3: Uncertainties in $HNO_3$ determination" to address the reviewer's comments.

[revised manuscript text omitted]

Additional text also added to revised/expanded text in Section 3.6 Seasonal variation in acid gases and aerosols (paragraph 2):

"$HNO_3$ is a secondary product of $NO_x$, but $NO_x$ emissions are dominated by vehicular sources which are not expected to show large seasonal variations. Seasonal changes in chemistry and meteorology are therefore more likely to be a source of the observed variations in $HNO_3$ and $NO_3^-$ (Figure 8). A weak seasonal cycle is observed in $HNO_3$, with slightly higher concentrations in late winter and early spring that may be due to photochemical processes with elevated ozone in spring (AQEG 2009) leading to formation of $HNO_3$ during this period (Pope et al., 2016). **As discussed in section section 3.3, a constant correction factor was applied to all $HNO_3$ data, which does not take into account seasonal dependency. The concentrations in $HNO_3$ may therefore be over-estimated in winter (less $HNO_3$ formed from photochemical processes) and under-estimated in summer (larger $HNO_3$ concentrations due to increased $^{\cdot}OH$ radicals for reaction with $NO_2$ to form $HNO_3$), masking the true extent in the seasonal profile."**

*2)      Specific Comments*
*p6, l4, '...sampling rate of 0.2-0.4 l/min...' Please mention at this stage, or just below in the paragraph describing the aerosol collection system, what the particle size cut-off is for the DELTA sampler (mentioned later on p10, l3). It is important to know what the size spectrum of collected aerosols is, and that some (coarse) particles are not sampled, eg dust, large marine aerosols.*

**Author Response:**
The text below was inserted at the end of the paragraph 3 (section 2.2 Extended DELTA methodology for sampling acid gases and aerosol in AGANet).

"A particle size cut-off of around 4.5 µm was estimated for the DELTA air inlet (Tang et al., 2015). The DELTA will therefore also sample fine mode aerosols in the $PM_{2.5}$ fraction, as well as some of the coarse mode aerosols < $PM_{4.5}$."

*3)       p7, l6-7, change of analytical labs from Harwell to CEH Lancaster in 2009: was there a transitional period of overlapping parallel measurements by the two labs, to make sure no bias was introduced in the long term time series by the change of laboratory?*

**Author Response:**
There was no transitional period of overlapping measurements, but measurements of replicated samples were compared between the two labs to ensure that there was no bias in chemical analysis prior to the lab. switch. CEH Lancaster laboratory is UKAS accredited, with experience of DELTA measurements prior to taking over the network measurements from Harwell lab.

*4)       p7, l26, '...flagging up occurrences of poorly coated denuders and/or sampling issues...'*

*Another possibility is that concentrations are so large that the first denuder saturates and thus much is collected by the second denuder. This can happen for NH3 at agricultural sites after fertilisation; it is much less likely for acid gases due to lower concentrations, unless perhaps at some polluted urban stations?*

**Author Response:**
At high concentrations, saturation of the first denuder can indeed lead to lower gas capture efficiencies (breakthrough and capture on second denuder). The monitoring network sites are however located away from sources to monitor ambient concentrations. In 2015, the mean capture efficiencies for $NH_3$, $HNO_3$, $SO_2$ and HCl were 96%, 83%, 91% and 79 %, respectively.

*5)       p8, l15, the term 'bias' is used in relation to the 0.45 correction factor for HNO3, in the title of 2.6 and also other parts of the text. This is perhaps misleading as a bias suggests an offset, while the multiplicative correction applied acts on the span. Further, in the Tang et al 2015 report, the authors write that '... It is recommended that a correction factor of 0.45 be applied to the historic HNO3 measurements. The range of ratios was 0.44±0.15 (±2SD), i.e. 0.29-0.59, therefore it is reasonably likely that the value lies between 0.4 and 0.5. Therefore a correction factor of 0.45 should be applied...' It is quite clear that the percentage of non-HNO3 NOy compounds that is measured after extracting K2CO3-coated denuders depends on the relative abundances of these gases compared with HNO3, as well as their collection efficiencies on K2CO3 and their oxidation/reaction rate following adsorption. I would expect large seasonal changes, and large spatial/geographical variations, in these concentrations and the associated chemical processes, as reflected in the observed 0.29-0.59 range.*
*Applying the same correction factor at all sites of the network, that range from remote to coastal to rural to sub-urban and urban, does not seem to be adequate. This is hinted at in the Eskdalemuir example of Fig. 3, where applying the large 0.45 multiplier makes the DELTA TIN values diverge from the EMEP filter pack measurements, ie at this rural background site the need for such a large correction is not warranted.*
*The correction factor should account for the differences in pollution climates between sites, and also for changes over the 20-year period. Could an empirical correction be derived from chemical transport modelling (eg EMEP4UK), whereby the ratios of modelled HNO3 to NO2, HONO, PAN, etc, are used to construct a geographically- and temporally-varying index to drive the correction function? The HNO3 data reported in Tang et al (2015) for NaCl vs K2CO3 coating, with measurements made in contrasted situations (rural, urban, remote, see Table 1 in that report), may be used for calibrating such a function.*

**Author Response:**
The tile of section 2.6 has been changed to "$HNO_3$ measurement artefacts and correction"

Regarding the empirical correction of the $HNO_3$ data, please see author response to general comments on pages 1 - 2.

*6)	p8-11, section 2.7 Performance of the DELTA method: strictly speaking, this section describes measurement results from intercomaprison experiments or even long term datasets (Bush, Eskdalemuir) does not belong in Methods, and should be moved to the beginning of Section 3- Results*

**Author Response:**

Section 2.7 Performance of the DELTA method was included in the method section to separate this component from the main focus of presenting AGANet data in the results and discussion section.

But agree:

Moved to beginning of section 3 – Results and Discussion.

3.1 Performance of DELTA method

3.1.1 Comparison with daily annular denuder measurements

3.1.2 Comparisons with filter pack measurements: $HNO_3/NO_3^-$ and $NH_3/NH_4^+$

3.1.3 Comparisons with bubbler and filter pack measurements: $SO_2$ and $SO_4^{2-}$

*7)	p10, l12: Ca++ and Mg++ concentrations were near detection limits because they are mostly in the coarse fraction, with particle sizes near or above the DELTA cut-off. How much of the NaCl is similarly not collected by the DELTA system? Also, in relation to the DELTA v. ADS intercomparison, the loss of NO3, Na and Cl on the surface of the cyclone is put forward as an explanation for the lower ADS aerosols concentrations (compared with DELTA) (p10, l10), but why in that case is SO4= 23% larger in the ADS?*

**Author Response:**

NaCl is the main constituent of seasalt aerosol and size of seasalt aerosols ranges widely from ~0.05 to 10 µm in diameter, with their particle sizes also varying with humidity. The DELTA cut-off is estimated to be around 4.5 µm, so the DELTA will sample NaCl aerosols in the $PM_{4.5}$ particle size region. $Na^+$ measured on the DELTA are above detection limits, with concentrations ranging between 0.4 to 1.8 ug Na $m^{-3}$ (annual mean in 2015). In the DELTA v ADS intercomparison of base cation measurements, the slope for Na was 3.0 ($R^2 = 0.24$) and for Mg, the slope was 2.4 ($R^2 = 0.24$), but a lot of scatter for $Ca^{2+}$ as both ADS and DELTA $Ca^{2+}$ data were at or below LOD. The DELTA therefore captures $Mg^{2+}$ and $Na^+$ well, but not $Ca^{2+}$, which is what we find in the AGANet data.

The slope for $SO_4^{2-}$ in DELTA v ADS intercomparison is 0.69 ($R^2 = 0.89$). The smaller $SO_4^{2-}$ signal on the DELTA may be due to incomplete capture of fine mode sulphate on the DELTA base coated cellulose filters. In the DELTA assessment report by Tang et al. (2015), up to 30 % of the total acid sulphate was measured on a 2µm porosity PTFE membrane placed behind the $K_2CO_3$ coated filter to capture break-through. Since 2016, an additional PTFE membrane is added in front of the carbonate and acid coated cellulose filters.

A detailed assessment of the DELTA system against filter pack with a focus on $SO_2$ and $SO_4^{2-}$ in 1999 by Hayman et al. (2006) had previosuly shown close agreement between the two methods, providing confidence in $SO_2$ and $SO_4^{2-}$ measurements by the DELTA. Sulphur measurements provided by the DELTA replaced filter pack measurements in 1999.

Further work is ongoing to understand, assess and correct the bias in $SO_4^{2-}$ measurements in historic data. Since the DELTA method was unchanged for the assessment period in this paper, the bias in $SO_4^{2-}$ should not influence the interpretation of long-term trends in the data.

*8)	p13, l2: the highest HCl concentrations are in the SE and SW of England, but also in the Midlands*

**Author Response:**

Thank you. For an international audience, they may not know where the Midlands is. I propose to use "central England" instead of "Midlands",

Revised text below:
"HCl in the atmosphere are mostly emitted from coal combustion and the highest concentrations of HCl are in the source areas in SE and SW of England, and also in central England (north of the Ratcliffe-on-Soar power station)…,…"

9)          p13, l5: '...Further away from the coast and influence of marine aerosol, the smallest concentrations of Cl and Na+ are measured in the west of the country (Lough Navar in Northern Ireland...' Lough Navar is very near the coast (10km) and yet NaCl concentrations are very low (similar to concentrations in the Midlands), compared with all other western sites in the network (Fig. 5); what could be the reason?

**Author Response:**

The Lough Navar site is actually approx. 40 km inland, close to the border between Northern Ireland and Republic of Ireland, within a forested area. The UK maps in the manuscript are all shown without Republic of Ireland, which may have given a false impression of Lough Navar being closer to the sea than it is in reality. Given its location inland, and the prevailing wind direction coming from the SW, it is far from the influence of seasalts.

[Figure]

10)          p13, l5: For Cl- and Na+, '...largest concentrations at coastal sites in the south (Barcombe Mills) and west (Yarner Wood)...'; actually the highest concentrations are at Goonhilly at the SW tip of Cornwall.

**Author Response:**

Thank you for spotting that.

Barcombe Mills and Yarner Woods are two of the original 12 sites that were established in 1999 in AGANet. As coastal sites, $Na^+$ and $Cl^-$ were always highest at these two sites up to the point when the new Goonhilly site in Cornwall was added as part of the network expansion in 2006. $Na^+$ and $Cl^-$ are indeed higher at Goonhilly than Barcombe Mills and Yarner woods.

Text has been corrected accordingly.

"The spatial distributions of $Cl^-$ and $Na^+$ were similar, with largest concentrations at the coastal sites Goonhilly in SW England and Lerwick-Shetland in the Shetland Isles,…."

*11)  p13, l9-10, '...There is however no clear spatial pattern for Ca2+, with concentrations that are mostly at or below LOD...' For both Ca and Mg, which are mostly in the coarse fraction, it may be argued that the DELTA system does not allow a realistic assessment of the total concentration, because a large share of coarse particles are not collected.*
*Please comment.*

**Author Response:**

Ca, Na and Mg are mainly in the 1 – 10 µm fraction in ambient aerosol. The size cut-off for the DELTA is around 4.5 µm, which means it will sample base cations in the $PM_{4.5}$ fraction.

In the DELTA ($PM_{4.5}$) v ADS ($PM_{2.5}$) intercomparison of base cation measurements (see response to comment 7 earlier), the slope for Na was 3.0 ($R^2 = 0.24$) and for Mg, the slope was 2.4 ($R^2 = 0.24$), but no relationship was established for Ca2+ as both ADS and DELTA $Ca^{2+}$ data were at or below LOD.

This suggests that the DELTA captures $Mg^{2+}$ and $Na^+$ in the $PM_{4.5}$ fraction reasonably well. At all AGANet sites, Na and Mg measurements are above LOD, whereas $Ca^{2+}$ are mostly at or below LOD. Aerosol filter blanks for $Ca^{2+}$ are also much more variable than Na or Mg. $Ca^{2+}$ is particularly problematic in chemical analysis as adsorption losses readily occur due to electrostatic interaction between $Ca^{2+}$ and surfaces, especially plastic. To this end, aerosol sample extracts are acidified to minimise adsorption of $Ca^{2+}$ to surfaces.

Sampling of $Ca^{2+}$ is also likewise problematic as $Ca^{2+}$ can potentially stick to inlets and surfaces. Tests conducted to assess adsorption losses of components to the connecting 6-mm diameter LDPE tube in the DELTA sampling train showed that measured concentrations of $Ca^{2+}$ were within the noise of the LDPE tube blanks (i.e. clean LDPE tubes extracted with deionised water), adding to the uncertainty in $Ca^{2+}$ measurements (see also response to next reviewer comment 12).

[Figure]

Retention and transmission efficiency of the DELTA sampling train as used in the AGANET network

*12)        Further, our own tests with DELTA systems at INRA indicated very substantial losses for Mg and Ca in all the non-filter parts of the sampling train (particularly the 6-mm diameter LDPE elbow connecting the 2nd acid denuder to the first NH3 denuder, Fig.S1), which are therefore not measured on the filter. We analysed the loss fraction LDPE / (LDPE + den + filter) for all compounds; for NH4+ and NO3- this was less than 5%; for Cl- and Na+ this was 5-10%; for SO4= and Mg2+ this was 10-15%; while for Ca2+ this was 30-40%. Beyond the question of coarse aerosols that were not sampled at all (did not enter the sampling train), there is the question of those coarse aerosols that 'did not make it' to the filter pack. Did the authors carry out similar tests, and could the results be shown in the supplement? It may be that the new straight design for the DELTA sampling train allowed a reduction of these losses?*
*Please comment.*

**Author Response:**

Potential loss of particulate components to the connecting tube 6-mm diameter LDPE in the DELTA sampling train (Fig.S1) was investigated and reported in the DELTA assessment report by Tang et al. (2015).

Our test results (extracted from Tang et al., 2015) are similar to the INRA findings outlined above:

- $NO_3^-$: loss to LDPE tube is negligible ($2.4 \pm 0.8$ % (mean $\pm$ SD) across all sites for all available data).
- $NO_2^-$, $SO_4^{2-}$ and $Cl^-$: losses to LDPE are small ($< 6\%$).
- Base cations $Na^+$ and $Mg^{2+}$: losses to LDPE are slightly higher ($<7\%$).
- Base cations $Ca^{2+}$: there is a large degree of uncertainty in the calcium assessment, due to 1) variability of $Ca^{2+}$ in the blank LDPE tube extracts and 2) very low $Ca^{2+}$ on LDPE tubes from sites, that were similar to blank values and close to the detection limit (LOD = 0.05 mg/L $Ca^{2+}$).

Since January 2016, the new DELTA sample train configuration is linear, eliminating the use of the LDPE connecting tube.

*13)        P15, l1: This section 3.4 is mostly about sub-annual (seasonal) variations, so could be re-named 'Seasonal variations in acid gases and aerosols', as opposed to long term trends of Sections 3.5-3.6*

**Author Response:**

OK. Agree. Renamed
"3.6 Seasonal variation in acid gases and aerosols"

*14)        p15, l9-10: '...In spring, the peak in HNO3 and NO3...' Fig.7 does not actually show any spring peak for HNO3; the late winter (Feb-Mar) concentrations are only marginally higher (but not significantly different accoring to the error bars) than the rest of the year? The opening sentence of the paragraph should read '...maximum in late winter and early spring...'*

**Author Response:**

Thank you. Text revised in "Section 3.6 Seasonal variation in acid gases and aerosols"

See below:

"$HNO_3$ is a secondary product of $NO_x$, but $NO_x$ emissions are dominated by vehicular sources which are not expected to show large seasonal variations. Seasonal changes in chemistry and meteorology are therefore more likely to be a source of the observed variations in $HNO_3$ and $NO_3^-$ (Figure 8). $HNO_3$ has a weak seasonal cycle with slightly higher concentrations in late winter and early spring that may be due to photochemical processes with elevated ozone in spring (AQEG 2009) leading to formation of $HNO_3$ during this period (Pope et al., 2016). As discussed in section 3.3, a constant correction factor was applied to all $HNO_3$ data, which does not take into account seasonal dependency. The concentrations in $HNO_3$ may therefore be over-estimated in winter (less $HNO_3$ formed from photochemical processes) and under-estimated in summer (larger $HNO_3$ concentrations due to increased $\cdot OH$ radicals for reaction with $NO_2$ to form $HNO_3$), masking the true extent in the seasonal profile."

*15)     p15, l22, '...this contributes to the winter minimum in NH4NO3...' : the minimum NO3- actually occurs in July?*

**Author Response:**
Thank you. Text revised in "Section 3.6 Seasonal variation in acid gases and aerosols"

See below:
"Warm, dry conditions in summer promotes dissociation, increasing gas-phase $HNO_3$ relative to particulate-phase $NH_4NO_3$, limiting peak $NO_3^-$ aerosol concentrations (Figure 8). This process accounts for the minima in $NO_3^-$ concentrations (Figure 7) and the highest ratio of $HNO_3$ to $NO_3^-$ seen in July (Figure 8). Cooler conditions in the spring than early autumn sees a larger fraction of the volatile $NH_4NO_3$ remaining in the aerosol phase. The peak in $NO_3^-$ concentrations and the low $HNO_3:NO_3^-$ ratio in spring-time (Figure 8) is thus a combination of larger $NO_3^-$ from reaction between higher concentrations of the precursor gases $HNO_3$ and $NH_3$, and partitioning to the aerosol phase. Import from long-range transboundary transport of particulate $NO_3^-$ e.g. from continental Europe into the UK, as discussed in Vieno et al. (2014, 2016) adds to the elevated $NO_3^-$ concentrations. In winter, low temperature and high humidity also shifts the equilibrium to formation of $NH_4NO_3$ from the gas-phase $HNO_3$ and $NH_3$. Since $NH_3$ concentrations are lowest in winter however, with less $NH_3$ available for reaction, $NH_4NO_3$ concentrations are correspondingly smaller in winter than in spring or autumn."

*16)     p16, l9-13: how far should seasonal cycles for Mg and especially Ca be discussed, given the low collection efficiency (and thus high uncertainty) of filter data (see my comment above on aerosol size cut-off and losses in sampling train for these large aerosols)?*

**Author Response:**
The discussion of the seasonal cycles on $Mg^{2+}$ and $Ca^{2+}$ are based on what the measurement shows. $Mg^{2+}$ measurements were above LOD, with similar trends (spatial and seasonal) to $Na^+$, so a discussion on seasonal cycle for Mg is warranted. In the case of $Ca^{2+}$, uncertainties in interpretation of the $Ca^{2+}$ data is discussed.

*17)     p17 and beyond, general comment on sections 3.5-3.6: a linear regression is fitted to all datasets from 1999 through 2015, but looking closely at the 15-yr time series for the 12 sites (eg Fig. 12-13), for HNO3, NO3-, SO4=, NH4+, NH3, it appears that concentrations were rather stable (with some interannual variability but no trend) in the period 2000-2007, and then only started declining after 2007. The only exception is SO2 with a continuous decline all the way. Fitting a linear trend is helpful to quantify an multi-annual rate of decrease (which is what you do), but is not an accurate representation of the time course of concentrations. Can you think of any plausible explanation for a change of course around the year 2007: change or implementation of pollution control policies? Decadal change in weather patterns? It might be useful to show (in the supplement)a summary of weather patterns for all sites of the network, the 15-yr time course of temperature, rainfall, wind speed etc.*

**Author Response:**
"Section 3.8 Assessment of trends in relation to UK emissions" has been revised and expanded to include a more thorough discussion of trends under new sub-headings.
3.8.1 Trends in $HNO_3$ and $NO_3^-$ *vs* $NO_x$ emissions
3.8.2 Trends in $SO_2$ and $SO_4^{2-}$ *vs* $SO_2$ emissions
3.8.3 Trends in HCl and $Cl^-$ *vs* HCl emissions
3.8.4 Trends in $NH_3$ and $NH_4^-$ *vs* $NH_3$ emissions
3.8.5 Changes in UK chemical climate

Revised/expanded text:
"The overall downward trends in $HNO_3$ and $NO_3^-$ are seen to be broadly consistent with the −49 % fall in estimated $NO_x$ emissions (NAEI, 2018) over the 16 year period between 2000 and 2015 (Figure 14). Reductions in combustion (power stations and industrial) and vehicular sources (fitting of catalytic converters), coupled to tighter emission regulations are major contributory factors to the decrease in UK $NO_x$ emissions. The rate of reduction however stagnated in the period 2009 and 2012 (improvement in emissions abatement offset by

proportionate increase from diesel combustion and increase in vehicle numbers), followed by a 16 % decrease between 2012 and 2015 due to the closure of a number of coal-fired power stations.

It is notable that the first 6 years (2000-2006) of $HNO_3$ and $NO_3^-$ annual data show substantial inter-annual variability and in particular are dominated by the large 2003 peak in concentrations (see sect. 0). Variability in the annual data thus highlights the sensitivity of the trend assessment to the selection of a reference start for the time series, since the annual mean concentrations of both $HNO_3$ and $NO_3^-$ in 2000 are in fact smaller than concentrations in the following 6 years. Re-analysis of the same annual data normalised against 2001 instead of 2000 takes the relative trend line for $HNO_3$ and $NO_3^-$ much closer to the relative trend line in $NO_x$ emissions. In the later period between 2006 and 2015, the relative trend lines in $HNO_3$ and $NO_3^-$ using mean data from 12 or 30 sites were not significantly different and emissions and concentrations trends followed each other closely."

[Figure]

Regarding the reviewers comment on the possibility of change in weather patterns to explain the apparent biphasic trend, the UK annual average temperature and rainfall (https://www.metoffice.gov.uk/climate/uk/summaries) show no overall trend in the 16 years of climate data between 1998 and 2015. 2010 was however an unusual year, with a lower than average mean annual temperature of 7.9 °C due to an exceptional cold winter, with Dec 2010 recorded as the coldest for over 100 years (cf. 9.2 °C average for 2000 to 2015) and lower than average rainfall of 950 mm (cf. 1180 mm average for 2000 to 2015). Graph of UK annual mean temperature and rainfall has been added to supplementary materials.

[Figure]

*In terms of implementation of pollution control policies that could explain the change in course of the pollutant trend concentrations*:

In 2007, the designation of Nitrate Vulnerable Zones (NVZs) in the UK was introduced to strengthen the range of measures in the Nitrates Action Programme under the Nitrates Directive (91/676/EEC). NVZs are areas designated as being at risk from agricultural nitrate pollution and farms within NVZs must comply with the rules laid down on use of nitrogen fertiliser and storage of organic manure. Adoption of NAP by farms will also likely reduce emissions of $NH_3$. $NH_3$ data from the 12 sites in AGANet were stable from 2000-2010 and decreased between 2010 and 2012 with concentrations again stabilising after 2012.

It could be surmised that there was more $NH_3$ before 2007 to react with the acid gases and form / maintain higher concentrations of aerosols. But it has also to be borne in mind that the period between 2000 and 2007 was subject to a pollutant episode in 2003 and the data, as you also pointed out, is extremely variable. The apparent change in course of pollutant concentrations in $NO_3^-$ and $SO_4^{2-}$ is more likely due to influences of import from long range transboundary pollutant transport and meteorology.

*18)    p18, l6: '...The long-term time series in annually averaged concentrations of the gas and aerosol components are shown in Figure 12a and Figure 12b...': would it be possible to show, alongside the measured DELTA time series, the modelled NO/NO2 time series (from a CTM, eg EMEP4UK) for the same sites? In a way this would account for both NOx emission changes as well as climatic variability over the period.*

**Author Response:**
Dr Massimo Vieno (CEH) is currently working on a paper comparing EMEP4UK with measurement data from NAMN and AGANet.

$NO_2$ concentrations is however measured at rural sites across the UK in the UKEAP NO2-net ($NO_2$ diffusion tube network), some of which are co-located with the AGANet. The network average in annual mean $NO_2$ concentrations showed a downward trend, decreasing from ~8 µg $NO_2$ m$^{-3}$ in 2000 to ~ 4 µg $NO_2$ m$^{-3}$ in 2015 (Conolly et al. 2016).

In terms of climatic variability from the UK, there is no apparent trend in the UK rainfall and temperature data (see earlier response to comment 17).

Additional text added in section 3.8.1 Trends in $HNO_3$ and $NO_3^-$ *vs* $NO_x$ emissions, end of last paragraph.
 A comparison of the network averaged $NO_2$ concentrations with $NO_x$ emissions by Conolly et al (2016) showed matching decreasing trends between 2000 and 2015, with annual mean $NO_2$ concentrations falling 2-fold to 4 µg $NO_2$ m$^{-3}$ in 2015 (Conolly et al. 2016). Although there is uncertainty in the corrected $HNO_3$ data (see section3.3), the encouraging agreement between $HNO_3$, $NO_2$ concentrations and $NO_x$ emissions lends support to a linear response in $HNO_3$ concentrations to reductions in $NO_x$ emissions.

*19)    p18, l12 '...The exceptions are Na+ and Cl- that have higher mean concentrations...' : Na+ is not shown in Fig.12.*

**Author Response:**
Mean concentration of $Na^+$ from 12 and 30 sites are compared in Table 4.
Table 4 inserted at the end of the sentence:
"The exceptions are $Na^+$ and $Cl^-$ that have higher mean concentrations from the 30 sites than the original 12 sites (Table 4)."

*20)    Figures 13, 14: use only one type of regression to simplify the figures (LR and MK give almost identical results)*

**Author Response:**
Figures 13 and 14 have been amalgamated into one single figure, with LR analyses taken out and moved to supplementary materials.

*21)        p20, l24-25 '...the reduction in SO2 emission and measured concentration is accompanied by a smaller negative trend in particulate SO4=...', and l27, '...The smaller decrease in particulate SO4= compared with its gaseous precursor, SO2, is similar to that observed at Eskdalemuir...'. Question: Is the smaller reduction rate in SO4= (compared with SO2) a reflection of the fact that increasingly in the UK, total sulphate includes a larger and larger fraction of marine sulphate, such that the decrease in anthropogenic SO4= (resulting from SO2 abatement) has a increasingly small effect on total sulphate?*
*Is it possible to re-calculate the SO4= trend separately for coastal and inland (eg Midland/London) sites?*

**Author Response:**

Additional text added in "section 3.8.2 Trends in $SO_2$ and $SO_4^{2-}$ vs $SO_2$ emissions" to discuss sea salt $SO_4^{2-}$ (SS_SO4) – see below:

"Sea salt $SO_4^{2-}$ (SS_SO4) aerosol, as discussed in section 3.5, makes up a significant fraction of the total $SO_4^{2-}$. It is possible that the smaller reduction in particulate $SO_4^{2-}$, compared with $SO_2$, may be explained by an underlying increase in the relative proportion of SS_SO4 to total $SO_4^{2-}$. To assess the contribution of SS_SO4 to the observed trends in total $SO_4^{2-}$, SS_SO4 concentrations (estimated according to the empirical equation described in Sect. 3.5) and NSS_SO4$^-$ (= total $SO_4^{2-}$ – SS_SO4)  are compared with the long-term trends in total $SO_4^{2-}$ in Figure 17. Overall, there is no trend in the long-term annual mean SS_SO4 data, with concentrations in range of 0.16 to 0.21 µg $SO_4^{2-}$. Since SS_SO4 is derived from an empirical relationship with Na$^+$ (sect.3.5), the long-term trend data for Na$^+$ is also included in the analysis (Figure 17). Similar to SS_SO4, there is no overall trend in the Na$^+$ data either, with small inter-annual variability and annual mean concentrations in the range of 0.65 – 0.85 µg Na$^+$ m$^{-3}$. SS_SO4 made up just 10% of the total $SO_4^{2-}$ in 2000, but by 2015, this had increased to just over 50% due to the decrease in NSS_SO4 over that time. MK analysis of the NSS_SO4 (Tables 4 and 5) showed decrease in concentrations of –78 % (2000-2015) and –62% (2006-2015), similar to that observed in $SO_2$ (–81 %: 2000 –2015 and –60 %: 2006 – 2015), indicating a closer relationship between NSS_SO4 and $SO_2$ than between total $SO_4^{2-}$ and $SO_2$."

[Figure]

[Figure]

NSS_SO4                    Total SO4

*22)     p21, l5-22: The argument about the NH3/SO2 ratio impacting the dry deposition velocities of SO2 and NH3 was developed in the 1980s and early 1990s, when SO2 concentrations were still very large in W. Europe. It is no longer sufficient to consider the NH3/SO2 ratio alone, since SO2 no longer massively dominates the acid load in W. European atmospheres. Instead, the ratio NH3/(2\*SO2 + HNO3 + HCl) should be computed to analyse long terme trends, as shown in Fowler et al. (Atmospheric Environment 43 (2009) 5193–5267, see Fig. 4.5). It is the combined effects of all acids and NH3 that determines the pH of cosystem/vegetation surfaces and hence their sink strength for water-soluble pollutants.*

**Author Response:**

Additional analysis of the change in molar ratios of $NH_3$ to acid gases and molar ratios of $NH_4^+$ to $NO_3^-$ and $SO_4^{2-}$ with time has been carried out – new figure added in manuscript:  Figure 18: Long-term changes between 2000 and 2015 in (a) molar ratio of $NH_3$ to acid gases ($SO_2$, $HNO_3$ and HCl) and (b) molar ratio of particulate $NH_4^+$ to acid aerosols ($SO_4^{2-}$ and $NO_3^-$) from measurements made at 12 sites in AGANet.

1)

[Figure]

Long-term changes in the molar ratio of $NH_3$ to acid gases ($SO_2$, $HNO_3$ and HCl) between 2000 and 2015 from measurements made at 12 sites in AGANet.

Revised/expanded text added, replacing  text on *p21, l5-22*

"3.8.5 Changes in UK chemical climate"

"Past studies have shown that the increasing ratio of $NH_3$ to $SO_2$ in the atmosphere leads to enhanced dry deposition of $SO_2$, accelerating the decrease in atmospheric $SO_2$ concentrations than would be achieved by emissions reduction alone (Fowler et al., 2001, 2009; ROTAP 2012). The dry deposition of $SO_2$ and $NH_3$, by uptake of the gases in a liquid film on leave surfaces, is known to be enhanced when both gases are present in a process termed "co-deposition" (Fowler et al., 2001). Where ambient $NH_3$ concentrations exceed that of $SO_2$, there is enough $NH_3$ to neutralize acidity in the liquid film and oxidise deposited $SO_2$, and maintain large rates of deposition of $SO_2$. With changes in the relative concentrations of acid gases in the UK and across Europe however, the deposition rates will increasingly be controlled by the $NH_3$/combined acidity (sum of $SO_2$, $HNO_3$ and HCl) molar ratio (Fowler et al., 2009).

To look at the UK situation, an analysis of the molar ratios of $NH_3$ to acid gases is presented in Figure 18a. The molar ratio of $NH_3$ to acid gases (sum of $SO_2$, $HNO_3$ and HCl) increased with time, from 1.9 in 2000 to 4.7 in 2015, confirming that $NH_3$ is increasingly in molar excess over atmospheric acidity. The ratio of annual mean molar concentrations of $NH_3$ (80 nmol m$^{-3}$) to $SO_2$ (29 nmol m$^{-3}$) was 2.7 in 2000, which increased in 2015 to 15 (annual mean concentrations of $NH_3$ = 58 nmol m$^{-3}$ *cf.* $SO_2$ = 4 nmol m$^{-3}$). Molar concentrations of $HNO_3$ (4 nmol m$^{-3}$) and HCl (6 nmol m$^{-3}$) were comparable to $SO_2$ in 2015, highlighting the increasing importance of $HNO_3$ and HCl in contributing to atmospheric acidity.  A larger decrease in $SO_2$ (−81 %) than particulate sulphate (−69%) in the AGANet data (Table 4) would appear at first to suggest that the large $NH_3$:$SO_2$ ratio is contributing to a more rapid decrease in $SO_2$ concentrations. However, when the seasalt fraction of $SO_4^{2-}$ is removed from the sulphate trend, the decrease in NSS_SO4 (−78%) is similar to $SO_2$ (−81%) which would suggest that maximum deposition rates for $SO_2$ may have been reached with the smaller $SO_2$ concentrations since 2000."

*23)      p21, l30: '...The increase in ratio of HNO3:NO3- is similar to changes in upward trend in gas-aerosol partitioning between NH3 and NH4+ over time...': what do you call similar? For HNO3/NO3-, the ratio increases by _20%, while for NH3/NH4+, the ratio increases by 100% (according to Fig. 18) ?*

**Author Response:**

Apologies for the ambiguity in the sentence. I simply meant that both sets of data ($HNO_3:NO_3^-$ and $NH_3:NH_4^+$) show an upward trend.

Text revised/expanded in section 3.8.5. Changes in UK chemical climate, paragraph 5.

"A change to an $NH_4NO_3$ rich atmosphere and the potential for $NH_4NO_3$ to release $NH_3$ and $HNO_3$ in warm weather, together with the surfeit of $NH_3$ also means that a larger fraction of the reduced and oxidised N is remaining in the gas phase as $NH_3$ and $HNO_3$. An increased partitioning to the gas phase may account for the larger decrease in particulate $NH_4^+$ (MK −62% between 2000-2015, $n = 12$) and $NO_3^-$ (MK −52% between 2000-2015, $n = 12$) than $NH_3$ (MK −30% between 2000-2015, $n = 12$) and $HNO_3$ (MK −45 % between 2000-2015, $n = 12$) (Table 5) and the increase in gas to aerosol ratios ($NH_3:NH_4^+$ and $HNO_3:NO_3^-$) over the 16 year period (Figure 17). A higher concentration of the gas-phase $HNO_3$ and $NH_3$ may therefore be maintained in the atmosphere than expected on the basis of the emissions trends in $NO_x$ and $NH_3$. Given the larger deposition velocities of $NH_3$ and $HNO_3$ compared to aerosols, more of the $NH_3$ and $HNO_3$ emitted will have the potential to deposit more locally with a smaller footprint within the UK. "

*24)      p22, l11-12, '...a change in the particulate phase from (NH4)2SO4 to NH4NO3. This change is expected to increase residence times of NH3 and HNO3 in the atmosphere...'I am not convinced the shift from ammonium sulphate to ammonium nitrate should increase the residence time, since NH3 and HNO3 will deposit faster (higher deposition velocities) than either aerosol form?*

**Author Response:**

See revised/expanded text in "section 3.6. Seasonal variation in acid gases and aerosols"

Specifically:

"In contrast, the seasonal cycle for particulate $NO_3^-$ is more distinct with a large peak in concentrations that occur every spring, together with a second smaller peak in autumn (Figure 8**Error! Reference source not found.**). $NH_3$, the main neutralising gas in the atmosphere that reacts with $HNO_3$ to form $NH_4NO_3$, has a correspondingly large peak in concentration in spring, a second smaller peak in autumn, but with elevated concentrations in summer and lowest in winter (Figure 9). Although particulate $NO_3^-$ formation is dependent upon the availability of $NH_3$ for reaction with $HNO_3$, its' concentration is also governed by the equilibrium that exists between gaseous $HNO_3$, $NH_3$ and particulate $NH_4NO_3$, the latter of which is appreciably volatile at ambient temperatures (Stelson and Seinfeld, 1982). Partitioning between the gas and aerosol phase is therefore also a key driver for their atmospheric residence times and concentrations. $HNO_3$ and $NH_3$ that are not removed by deposition may react together in the atmosphere to form $NH_4NO_3$, when the concentration product $[NH_3].[HNO_3]$ exceeds equilibrium values, with $NH_4NO_3$ serving as a potential reservoir for the gases. **Since $NH_4NO_3$ is semi-volatile, any that is not dry or wet deposited can potentially dissociate to release $NH_3$ and $HNO_3$, effectively increasing their residence times in the atmosphere**. The formation and dissociation in turn are strongly influenced by ambient temperature and humidity."

*25)      p22, l12 '...expected to increase residence times of NH3 and HNO3 in the atmosphere...' and p22, l15 '...NH3 and NOx emitted will deposit more locally with a smaller footprint...': these two statements appear to contradict each other?*

**Author Response:**

See response to comment 23 above and response to comment 26 after this.

*26) p22, I12: perhaps another way to analyse this trend is to calculate the (changing) linear regression slopes of NO3- vs NH4+ and SO4= vs NH4+, for each year of the 2000-2015 period (as in Fig. 6b), and examine how the two slopes change over time, as an indicator of the fraction of the total NH4+ that neutralizes NO3- and SOA= and its trend over time.*

**Author Response:**

New Figure 18: Long-term changes between 2000 and 2015 in (a) molar ratio of $NH_3$ to acid gases ($SO_2$, $HNO_3$ and HCl) and (b) molar ratio of particulate $NH_4^+$ to acid aerosols ($SO_4^{2-}$ and $NO_3^-$) from measurements made at 12 sites in AGANet.

Text revised/expanded in section 3.8.5. Changes in UK chemical climate (paragraph 3)

Specifically:

To look at the UK situation, an analysis of the molar ratios of $NH_3$ to acid gases is presented in Figure 18a. The molar ratio of $NH_3$ to acid gases (sum of $SO_2$, $HNO_3$ and HCl) increased with time, from 1.9 in 2000 to 4.7 in 2015, confirming that $NH_3$ is increasingly in molar excess over atmospheric acidity. The ratio of annual mean concentrations of $NH_3$ (80 nmol m$^{-3}$) to $SO_2$ (29 nmol m$^{-3}$) was 2.7 in 2000. By 2015, this ratio had increased to 15 (annual mean concentrations of $NH_3$ = 58 nmol m$^{-3}$ *cf* $SO_2$ = 4 nmol m$^{-3}$). Molar concentrations of $HNO_3$ (4 nmol m$^{-3}$) and HCl (6 nmol m$^{-3}$) were comparable to $SO_2$ in 2015, highlighting the increasing importance of $HNO_3$ and HCl in contributing to atmospheric acidity. A larger decrease in $SO_2$ (−81 %) than particulate sulphate (−69 %) in the AGANet data (Table 4) would appear at first to suggest that the large $NH_3$:$SO_2$ ratio is contributing to a more rapid decrease in $SO_2$ concentrations. However, when the seasalt fraction of $SO_4^{2-}$ is removed from the sulphate trend (Sect.3.8.2), the decrease in NSS_SO4 (−78%) is similar to $SO_2$ (−81%) (Table 4). Since the decreasing trend in the ratio of $SO_2$ to $SO_4^{2-}$ also appeared to stabilise after 2006 (Sect.3.8.2), this would suggest that maximum deposition rates for $SO_2$ may have been reached with the smaller $SO_2$ concentrations since 2006.

*27) p24, I14 '...Higher concentrations of the NH3 and HNO3 in the atmosphere will deposit more locally...' But then, NH3 and HNO3 concentrations are actually decreasing; they are not higher than before?*

**Author Response:**

Text revised/expanded in section 3.8.5. Changes in UK chemical climate (paragraph 5)

"A change to an $NH_4NO_3$ rich atmosphere and the potential for $NH_4NO_3$ to release $NH_3$ and $HNO_3$ in warm weather, together with the surfeit of $NH_3$ also means that a larger fraction of the reduced and oxidised N is remaining in the gas phase as $NH_3$ and $HNO_3$. The increased partitioning to the gas phase may account for the larger decrease in particulate $NH_4^+$ (MK −62% between 2000-2015, n=12) and $NO_3^-$ (MK −52% between 2000-2015, n=12) than their gaseous precursors ($NH_3$: MK −30% between 2000-2015, n=12 and $HNO_3$: MK −45 % between 2000-2015, n=12) (Table 5) and the increase in ratios of $NH_3$:$NH_4^+$ and $HNO_3$:$NO_3^-$ over the 16 year period (Figure 15). **A higher concentration of the gas-phase nitrogen species ($HNO_3$ and $NH_3$) may therefore be maintained in the atmosphere than expected on the basis of the emissions trends in $NO_x$ and $NH_3$.** Given the larger deposition velocity of $NH_3$ and $HNO_3$ compared to particulate $NH_4^+$ and $NO_3^-$, more of the $NH_3$ and $HNO_3$ emitted will have the potential to deposit more locally with a smaller footprint within the UK."

*28)      Technical Corrections*
*Units: different units are used. They should either be harmonized, or else each figure should state explicitly what the unit is, especially for the difference between element (N,S) based or molecule (HNO3, SO2) based. For example, mean HNO3 at the Bush site is reported as 0.55 µg m-3 in Fig.2 (average of 0.54 and 0.56 for samplers A and B), while the color code on the concentration map (Fig. 5) indicates a concentration in the range 0.15-0.25, from which I infer that Fig.2 is µg HN03 m-3, while Fig.5 is µg N m-3 ? Similarly, p12, l16, is the Cromwell site HNO3 concentration 1.3 µg HNO3 m-3, or 1.3 µg HNO3-N m-3? From Figure 5 I expect it is the latter (N, not HNO3 as written in the text). Further below, are the SO2 concentrations at Sutton Bonington given as µg SO2 m-3, or in fact µg SO2-S m-3 ? Given that the map in Fig. 5 gives numbers in µg N or µS per m3, it would be good to use the same units. Thus I would recommend to check carefully throughout the text in this paragraph and in the whole paper and make the necessary text changes to eliminate the ambiguity in units.*

**Author Response:**
Thank you – checked and corrected.

*29)      p4, l26-27, delete '...that is also deployed at some CASTnet sites (Rumsey and Walker, 2016).' (already mentioned same page, l14)*
**Author Response:**
OK – deleted.

*30)      p4, l32, suggest change 'temporal' to 'seasonal'*
**Author Response:**
OK. Changed 'temporal' to 'seasonal'

*31)      p5, l9-12: this mostly repeats what was said in the introduction p4, l20-25*
**Author Response:**
OK – sentence below deleted.

*32)      p7, l20, please provide the equation for the calculation of the denuder capture efficiency*
**Author Response:**
Calculation of denuder capture efficiency is described in "section 4 Calculation of air concentrations"
 "The denuder capture efficiency for each of the gas is calculated by comparing the concentrations of the individual gases in the denuder pairs"

Equation is now also provided:

$$\text{Denuder capture efficiency (\% CE)} = 100 \text{ x } \frac{\text{Denuder 1}}{(\text{Denuder 1} + \text{Denuder 2})} \qquad (2)$$

*33)      p35: Figure 2 contains scatter plots and a statistical summary table for the Bush DELTA intercomparison (parallel sampling). It would be good to adapt the same or similar style of display for the other intercomparisons (scatter plots + stats table). Thus for the comparison with ADS (2.7.2), take Fig. S2 out of the supplement and stack it above the statistics given in Table 2. Similarly for the intercomparisons of DELTA vs EMEP TIA/TIN (add statistical table), as is already also done for DELTA vs Bubble/FP Eskdalemuir (Fig 4).*

**Author Response:**

Thank you for suggestions:
Fig S2 and Table 2 combined into Figure 2
Figure 3 (DELTA vs EMEP TIA/TIN), summary stats table added.

There are now however quite a large number of figures.

*34)      p11, sections 2.8 and 3.6: throughout the time series trend analysis, both linear regressions and non-parametric MK tests are used, but as far as I can see, there is essentially no difference between the slopes for any of the pollutant time series. To improve readability and reduce unnecessary redundant information, I would suggest to stick to just one of the methods; it would suffice to say in the methods that both regressions were used and no significant differences were found, and thus henceforth only one regression is displayed.*

**Author Response:**
Thank you for the suggestion.
Figures 13 and 14: graphs with both linear regression and MK analyses moved to supplementary materials.
Replaced by a single Figure 14 showing results of MK analysis only for both time series.

Tables 5 and 6: summary tables comparing LR and MK moved to supplementary section.
Replaced by a single Table 4 showing results of MK analysis only for both time series.

Additional text included at end of <section 2.8 Time series trend analyses
"…but since there was no difference between either tests, MK results only are presented and discussed in the paper. A comparison of trend analyses from both approaches is however provided in supplementary materials (Figures S7, S8 and Tables S4 - S6). "

*35)      p12, l22, '...A peak MONTHLY concentration of...'*
**Author Response:**
Thank you – corrected

*36)      p12,l28, '...expected to be more SPATIALLY homogeneous...'*
**Author Response:**
Thank you – corrected

*37)      p15, l19 '...in summer promotes AEROSOL dissociation...'*
**Author Response:**
Thank you – corrected

*38)      p17, l12 change to '...are available SINCE 1989...'*
**Author Response:**
Thank you – corrected

*39)        p18, l31, '...To QUANTIFY changes...'*
**Author Response:**
Thank you – corrected
(paragraph moved to section 2.7 Time series trend analyses)

*40)        p18, l32, the unit for the annual trend is µg HNO3-N m-3 y-1*
**Author Response:**
The unit is for annual trend is µg $HNO_3$ $m^{-3}$ $y^{-1}$
Units used in trend analysis are on a molecule basis

*41)        p19, l3: '...The LR % annual trends for each time series...' Delete 'annual', since the % reduction are not expressed per year, but over the whole period ? Note that if the concentration reduction were a constant percentage every year, say -10% per year, then the overall time course over 15 years would not look linear, but exponential: if yr1=100, then yr2=90, yr3=81, yr4=72.9, yr5=65.6, ...yr15=20.6*

**Author Response:**
Thank you - text corrected:
"The LR and MK % change in annual mean concentrations for the two time series are estimated from the slope and intercept…."

*42)        p19, l6: same as above, delte 'annual'*
**Author Response:**
Thank you - equation corrected:
$$\% \text{ change} = 100 . \frac{[Yi - Yo)}{Yo}$$

*43)        p21, l17-18 '...The dry deposition... IS known to be enhanced...'*
**Author Response:**
Thank you - corrected

*44)        p23, l9, delete 'from coal combustion'*
**Author Response:**
Thank you - deleted

*45)        p24, l1, '... modest reductionS in HNO3...' (plural)*
**Author Response:**
Thank you - corrected

*46)        p24, l12, '...smaller THAN emission trends...'*
**Author Response:**
Thank you – corrected

*47)      All figures: when the units displayed on axes or legends are given in µg m-3, please specify whether this is on an element basis (NH3-N, HNO3-N, SO2-S) or molecule basis (NH3, HNO3, SO2)*

**Author Response:**

Figure 10a, Y-axis changed to Oxidised N ($\mu$g N m$^{-3}$)

Figure 10b, Y-axis changed to Reduced N ($\mu$g N m$^{-3}$)

Figure 11, Y-axes changed to $SO_2$ ($\mu$g S m$^{-3}$) and $SO_4^{2-}$ ($\mu$g S m$^{-3}$)

To show more clearly that the units are on an element basis ($NH_3$-N, $HNO_3$-N, $SO_2$-S, etc.)

All other figures are on a molecule basis ($NH_3$, $HNO_3$, $SO_2$, etc)– axis and legends should be correct.

*48)      Figure 8: "...Average annual cycles in the ratios of gas:aerosol component concentrations (µg m-3)...' The unit for the ratio is not µg m-3, it must be dimensionless, or mol mol-1?*

**Author Response:**

The Y axis label on the graphs are dimensionless.

In the figure caption, ($\mu$g m$^{-3}$) is the unit of gas and aerosol concentrations that are compared. As this is causing confusion, the caption in Figure 8 (now Figure 9, because Suppl. Figure S2 added as Figure 2) has been revised to:

"Figure 9: Average annual cycles in the ratios of gas:aerosol component concentrations. $HNO_3$, $SO_2$, HCl and aerosol $NO_3^-$, $SO_4^{2-}$, Cl$^-$ data (annual mean, $\mu$g m$^{-3}$) are from the UK Acid Gases and Aerosol Monitoring Network (AGANet). $NH_3$ and $NH_4^+$ data (annual mean, $\mu$g m$^{-3}$) are from the UK National Ammonia Monitoring Network (NAMN, Tang et al., 2018) measured at the same time. Each data point in the graphs represents the mean $\pm$ SD of monthly measurements of 12 sites operational in the network over the period 2000 to 2015.

*49)      Figures 13-14: keep only one of the two trend lines (LR or MK); and delete Fig.14 but add the n=30 datapoints to Fig. 13 as a different symbol shape or color*

**Author Response:**

Figures 13 – 14 replaced with a single figure as suggested by reviewer above.

*50)      Figure 18: the left-hand side panels show the same data as Figs. 13-14 and should therefore not be repeated here.*

**Author Response:**

The left hand panels provides a direct comparison of the concentrations and trends of each of the gas and aerosol pairs ($HNO_3$/ $NO_3^-$, $SO_2$/$SO_4^{2-}$ etc.).

The author agrees with the reviewer that the left hand panels shows the same data as Figures 13 and 14 and they have been removed.

**RESPONSE TO REVIEWER 2**

**Anonymous (Referee)**

The authors thank reviewer 2 for his constructive comments and for taking the time to look at all the details described in the manuscript. We have carefully considered all comments. Please refer to the specific responses.

**General comments**

*1. Currently the manuscript is primarily focused on documenting trends and events and with a smaller focus on the changes in atmospheric composition and pollutant fate due to changes in emissions. This manuscript would benefit from a bit more focus. I suggest focusing more on the trends and how they relate to emission changes and less on specific events captured in the data.*

**Author Response:**
Section 3.8 Assessment of trends in relation to UK emissions" has been revised and expanded to include a more thorough discussion of trends under new sub-headings.
3.8.1 Trends in $HNO_3$ and $NO_3^-$ *vs* $NO_x$ emissions
3.8.2 Trends in $SO_2$ and $SO_4^{2-}$ *vs* $SO_2$ emissions
3.8.3 Trends in $HCl$ and $Cl^-$ *vs* $HCl$ emissions
3.8.4 Trends in $NH_3$ and $NH_4^-$ *vs* $NH_3$ emissions
3.8.5 Changes in UK chemical climate

Discussion on specific events captured in the data have not been revised/truncated as they are important for interpreting anomalies in the trends.

*2. The discussion of the trends of NH3 and HNO3 are sometimes a bit difficult to follow as the change in aerosol composition and loading over the time frame of the measurements impacts the gas phase concentrations. Consider discussing these trends as total nitrate (gaseous HNO3 + aerosol NO3) and NHx (gaseous NH3 + aerosol NH4).*

**Author Response:**
Interactions and partitioning between the gas phase ($SO_2$, $HNO_3$, $NH_3$) and aerosol phase ($SO_4^{2-}$, $NO_3^-$, $NH_4^+$) are important drivers for concentrations and trends in the respective components. Discussion of the gas phase and particulate phase atmospheric components for oxidised and reduced nitrogen, rather than total inorganic nitrate (TIN, sum of gaseous $HNO_3$ + aerosol $NO_3^-$) and total inorganic NHx (TIA, sum of gaseous $NH_3$ + aerosol $NH_4^+$) allows a clearer understanding of the processes occurring in the atmosphere, which drive trends and environmental effects. TIN and TIA is only considered in the manuscript for comparing DELTA with EMEP filter pack measurements at Eskdalemuir.

The expanded "Section 3.8 Assessment of trends in relation to UK emissions" (see response to your comment 1 above) should hopefully provide a clearer discussion on the change in gas and aerosol composition and their interactions in a changing chemical climate.

*3. There are lots of small sections in this manuscript, some consisting of single sentences. Consider combining them into more general sections. Specifically, 2.3.1-2 and 2.5-6.*

**Author Response:**
Following your suggestion:
"*2.3.1 Base coated denuders and filters*" and "*2.3.1 Acid coated denuders and filters*" combined into a single section "2.3.1 Chemically coated denuders and filters"

"*2.5 Data Quality Control*" and "*2.6 Bias correction applied to HNO₃ data*" have not been combined as they cover different aspects.

*4. Many sentences leading paragraphs are structured as "For {atmospheric constituent},. . .". This is a bit formulaic and the authors may want to revise these sentences.*

**Author Response:**

Thank you. We have gone through and revised where appropriate.

**Specific comments**

*1. Abstract: I find the final two sentences of the abstract to be the most compelling. There is a lot of detail, primarily on page 1, that would be better suited for the results section. Consider summarizing the text on the spatial and temporal trends and better connecting them to the changes in atmospheric HNO3 and NH3.*

**Author Response:**

Text revised in abstract.

2. Abstract Page 2 lines 5-6: ". . . indications that the atmospheric lifetime of HNO3 and NH3 has increased . . .". This does not seem correct to me. The lifetime of these gases has not increased but rather the phase/composition of these species have. There are now more gaseous and less aerosol bound NO3 and NH3 due to changes in SO2. This likely decreases the atmospheric lifetime of total nitrate and reduced nitrogen compounds as NH3 and HNO3 typically dry deposit faster than aerosol NO3 and NH4.

**Author Response:**

See also response to Reviewer 1 (comment 32).

Text revised in abstract:

"Since 1999, AGANet has shown substantial decrease in $SO_2$ concentrations relative to $HNO_3$ and $NH_3$, accompanied by large reductions also in the aerosol components, with evidence of a shift in the particulate phase from $(NH_4)_2SO_4$ to $NH_4NO_3$. The potential for $NH_4NO_3$ to act as a reservoir for $NH_3$ and $HNO_3$, together with the surfeit of $NH_3$ means that a larger fraction of the nitrogen is remaining in the gas phase, maintaining higher concentrations of $NH_3$ and $HNO_3$ in the UK. …"

3. Page 4 lines 17-29: This paragraph contains similar information as the previous paragraph. Consider combining it with the previous paragraph

**Author Response:**

See below (replicated information deleted):

Page 4 lines 17-29: "testing, but the high costs and resources required for these measurements make them unsuitable for the assessment of long-term trends at many sites, particularly where spatial patterns are required. To achieve this, a larger number of sites operated at lower time-resolution is needed. In the UK, the Eutrophying and Acidifying Atmospheric Pollutants (UKEAP) network provides long-term measurements for the UK rural atmospheric concentrations and deposition of air pollutants that contribute to acidification and eutrophication processes (Conolly et al., 2016). UKEAP comprises of two EMEP supersites and four component networks: precipitation network (Precip-net), $NO_2$ diffusion tube network ($NO_2$-net), National Ammonia Monitoring Network (NAMN) and the Acid Gases and Aerosol Network (AGANet). At the two EMEP supersites (Auchencorth and Harwell – relocated to Chilbolton in 2016), semi-continuous hourly speciated measurements of reactive gases and aerosols are made with the MARGA system (Twigg et al., 2016) . These measurements are contributing to the validation and improvement of atmospheric models, such as FRAME (Dore et al., 2015) and EMEP4UK (Vieno et al., 2014, 2016) that are used to develop and provide the evidence base for air quality policies, both nationally and internationally. "

Page 5, lines 9 – 12 deleted (repeat of what has already been written on Page 4 lines 17-29)

**Author Response:**
Revised text
Page 6 lines 21-22: "For the base coating, $K_2CO_3$ is used instead of $Na_2CO_3$ (Ferm et al., 1986) to sample acid gases so that the system can also measure aerosol $Na^+$ concentrations. Glycerol increases adhesion, stabilizes the base coating (Ferm, 1986; Finn et al., 2001)……"

Revised to:
"Sodium carbonate ($Na_2CO_3$) is an effective sorbent for acid gases, allowing simultaneous collection of $HNO_3$, $SO_2$ and HCl on denuders (e.g. Ferm 1986). Since the measurement of aerosol $Na^+$ is also of key interest in AGANet however, a potassium carbonate ($K_2CO_3$) coating is used instead to eliminate the possibilities of $Na^+$ contamination from $Na_2CO_3$. Glycerol is added to the $K_2CO_3$ coating, as it increases adhesion……."

5. Sections 2.5: Are data that failed the quality checks removed from the analysis?
**Author Response:**
Sections 2.5 Data Quality Control
"i) Air flow rate ($0.2 – 0.4$ L min$^{-1}$): where this is below the expected range for a sampling period, the data is flagged as valid but failing the QC standard.
ii) Denuder capture efficiency: where this is less than 75% for a sample, the data is flagged as valid but less certain.
iii) Ion balance checks: close agreement expected between $NH_4^+$ and the sum of $NO_3^-$ and $2 \times SO_4^{2-}$, as $NH_3$ is neutralised by $HNO_3$ and $H_2SO_4$ to form $NH_4NO_3$ and $(NH_4)_2SO_4$, respectively (Conolly et al., 2016), and for $Na^+$ and $Cl^-$, as these are marine (sea salt) in origin."

Data failing the above quality check are not automatically removed from analysis.

Air flow rates:
The air pumps used are relatively stable, at $0.2 – 0.4$ l min$^{-1}$. If a low air flow rate is due to temporary loss of power and/or air pump issue, the data is accepted provided that the flow rate does not drop below 25% of the normal range. The data is flagged as valid (EMEP data flag), but has higher uncertainty. If low air flow rate is due to a leak or obstruction (e.g. kinking of tubing), the data is rejected.

Denuder capture efficiency:
Two denuders in series are used for every sample to check capture efficiency for reactive gas: two carbonate denuders for capture of $HNO_3$, $SO_2$ and HCl and two acid coated denuders for capture of NH3. Samples with < 75% of the total gas captured in the first of the two denuders are accepted but are flagged as valid (EMEP data flag) but has higher uncertainty.

Ion balance checks:
**Ratio of $NH_4^+$ (µeq): ($2*SO_4^{2-} + NO_3^-$) (µeq)**
Expect 1:1 as $NH_3$ neutralised by $HNO_3$ and $SO_2$ ($H_2SO_4$) to form $NH_4NO_3$ and $(NH_4)_2SO_4$
Acceptable range = 0.2 to 3.
Ion balance checks are carried out at site level and in collated file (with regression plots for outliers)
Data are rejected if the ratio is outside the range.

**Ratio of Na (µeq):Cl (µeq).**
Expect 1:1 as Na and Cl aerosols derived mainly from sea salt.
Acceptable range 0.2 to 3.
Ion balance checks are carried out at site level and in collated file (with regression plots for outliers)
Data are rejected if the ratio is outside the range.

6. Section 2.5 iv) What is the criteria to determine anomalies and outlies?
**Author Response:**
Sections 2.5 Data Quality Control

"i) Screening the whole dataset for sampling anomalies and outliers, e.g. due to contamination or other issues."

This is a screening process carried out for a small number of runs where there was clearly a sampling malfunction. This exclusion includes events such as vandalized or damaged samples, water ingress or equipment/analytical problems (e.g. mix-up between carbonate and acid coated filters).

7. Section 2.6 Line 22: Does the empirical factor used for HNO3 bias correction exhibit any dependence on season, temperature or solar radiation? If the bias is due to oxidants, then I would expect a dependence in the bias on seasonal and environmental parameters.
**Author Response:**
See response to reviewer 1 (comment 1, pages 1 – 2).

8. Page 9 lines 28-29: The mean difference between the measurements are given here but what is the scatter between the measurements and the median difference. A correlation coefficient would provide some information about the scatter and a median difference would indicate how normal the distribution is and if the bias is being driven by high values in one of the measurement techniques.
**Author Response:**
*Page 9 lines 28-29: "Agreement between the DELTA and ADS was within 19 % for $SO_2$ (mean DELTA = 1.75 $\mu g\ m^{-3}$ cf mean ADS = 2.18 $\mu g\ m^{-3}$) and 4 % for HCl (mean DELTA = 0.40 $\mu g\ m^{-3}$ cf mean ADS = 0.41 $\mu g\ m^{-3}$)."*

Linear regression ($R^2$) is provided in table 2.
Regression plots (DELTA v ADS) were provided in Supplementary materials (Figure S2) – since there were already a lot of figures and tables in the paper.

Supplement Figure S2 and Table 2 has been combined into a single Figure 2 – see response to review 1 (comment 32, page 16).

9. Page 9 line 32: Difference in the instrumentation flow rates and/or inlets could result in the instruments measuring different sized aerosols and my influence the differences in SO4.
**Author Response:**
Page 10 line 3: "A particle size cut-off of around 4.5 µm was estimated for the DELTA air inlet) (Tang et al., 2015)".
Page 10 lines 6 to 7: "In comparison, the ADS has a 2.5 µm cyclone in front of the aerosol filters to collect aerosols < 2.5 µm on the aerosol filters"

ADS size cut-off (cyclone) = 2.5 um @ flow rate of 10 LPM. Note that the cyclone is in front of the aerosol filters, and not at the inlet, as in the case for the URG ADS
(http://www.urgcorp.com/index.php/systems/manual-sampling-systems/annular-denuder-system)

10. Page 16 lines 1-2: The peaks in NHx and SO4 in the spring may just be coincidental. The spring time could also be a time in which the aqueous formation pathway of SO4 is at its maximum or the SO2 emissions from heating or transportation may be larger. In the US, the SO4 concentrations typically peak in the summer while the NH3 concentrations peak in the spring.

**Author Response:**

In the UK, $SO_2$ concentrations are highest in winter (January and February) and lowest in summer (June – August). The peak in $SO_4^{2-}$ concentrations however occur in March every year, where peaks in concentrations of $NH_3$ and $NH_4^+$ from measurements made at the same time from the same sites are also observed (please note that the seasonal cycle of ammonia vary, depending on the emission source types in the vicinity of the measurement site, see Tang et al. 2018).

Formation of $SO_4^{2-}$ is largely governed by the availability of $SO_2$ and $NH_3$ to form the stable $(NH_4)_2SO_4$, and the spring peak may be attributed to enhanced formation of $(NH_4)_2SO_4$, since peaks in concentrations of $NH_3$ and $NH_4^+$ also occur in spring. Import of particulate $(NH_4)_2SO_4$ from long-range transboundary transport also enhances the springtime concentration. In summer, sunny, warmer conditions increases photochemical oxidation of $SO_2$ to $H_2SO_4$ and subsequent formation of sulphate aerosols leading to higher aerosol $SO_4^{2-}$ concentrations in summer than in winter.

Revised/expanded text in "section 3.6 Seasonal variation in acid gases and aerosols (paragraph 5)

"$SO_2$, by contrast, are highest in the winter, with concentrations exceeding summer values on average by a factor of 2 (Figure 8). Increased emissions of $SO_2$ from combustion processes (heating) during the winter months, coupled to stable atmospheric conditions resulting in build-up of concentrations at ground level contributes to the winter maximum. Since the reaction of $SO_2$ with $NH_3$ to form $(NH_4)_2SO_4$ is effectively irreversible (Bower et al., 1997), the ratio of the concentrations of $SO_2$ and $SO_4^{2-}$ (Figure 9) is largely governed by the availability of $SO_2$ and $NH_3$ to form $(NH_4)_2SO_4$. The temporal profile of $SO_4^{2-}$ has a peak in concentrations in spring, although not as pronounced as the $NO_3^-$ peak (Figure 8). The spring peak may be attributed to enhanced formation of $(NH_4)_2SO_4$, since peaks in concentrations of $NH_3$ and $NH_4^+$ also occur in spring (Figure 8) and from the import of particulates from long-range transboundary transport. Unlike $SO_2$, aerosol $SO_4^{2-}$ concentrations are higher in summer than in winter, due to increased photochemical oxidation of $SO_2$ to $H_2SO_4$ and subsequent formation of sulphate aerosols in sunnier and warmer conditions (Mihalopoulos et al., 2007). In winter, lower $SO_2$ oxidation rates limits $H_2SO_4$ formation and therefore also the formation of $(NH_4)_2SO_4$."

**Author Response:**

Yes, indeed rocksalt (NaCl) is used to treat road surfaces in the winter in the UK. About 2 million tonnes are used every year, with most of it going on motorways, trunk roads and main roads.

Aerial salt spray has been shown to increase the concentrations of particulate NaCl in the immediate vicinity of treated road surfaces. For example, a study by Palmer et al. (2004) found that the concentrations of particulate NaCl at the road edge (2m) was 25 - 70% higher than at 10 m away from the road edge, with concentrations at 10 m approaching background NaCl concentrations (*ca*. 35 nmoles m$^{-3}$ from UK AGANet).

Aerial salt spray from treatment of road surfaces in winter is however unlikely to contribute to an increase at regional / national level in the UK. Sources of seasalt aerosol (NaCl) in UK has been shown by high time-resolution measurements (MARGA) in the UK to originate from air masses coming into the UK with the predominant south-westerly winds (seasalt aerosol generated off the atlantic) (pers. comm.).

AGANet sites are all located away from roads, and should not be affected by local seasalt from gritting.

Reference:
Palmer S.C.F, Cape J.N, Truscott A-M, Black H.I.J, Tang Y.S, Swaine, Van Dijk N, Smart S.M, Sutton M.A, Fowler D, Biodiversity in roadside verges: CEH Final Report to SEERAD. 108 pages. 2004.

**Author Response:**
Thank you. "significant" replaced with "substantial"

**Author Response:**
OK, thanks.
Sentence has been deleted.
Changes in $NH_3$ relative to other acid gases are discussed in section 3.8.5 Changes in UK chemical climate.

14. Page 22 line 12: "expected to increase residence times of NH3 and HNO3 in the atmosphere" If we are in an NH3 limited environment, I can see how this would increase HN4NO3 and how that could increase the atmospheric lifetime of HNO3 as it is partitioned to NO3 aerosols. However, I do not see how this increases the NH3 lifetime. NH3 will preferentially partition with SO4, which is more thermodynamically stable than NH4NO3, this should decrease the lifetime of NH3 if anything as the NH4NO3 will evaporate where the (NH4)2SO4 aerosol would not.

**Author Response:**
See revised/expanded text in "3.8.5 Changes in UK chemical climate (paragraph 5 and 6)"

"At the same time, reduction in emissions of the precursor gases have also led to a lower formation of particulate phase $NH_4^+$, $NO_3^-$ and $SO_4^{2-}$ in the atmosphere and changes in atmospheric composition. Since the affinity of $H_2SO_4$ (oxidation product of $SO_2$) for $NH_3$ is much larger than that of HNO3 and HCl, available $NH_3$ is first taken up by $H_2SO_4$ to form ammonium sulphate compounds ($NH_4HSO_4$ and $(NH_4)_2SO_4$), with any excess $NH_3$ then available to react with $HNO_3$ and HCl to form $NH_4NO_3$ and $NH_4Cl$ that are volatile. Analysis of the different particulate components in sect. 3.5**Error! Reference source not found.** showed that the ammonium aerosols are mainly made up of (NH4)2SO4 and $NH_4NO_3$. With the large reduction in $SO_2$, more $NH_3$ is available to react with $HNO_3$ to form $NH_4NO_3$ and concentrations of $NH_4^+$ and $NO_3^-$ are now observed to be in molar excess over $SO_4^{2-}$, providing evidence of a change in the particulate phase from $(NH_4)_2SO_4$ to $NH_4NO_3$ (Figure 18b).

A change to an $NH_4NO_3$ rich atmosphere and the potential for $NH_4NO_3$ to release $NH_3$ and $HNO_3$ in warm weather, together with the surfeit of $NH_3$ also means that a larger fraction of the reduced and oxidised N is remaining in the gas phase as $NH_3$ and $HNO_3$. An increased partitioning to the gas phase may account for the larger decrease in particulate $NH_4^+$ (MK −62% between 2000-2015, $n = 12$) and $NO_3^-$ (MK −52% between 2000-2015, $n = 12$) than $NH_3$ (MK −30% between 2000-2015, $n = 12$) and $HNO_3$ (MK −45 % between 2000-2015, $n = 12$) (Table 5) and the increase in gas to aerosol ratios ($NH_3:NH_4^+$ and $HNO_3:NO_3^-$) over the 16 year period (Figure 17). A higher concentration of the gas-phase $HNO_3$ and $NH_3$ may therefore be maintained in the atmosphere than expected on the basis of the emissions trends in $NO_x$ and $NH_3$. Given the larger deposition velocities of $NH_3$ and $HNO_3$ compared to aerosols, more of the $NH_3$ and $HNO_3$ emitted will have the potential to deposit more locally with a smaller footprint within the UK. "

15. Figure 8: I am happy to see a measure of scatter on these plots as the SD. However, a 5% and 95% CI would be more informative as it would give the reader an idea about the distribution of the data.

**Author Response:**
The 4 graphs in Figure 8 have been replotted showing the 95% CI (T test).

[revised manuscript text omitted]